# Training-Free Rate-Distortion-Perception Traversal With Diffusion

**Yuhan Wang** [1] **Suzhi Bi** [2] **Ying-Jun Angela Zhang** [1]

## Abstract

The rate-distortion-perception (RDP) tradeoff characterizes the fundamental limits of lossy compression by jointly considering bitrate, reconstruction fidelity, and perceptual quality. While recent neural compression methods have improved perceptual performance, they typically operate at fixed points on the RDP surface, requiring retraining to target different tradeoffs. In this work, we propose a training-free framework that leverages pre-trained diffusion models to traverse the entire RDP surface. Our approach integrates a reverse channel coding (RCC) module with a novel score-scaled probability flow ODE decoder. We theoretically prove that the proposed diffusion decoder is optimal for the distortion-perception tradeoff under AWGN observations and that the overall framework with the RCC module achieves the optimal RDP function in the Gaussian case. Empirical results across multiple datasets demonstrate the framework's flexibility and effectiveness in navigating the ternary RDP tradeoff using pre-trained diffusion models. Our results establish a practical and theoretically grounded approach to adaptive, perception-aware compression.

## 1. Introduction

Traditionally, the objective of lossy compression is to represent data using the fewest possible bits while preserving acceptable fidelity to the original data. This can be formalized by Shannon's rate-distortion theory, which characterizes the tradeoff between compression rate and data distortion, e.g., mean squared error (MSE). However, distortion-centric metrics often fail in perceptual domains such as image and video compression. This has led to growing interest in the rate-distortion-perception (RDP) tradeoff (Blau & Michaeli, 2019; Niu et al., 2025), which incorporates perceptual quality into the classical framework, resulting in a ternary tradeoff that better aligns with the goals of modern compression systems. Mathematically, the information RDP function (Blau & Michaeli, 2019) for a source $X \sim p_X$ is defined as

$$R(D, P) = \min_{p_{\hat{X}|X}} I(X; \hat{X})$$
$$\text{s.t. } \mathbb{E}[\Delta(X, \hat{X})] \leq D, \quad d(p_X, p_{\hat{X}}) \leq P, \quad (1)$$

where $\Delta : \mathcal{X} \times \hat{\mathcal{X}} \to \mathbb{R}^+$ is a data distortion measure (e.g., squared error), and $d(\cdot, \cdot)$ is a divergence between probability distributions, such as total variation (TV) divergence or Wasserstein-2 (W2) distance (Panaretos & Zemel, 2020). From an information-theoretic perspective, $R(D, P)$ serves as a *lower bound* on the one-shot achievable rate (Theis & Wagner, 2021) when unlimited common randomness is shared between encoder and decoder.

Understanding and traversing the RDP surface is crucial for building adaptive and user-controllable compression algorithms. A line of work from the information theory community has established coding theorems for the RDP function under various perceptual constraints (Theis & Wagner, 2021; Chen et al., 2022; Yan et al., 2021; Salehkalaibar et al., 2024; Hamdi et al., 2024). Moreover, the universal RDP function studied by Zhang et al. (2021) shows the possibility to fix the encoder and adapt only the decoder to achieve multiple distortion-perception (DP) pairs.

Despite the theoretical potential, existing neural compression methods fall short of flexibly traversing the RDP tradeoff. Approaches like HiFiC (Mentzer et al., 2020), Conditional Diffusion Compression (CDC) (Yang & Mandt, 2023), and DLF (Xue et al., 2025a) operate at fixed tradeoffs, yielding only a single point on the RDP surface per pre-trained model. Recent diffusion based methods (Li et al., 2025a;b; Xue et al., 2025b; Ke et al., 2025; Zhang et al., 2025) utilize pre-trained diffusion models but still require training additional modules or adapters to operate at different points on the RDP surface. While methods such as DiffC (Theis et al., 2022; Vonderfecht & Liu, 2025), Posterior Sampling Compression (PSC) (Elata et al., 2025), Universally Quantized Diffusion Model (UQDM) (Yang et al., 2025), and Denoising Diffusion Codebook Model (DDCM) (Ohayon

[1]Department of Information Engineering, The Chinese University of Hong Kong, Hong Kong SAR, China, <wy023@ie.cuhk.edu.hk>, <yjzhang@ie.cuhk.edu.hk> [2]College of Electronic and Information Engineering, Shenzhen University, Shenzhen, China. Correspondence to: Suzhi Bi <bsz@szu.edu.cn>.

*Proceedings of the 43rd International Conference on Machine Learning*, Seoul, South Korea. PMLR 306, 2026. Copyright 2026 by the author(s).

et al., 2025) offer progressive rate control via adaptive sensing or diffusion encoding, they lack mechanisms to navigate the DP axis. As a result, no existing approach enables full traversal of the RDP tradeoff using one pre-trained model.

In this work, we propose a training-free framework to traverse the RDP surface based on the DiffC scheme, as shown in Figure 1. Specifically, we utilize the reverse channel coding (RCC) module (Li, 2024; Li & Gamal, 2018) to transmit the Gaussian perturbed data, and introduce a flexible ODE decoder powered by pre-trained diffusion models. The framework introduces two intuitive control parameters steering the ternary tradeoff among rate, distortion, and perception. Our contributions can be summarized as follows:

- We introduce a training-free framework that enables flexible traversal of the RDP surface using a pre-trained diffusion model based on DiffC. In particular, we propose a novel score-scaled probability flow ODE (PF-ODE) decoder, enabling single-parameter control of the DP tradeoff for each compression rate. The RCC module introduces a second parameter to control the compression rate.

- We derive new theoretical guarantees for the achievability of DP and RDP functions. We prove that the proposed score-scaled PF-ODE with per-dimension scaling is optimal for the DP tradeoff under additive white Gaussian noise (AWGN) observations in the multivariate Gaussian cases. The full framework with the RCC module achieves the optimal RDP function for scalar Gaussian sources.

- We conduct extensive experiments on CIFAR-10, Kodak, and DIV2K datasets. The results demonstrate superior flexibility and reconstruction quality of our scheme across a wide range of rate, distortion, and perception settings, using a single pre-trained diffusion model.

**Notations:** Let $X$ be a random variable (r.v.) with distribution $p_X(\mathbf{x})$ over the alphabet $\mathcal{X}$. Realizations are denoted by lowercase letters $\mathbf{x}$. The expectation of $X$ is denoted by $\mathbb{E}[X]$. The covariance between r.v.s $X$ and $Y$ is $\text{Cov}[X, Y]$. Matrices are denoted by bold uppercase letters (e.g., $\mathbf{\Sigma}$), with $\text{Tr}(\mathbf{\Sigma})$ and $\mathbf{\Sigma}^{-1}$ representing the trace and inverse, respectively. $H(X)$ and $I(X; Y)$ denote the Shannon entropy and mutual information.

## 2. Background

The existing DiffC scheme (Theis et al., 2022; Vonderfecht & Liu, 2025) provides a way to progressively control the compression rate, yet lacks a mechanism to navigate the DP axis. Our method is rooted in the DiffC scheme but

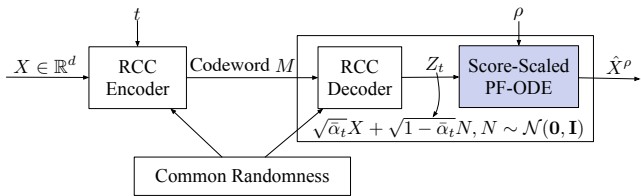

*Figure 1.* The proposed framework to traverse the RDP function using pre-trained diffusion models.

discovers a simple yet effective approach to fully traverse the RDP surface.

The core idea of DiffC is to transmit Gaussian-perturbed data and reconstruct it using a pre-trained diffusion model. In the following subsections, we first provide preliminaries for two key ingredients of the DiffC scheme: diffusion models and the RCC module, followed by a rate-distortion analysis of the existing DiffC framework.

### 2.1. Diffusion Models and Probability Flow ODE:

Diffusion models (or score-based generative models) define a forward process $(\overrightarrow{Z}_\tau)_{\tau \in [0, T_c]}$ that progressively perturbs data $X \in \mathbb{R}^d$ with Gaussian noise, governed by the stochastic differential equation (SDE) (Song et al., 2021b):

$$d\overrightarrow{Z}_\tau = -\frac{1}{2}\beta(\tau)\overrightarrow{Z}_\tau d\tau + \sqrt{\beta(\tau)}dW_\tau, \overrightarrow{Z}_0 = X \sim p_{\text{data}}, \quad (2)$$

where $(W_\tau)_{\tau \in [0, T_c]}$ is standard Brownian motion and $\beta(\tau)$ is the noise schedule. To generate samples from $p_{\text{data}}$, one can reverse the SDE and discretize the resulting process. According to Anderson (1982) and Song et al. (2021b), the reverse SDE associated with Eq. (2) is

$$d\overleftarrow{Z}_\tau = \left[-\frac{1}{2}\beta(\tau)\overleftarrow{Z}_\tau - \beta(\tau)\nabla \log p_{Z_\tau}(\overleftarrow{Z}_\tau)\right]d\tau + \sqrt{\beta(\tau)}d\tilde{W}_\tau,$$
$$\overleftarrow{Z}_{T_c} \sim p_{T_c}, \quad (3)$$

where $(\tilde{W}_\tau)_{\tau \in [0, T_c]}$ is an independent Brownian motion. The score function $\nabla_{\mathbf{z}_\tau} \log p_{Z_\tau}(\mathbf{z}_\tau)$ is approximated by a neural network $s_\theta(\mathbf{z}_\tau, \tau)$ via denoising score matching (Vincent, 2011). Note that the reverse process $\overleftarrow{Z}_\tau$ has the same distribution with $\overrightarrow{Z}_\tau$ for $\tau \in [0, T_c]$.

Meanwhile, there exists a deterministic counterpart to the SDE, known as the *probability flow ODE (PF-ODE)* (Song et al., 2021b), whose trajectories share the same path distribution as the SDE. By manipulating the Fokker-Planck equations (Maoutsa et al., 2020; Särkkä & Solin, 2019), the PF-ODE can be derived as:

$$d\overleftarrow{Z}_\tau = \left[-\frac{1}{2}\beta(\tau)\overleftarrow{Z}_\tau - \frac{1}{2}\beta(\tau)\nabla \log p_{Z_\tau}(\overleftarrow{Z}_\tau)\right]d\tau, \overleftarrow{Z}_{T_c} \sim p_{T_c},$$
$$(4)$$

which can also be simulated using the same score-based model $s_\theta(\mathbf{z}_\tau, \tau)$.

In the DiffC framework, diffusion models serve as *denoising samplers* to reconstruct data from Gaussian-perturbed observations. Specifically, given an input perturbed to a specific noise level, the decoder solves the reverse-time SDE (3) or the PF-ODE (4), starting from the time index corresponding to that noise level. Furthermore, the same diffusion model facilitates progressive coding within the RCC module, as detailed in the subsequent sections.

## 2.2. Reverse Channel Coding

Reverse channel coding (RCC), also known as the channel simulation problem (Li, 2024), addresses the efficient transmission of a sample drawn from a predefined conditional distribution $p_{Z|X}(\cdot|\mathbf{x})$ given data $\mathbf{x}$, using the shortest possible codeword. Practical DiffC implementations (Theis et al., 2022; Vonderfecht & Liu, 2025) adopt the Poisson Functional Representation (PFR) algorithm from Li & Gamal (2018), which achieves the tightest coding length and supports CUDA-accelerated implementation (Vonderfecht & Liu, 2025).

Specifically, in the PFR algorithm, the encoder transmits an index $M$ that allows the decoder to draw a sample from the target conditional distribution $p_{Z|X}(\cdot|\mathbf{x})$ using a shared reference distribution $p_Z$, which is the $Z$-marginal of the joint distribution $p_X p_{Z|X}$. Li & Gamal (2018) proved that the entropy of this index satisfies $H(M) \leq I(X;Z) + \log(I(X;Z) + 1) + 4$, and this upper bound is achievable via entropy coding $M$ using a Zipf distribution.

## 2.3. A Rate-Distortion Analysis of Current DiffC

Theoretically, the existing DiffC scheme supports progressive rate control and achieves *perfect realism*—that is, the distribution of the reconstruction $\hat{X}$ matches the source distribution $p_X$—due to the generative nature of diffusion models (Theis et al., 2022). Thus, standard DiffC operates at a single edge on the RDP plane, providing perfect realism at the cost of relatively high distortion.

Moreover, Theis et al. (2022) conducted a rate-distortion analysis of the DiffC algorithm, characterizing its performance for Gaussian sources. They showed that under the *perfect realism* constraint, reconstruction given by PF-ODE can provably achieve lower MSE than SDE when the source distribution satisfies certain regularity conditions.

*Remark* 2.1. Since the RCC module yields a Gaussian-perturbed observation at the decoder, the subsequent decoding process can be viewed as a specialized denoising problem. While methods for navigating the DP tradeoff have been proposed for general inverse problems (Ohayon et al., 2021; Freirich et al., 2021; Wang et al., 2025), fundamental differences exist between the DP and RDP tradeoffs. Specifically, RDP optimization requires the joint design of

the encoder and decoder, whereas the DP tradeoff in inverse problems assumes a fixed degradation model. Thus, certain optimality results and closed-form expressions derived for the DP tradeoff cannot be directly applied to the RDP setting. We provide a detailed comparison between DP and RDP tradeoffs in Appendix A for further clarification.

## 3. Score-Scaled Probability Flow ODE

While our ultimate objective is to navigate the full RDP tradeoff, in this section we first focus on the DP frontier. We propose a novel decoder design based on a *score-scaled probability flow ODE (PF-ODE)*, as highlighted in the blue module of Figure 1. This mechanism enables flexible traversal along the DP axis for any given compression level.

Note that in DiffC algorithm, the RCC module generates the noisy observation $Z_t$ indexed by the diffusion time $t$. Thus, the decoder is tasked with a specialized *denoising problem* for each compression rate, where a series of DP tradeoffs must be addressed. Mathematically, the DP function with MSE distortion and Wasserstein-2 distance at time $t$ is defined as (Blau & Michaeli, 2018)

$$D_t(P) = \min_{p_{\hat{X}|Z_t}} \mathbb{E}[\|X - \hat{X}\|_2^2], \quad \text{s.t.} \quad W_2^2(p_X, p_{\hat{X}}) \leq P, \quad (5)$$

where $W_2^2(\cdot, \cdot)$ denotes the squared Wasserstein-2 distance between two distributions. In the following, we detail the proposed score-scaled PF-ODE and analyze its optimality for the DP tradeoff.

### 3.1. Conditional and Marginal Distributions of Score-Scaled PF-ODE

The original PF-ODE in Eq. (4) yields reconstructions with perfect perception ($P = 0$) at cost of relatively high distortion (Theis et al., 2022). For general inverse problems, Xue et al. (2025c) showed that the reverse mean propagation process by casting out the randomness in a conditional version of Eq. (3) converges to the minimum MSE (MMSE) estimation.

In Appendix C.1, we further show that the mean propagation process of the reverse SDE in Eq. (3) starting from time index $t$ also converges to the MMSE estimation for Gaussian sources, which is equivalent to applying a scale on the score term in Eq. (4). Motivated by these observations, we propose the following *score-scaled PF-ODE* to bridge the two extremes:

$$d\overleftarrow{Z}_\tau = \Big[ -\frac{1}{2}\beta(\tau)\overleftarrow{Z}_\tau - \frac{1}{2}(2 - \rho)\beta(\tau)\nabla \log p_{Z_\tau}(\overleftarrow{Z}_\tau) \Big] d\tau,$$
$$\overleftarrow{Z}_t \sim \sqrt{\bar{\alpha}_t}X + \sqrt{1 - \bar{\alpha}_t}N, \quad (6)$$

where $\rho \in [0, 1]$. When $\rho = 1$, the above score-scaled PF-ODE collapses to the original PF-ODE (4) and recovers $X$ with perfect perception. When $\rho = 0$, it corresponds to

the mean propagation process and converges to the MMSE estimate for Gaussian sources. Using Euler-Maruyama discretization (Särkkä & Solin, 2019) and following approximations similar to those in (Song et al., 2021b), we can discretize Eq. (6) into $T$ intervals and simulate it via the following iterations:

$$Z_k = \frac{1}{\sqrt{1-\beta_{k+1}}}\Big(Z_{k+1}+\frac{1}{2}(2-\rho)\beta_{k+1}\nabla\log p_{Z_{k+1}}(Z_{k+1})\Big),$$
$$k \in \{0,1,\ldots,t-1\}, \quad (7)$$

starting from an instance $Z_t \sim \sqrt{\bar{\alpha}_t}X + \sqrt{1-\bar{\alpha}_t}N$ for $t \in \{1,\ldots,T\}$. Here, $\beta_T \geq \cdots \geq \beta_0 = 0$ is the variance schedule, $\alpha_k = 1 - \beta_k$, and $\bar{\alpha}_k = \prod_{i=1}^k \alpha_i$. The discretization details are provided in Appendix B.

We now investigate whether score-scaling enables flexible and effective control over the DP axis for any given compression rate. To this end, we first analyze the conditional and marginal distributions of the reconstructions produced by the proposed score-scaled PF-ODE for Gaussian sources. We then prove its optimality for the DP tradeoff in the multivariate Gaussian case in the next subsection.

**Lemma 3.1.** *Consider the multivariate Gaussian source* $X \sim \mathcal{N}(\boldsymbol{\mu}_0, \boldsymbol{\Sigma}_0)$. *Let* $\boldsymbol{\mu}_k = \sqrt{\bar{\alpha}_k}\boldsymbol{\mu}_0$ *and* $\boldsymbol{\Sigma}_k = \bar{\alpha}_k\boldsymbol{\Sigma}_0 + (1-\bar{\alpha}_k)\mathbf{I}$ *for* $k \in \{0,\ldots,t\}$. *Starting from* $Z_t \sim \sqrt{\bar{\alpha}_t}X + \sqrt{1-\bar{\alpha}_t}N$, $N \sim \mathcal{N}(\mathbf{0},\mathbf{I})$ *and applying the score-scaled PF-ODE iterations in Eq. (7), the* conditional *reconstruction given* $Z_t = \check{\mathbf{z}}_t$ *is* $Z_0(\check{\mathbf{z}}_t) = \mathbf{A}_t^\rho\check{\mathbf{z}}_t + \mathbf{B}_t^\rho\boldsymbol{\mu}_0$, *where* $\mathbf{A}_t^\rho := \sqrt{\bar{\alpha}_t}\boldsymbol{\Sigma}_0\prod_{i=0}^{t-1}\big(\mathbf{I}+\frac{1}{2}\rho\frac{\beta_{i+1}}{\alpha_{i+1}}\boldsymbol{\Sigma}_i^{-1}\big)\boldsymbol{\Sigma}_t^{-1}$ *and* $\mathbf{B}_t^\rho := \frac{1}{2}(2-\rho)\big(\sum_{i=2}^t\bar{\alpha}_{i-1}\beta_i\boldsymbol{\Sigma}_0\prod_{j=0}^{i-2}(\mathbf{I}+\frac{1}{2}\rho\frac{\beta_{j+1}}{\alpha_{j+1}}\boldsymbol{\Sigma}_j^{-1})\boldsymbol{\Sigma}_{i-1}^{-1}\boldsymbol{\Sigma}_i^{-1} + \beta_1\boldsymbol{\Sigma}_1^{-1}\big)$ *as* $T \to \infty$. *In particular, the reconstruction* $Z_0(\check{\mathbf{z}}_t) = \mathbf{A}_t^0\check{\mathbf{z}}_t + \mathbf{B}_t^0\boldsymbol{\mu}_0$ *can be proved to be the MMSE point when* $\rho = 0$. *When* $\rho = 1$, *the reconstruction matches the optimal transport solution with* $\mathbf{A}_t^1 = \boldsymbol{\Sigma}_0^{\frac{1}{2}}\boldsymbol{\Sigma}_t^{-\frac{1}{2}}$. *Then, the* marginal *distribution of* $Z_0$ *is given by*

$$Z_0 \sim \mathcal{N}\Big(\boldsymbol{\mu}_0,\ \boldsymbol{\Sigma}_0\prod_{i=0}^{t-1}\big(\rho\mathbf{I}+(1-\rho)\alpha_{i+1}\boldsymbol{\Sigma}_i\boldsymbol{\Sigma}_{i+1}^{-1}\big)\Big),$$

*as* $T \to \infty$. *When* $\rho = 0$, *the variance is* $\bar{\alpha}_t\boldsymbol{\Sigma}_0^2\boldsymbol{\Sigma}_t^{-1}$. *When* $\rho = 1$, *the marginal distribution of* $Z_0$ *is* $\mathcal{N}(\boldsymbol{\mu}_0,\boldsymbol{\Sigma}_0)$, *which coincides with that of the original source* $X$.

*Proof.* See the details in Appendix C. $\qquad\square$

From Lemma 3.1, we can observe that the reconstruction is deterministic for a fixed $Z_t = \mathbf{z}_t$, with behavior controlled by $\rho$ and $t$. For the marginal distribution, the mean remains fixed at $\boldsymbol{\mu}_0$, regardless of $\rho$ and $t$. When $\rho$ increases, the variance of $Z_0$ gradually approaches that of the original source $X$, leading to improved perception but increased distortion.

## 3.2. Optimal Distortion-Perception Tradeoff Through AWGN Channel

In Proposition 3.2, we first present the optimal solution to Eq. (5) for the multivariate Gaussian case. Then, we show that our score-scaled PF-ODE achieves this optimal DP tradeoff under AWGN.

**Proposition 3.2** (Freirich et al. (2021)). *Consider the $d$-dimensional source* $X \sim \mathcal{N}(\boldsymbol{\mu}_0, \boldsymbol{\Sigma}_0)$ *and the additive white Gaussian noise (AWGN) observation* $Z_t \sim \sqrt{\bar{\alpha}_t}X + \sqrt{1-\bar{\alpha}_t}N$, $N \sim \mathcal{N}(\mathbf{0},\mathbf{I})$. *Assume* $\boldsymbol{\Sigma}_0$ *admits an eigendecomposition* $\boldsymbol{\Sigma}_0 = \mathbf{Q}\boldsymbol{\Lambda}_0\mathbf{Q}^\top$ *with positive eigenvalues* $\boldsymbol{\Lambda}_0 = \mathrm{diag}(\lambda_1,\ldots,\lambda_d)$. *Then the optimal solution to Eq. (5) yields*

$$D_t = \Big(\sqrt{\sum_{i=1}^d\frac{\lambda_i}{\lambda_i^{(t)}}\big(\sqrt{\lambda_i^{(t)}}-\sqrt{\bar{\alpha}_t}\sqrt{\lambda_i}\big)^2}-\sqrt{P}\Big)^2+\sum_{\ell=1}^d\frac{(1-\bar{\alpha}_t)\lambda_\ell}{\lambda_\ell^{(t)}},$$
$$(8)$$

*for* $0 \leq P < \sum_{i=1}^d\frac{\lambda_i}{\lambda_i^{(t)}}\big(\sqrt{\lambda_i^{(t)}}-\sqrt{\bar{\alpha}_t}\sqrt{\lambda_i}\big)^2$, *where* $\lambda_\ell^{(t)} := \bar{\alpha}_t\lambda_\ell + (1-\bar{\alpha}_t)$.

*Proof.* In Freirich et al. (2021), the authors derived the optimal DP tradeoff and the optimal estimators for multivariate Gaussian sources. Our results are equivalent to theirs in the case of AWGN degradations. The closed-form expression in Eq. (8) and Lemma D.2 in our proof offer insights into the achievability of our newly proposed method. Therefore, we include the proof in Appendix D.2 for completeness. $\square$

**Theorem 3.3.** *The optimal DP tradeoff (8) can be achieved by simulating the score-scaled PF-ODE iterations in Eq. (7), with a dedicated choice of $\rho$ for each dimension of $Z_t$.*

*Proof.* In the achievability proof, we use a per-dimension variant of Eq. (7), which applies independently to each component of $Z_t$. See details in Appendix D.3. $\square$

*Remark* 3.4. In Freirich et al. (2021), the optimal estimators are constructed as linear combinations of two extreme-point estimators (i.e., MMSE and perfect realism), without specifying how to obtain these extremes. In contrast, our achievability proof provides a unified, concrete mechanism that not only covers these two extremes *across all compression rates* but also facilitates continuous control over the entire DP tradeoff.

## 4. Traversing RDP Function: Optimality and General Algorithm

This section presents a joint analysis of the RDP performance achieved by combining the RCC module with the proposed score-scaled PF-ODE. We prove that this framework provides an optimal solution to the RDP tradeoff for

scalar Gaussian sources. Furthermore, we detail practical algorithms for general sources that enable flexible traversal of the RDP surface using a pre-trained diffusion model.

Recall that we first transmit an instance of $Z_t \sim \sqrt{\bar{\alpha}_t}X + \sqrt{1-\bar{\alpha}_t}N$, $N \sim \mathcal{N}(\mathbf{0}, \mathbf{I})$ using the RCC algorithm (e.g., PFR) for selected compression level $t$. Then the decoder can simulate the score-scaled PF-ODE iterations in Eq. (7) with $\rho \in [0, 1]$ to determine the balance between distortion and perception under each compression rate.

The following theorem shows that, despite its modularity and flexibility, our scheme achieves the information RDP function (1) in the scalar Gaussian case.

**Theorem 4.1.** *Consider the scalar Gaussian source $X \sim \mathcal{N}(\mu_0, \sigma_0^2)$. The information RDP function under MSE distortion and squared W2 distance is given by*

$$R(D, P) = \begin{cases} \frac{1}{2}\log \frac{\sigma_0^2(\sigma_0 - \sqrt{P})^2}{\sigma_0^2(\sigma_0 - \sqrt{P})^2 - (\sigma_0^2 + (\sigma_0 - \sqrt{P})^2 - D)^2/4} \\ \qquad \text{if } \sqrt{P} < \sigma_0 - \sqrt{|\sigma_0^2 - D|}, \\ \max\{\frac{1}{2}\log \frac{\sigma_0^2}{D}, 0\} \\ \qquad \text{if } \sqrt{P} \geq \sigma_0 - \sqrt{|\sigma_0^2 - D|}. \end{cases}$$

*Let $D_t^\rho$ and $P_t^\rho$ be the squared error distortion and squared W2 distance achieved by the score-scaled PF-ODE iterations (7) for $\rho \in [0, 1]$ at compression level $t \in \{1, 2, \ldots, T\}$. Then we have*

$$R(D_t^\rho, P_t^\rho) \leq R_t^1 \leq R(D_t^\rho, P_t^\rho) + \log\left(R(D_t^\rho, P_t^\rho) + 1\right) + 4,$$
$$R_t^\infty = R(D_t^\rho, P_t^\rho),$$

*where $R_t^1$ and $R_t^\infty$ are the one-shot and asymptotic coding rate of the Poisson functional representation (Li & Gamal, 2018) to transmit samples of $Z_t \sim \sqrt{\bar{\alpha}_t}X + \sqrt{1-\bar{\alpha}_t}N$.*

*Proof.* The closed-form expression of the RDP function is derived in Zhang et al. (2021). In Appendix E, we show that the proposed score-scaled PF-ODE combined with the PFR encoder achieves $(R, D, P)$ triplets that asymptotically match the theoretical RDP function. $\square$

Figure 2 illustrates Theorem 4.1 by plotting the theoretical RDP surface alongside the empirical $(R, D, P)$ triplets achieved by our method. For this visualization, we omit entropy coding and directly compute the coding rate of the RCC algorithm. It can be observed that the empirical points, generated by iteratively simulating Eq. (7) for various values of $\rho$ and $t$, align closely with the theoretical RDP curves.

Theorem 4.1 establishes the optimality of our scheme for the scalar Gaussian case. For multivariate Gaussian sources, achievable RDP bounds can be derived by assigning dimension-specific parameters $t$ and $\rho$. According to the converse proof of the vector Gaussian RDP function (Qian

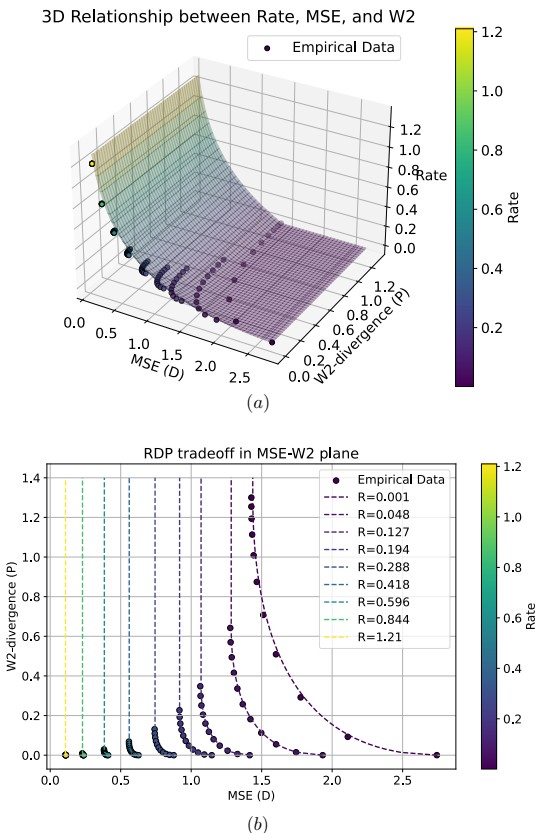

*Figure 2.* Information-theoretical RDP function for scalar Gaussian source (dashed line) and achieved rate, MSE, and W2 distance levels by our scheme (solid dots). (a) The RDP surface. (b) $R(D, P)$ function along DP planes. Different colors represent different rates.

et al., 2025), achieving the information-theoretic optimum may require different $t$ and $\rho$ across dimensions based on joint optimization, e.g., generalized reverse water-filling methods. A rigorous theoretical analysis of our scheme in high-dimensional settings remains a subject for future work. Nevertheless, we demonstrate the empirical effectiveness of our method on real-world datasets in Section 5.

We now detail practical algorithms for applying our method to general high-dimensional sources using pre-trained diffusion models in Algorithms 1 and 2. As discussed in (Theis et al., 2022; Vonderfecht & Liu, 2025), directly transmitting $Z_t \sim \sqrt{\bar{\alpha}_t}X + \sqrt{1-\bar{\alpha}_t}N$, $N \sim \mathcal{N}(\mathbf{0}, \mathbf{I})$ via RCC is challenging due to the intractability of $p_{Z_t}$ for complex sources $X$. Instead, we approximate the conditional distributions $p_{Z_k|Z_{k+1}}$ using a pre-trained diffusion model and progressively transmit $Z_k$ conditioned on $Z_{k+1}$ and $X$.

Note that we use the PFR algorithm for illustration in Algorithms 1 and 2. Other RCC algorithms (including sample-based methods and dithered quantization) can also be employed, provided that samples following the distribution of $\sqrt{\bar{\alpha}_t}X + \sqrt{1-\bar{\alpha}_t}N$ can be generated or approximated.

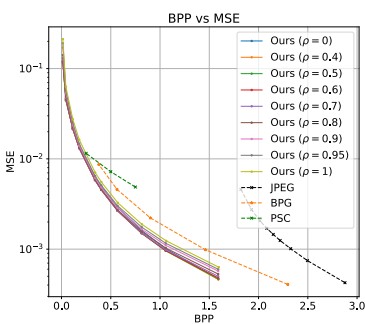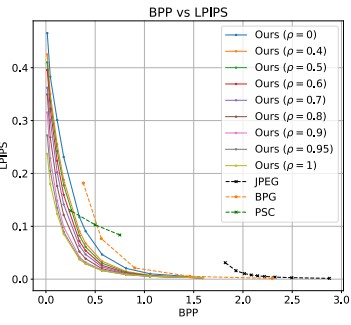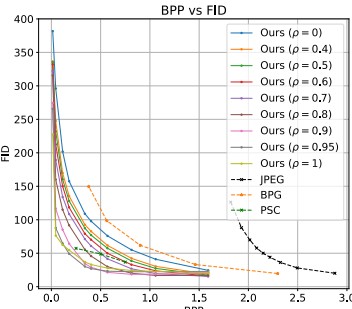

*Figure 3.* Effects of controlling $t$ and $\rho$ on different metrics for the CIFAR-10 dataset. Distortion is quantified by MSE, and perception is measured by LPIPS and FID.

*Remark* 4.2. Given all $(D, P)$ pairs associated with a fixed rate (e.g., one dashed line in Figure 2(b)), we can fix the time index $t$ at the encoder and adapt only the score-scaling parameter $\rho$ in the decoder to meet all these DP constraints. This provides great operational flexibility, as the encoder can be designed and deployed without prior knowledge of the specific DP requirements.

---

**Algorithm 1** Encoder

---

1: **Input:** Pre-trained model $p_\theta(\mathbf{z}_k|\mathbf{z}_{k+1})$, target time index $t \in \{1, 2, \dots, T\}$, source data $\mathbf{x}$
2: $\mathbf{z}_T \leftarrow$ An instance following $p_{Z_T} = \mathcal{N}(\mathbf{0}, \mathbf{I})$
3: **for** $k = T-1, \dots, t+1, t$ **do**
4: $\quad \triangleright$ Send $\mathbf{z}_k \sim p_{Z_k|Z_{k+1},X}(\cdot|\mathbf{z}_{k+1}, \mathbf{x})$ using $p_\theta(\mathbf{z}_k|\mathbf{z}_{k+1})$. Here we use PFR as an illustration.
5: $\quad W_1, W_2, \cdots \sim \text{Exp}(1), \; S_n = \sum_{i=1}^n W_i$
6: $\quad \bar{\mathbf{z}}_k^{(1)}, \bar{\mathbf{z}}_k^{(2)}, \cdots \sim p_\theta(\mathbf{z}_k|\mathbf{z}_{k+1})$
7: $\quad c_k = \arg\min_{n \in \mathbb{N}_+} S_n \frac{p_\theta(\bar{\mathbf{z}}_k^{(n)}|\mathbf{z}_{k+1})}{p_{Z_k|Z_{k+1},X}(\bar{\mathbf{z}}_k^{(n)}|\mathbf{z}_{k+1}, \mathbf{x})}$
8: $\quad$ **Output:** Entropy code and send $c_k$ to Decoder.
9: **end for**

---

---

**Algorithm 2** Decoder

---

1: **Input:** Pre-trained diffusion model $p_\theta(\mathbf{z}_k|\mathbf{z}_{k+1})$, $t \in \{1, 2, \dots, T\}$, score-scaling parameter $\rho$
2: $\mathbf{z}_T \leftarrow$ An instance following $p_{Z_T} = \mathcal{N}(\mathbf{0}, \mathbf{I}) \triangleright$ Using shared seed.
3: **for** $k = T-1, \dots, t+1, t$ **do**
4: $\quad \bar{\mathbf{z}}_k^{(1)}, \bar{\mathbf{z}}_k^{(2)}, \cdots \sim p_\theta(\mathbf{z}_k|\mathbf{z}_{k+1}) \triangleright$ Using shared seed.
5: $\quad \mathbf{z}_k = \bar{\mathbf{z}}_k^{(c_k)}$ after receiving $c_k$
6: **end for**
7: $\hat{\mathbf{x}}_t^\rho \leftarrow$ Simulate Eq. (7) given $Z_t = \mathbf{z}_t$ with chosen $\rho$.
8: **Output:** $\hat{\mathbf{x}}_t^\rho$

---

# 5. Experimental Results

In this section, we demonstrate the flexibility and effectiveness of the proposed framework through experiments conducted on high-dimensional, real-world datasets.

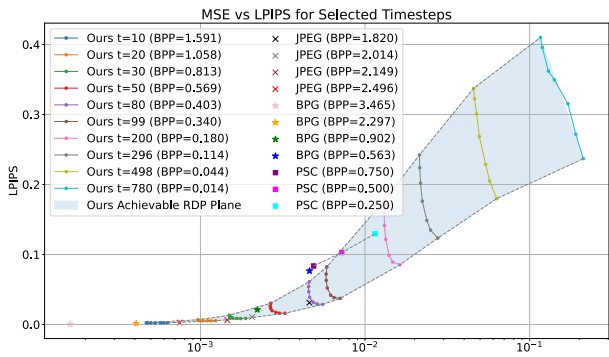

*Figure 4.* Rate-distortion-perception curves on the CIFAR-10 dataset. Distortion levels are quantified by MSE and perception levels are measured by LPIPS.

## 5.1. CIFAR-10 Dataset

We begin with the CIFAR-10 dataset to validate our theoretical findings and illustrate the effectiveness of adjusting both the diffusion time index $t$ in the PFR algorithm and the score-scaling parameter $\rho$ in the proposed PF-ODE iterations (7). We directly employ a pre-trained diffusion model from a third-party repository[1], which is a PyTorch implementation following the details in (Ho et al., 2020).

**Baselines:** We benchmark our method against two traditional codecs, JPEG and BPG, and a posterior sampling-based diffusion compression method, PSC (Elata et al., 2025). PSC utilizes adaptive compressed sensing with a pre-trained diffusion model for flexible-rate compression. Notably, both our method and PSC share the same diffusion model backbone, ensuring a fair comparison.

**Effect of RDP Control Parameters $t$ and $\rho$:** Figure 3 illustrates the impact of varying $t$ and $\rho$ on distortion and perception metrics. As expected, for a fixed $t$, increasing $\rho$ improves perceptual metrics (lower LPIPS and FID) at the expense of higher distortion (increased MSE), aligning with our theoretical predictions. Meanwhile, decreasing $t$ results

---

[1] https://github.com/w86763777/pytorch-ddpm

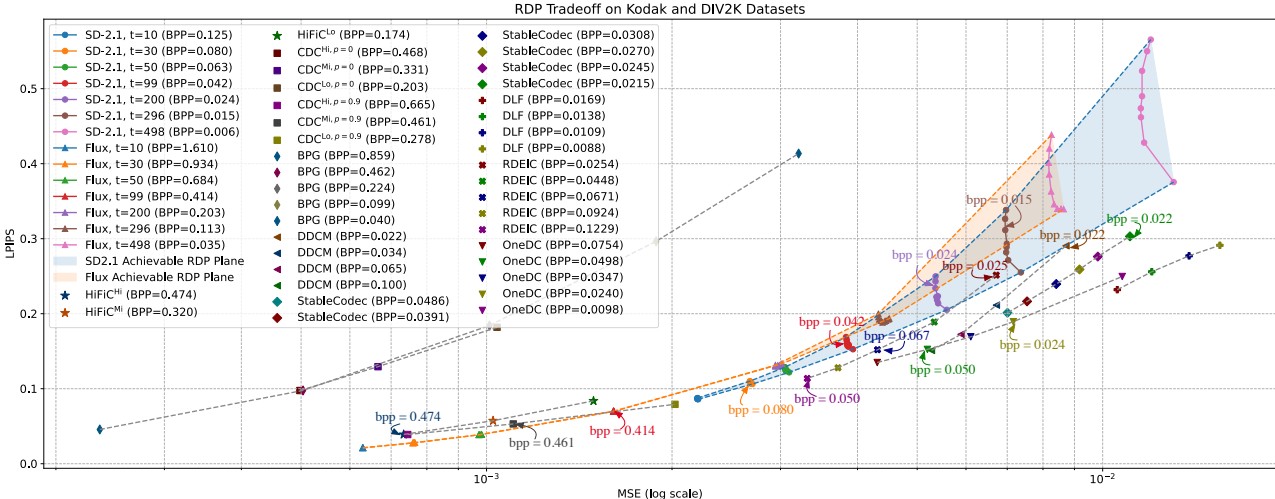

*Figure 5.* RDP tradeoff traversed by our proposed scheme on the Kodak and DIV2K datasets. We show the results obtained with Stable Diffusion (SD) 2.1 and the Flux model, respectively. More tradeoffs measured in different metrics (e.g., PSNR and FID) can be found in Appendix F.2.2.

in a higher bitrate (BPP), leading to improvements in both distortion and perception metrics. Our scheme achieves lower distortion and superior perceptual quality compared to JPEG, BPG, and PSC at comparable bitrates.

**Achievable RDP Regions:** We also plot the RDP regions in Figure 4 by varying $t$ and $\rho$. The curves are constructed by connecting points with the same $t$ but different $\rho$. They are convex and shift toward the lower-left as $t$ decreases, indicating improved tradeoffs at higher bitrates. These observations confirm the theoretical analysis. Our scheme provides *full* control over rate, distortion, and perception using a single pre-trained model, whereas PSC only supports adaptive rate control.

Experimental details and additional results (e.g., PSNR-FID tradeoffs), as well as visual illustrations, are provided in Appendix F.1.

### 5.2. Kodak and DIV2K Datasets

In this subsection, we further evaluate our method on the Kodak and DIV2K datasets (Agustsson & Timofte, 2017). The Kodak dataset consists of 24 high-quality $768 \times 512$ images, while the DIV2K validation set contains 100 high-resolution images. We aggregate both datasets to form our comprehensive test set.

**Pre-trained Diffusion Models:** We utilize various open-source diffusion models, including multiple versions of Stable Diffusion (SD) (Rombach et al., 2022) and Flux (Black-Forest-Labs et al., 2025). We adopt the CUDA-accelerated PFR algorithm from Vonderfecht & Liu (2025). In Appendix F.2.2, we provide a detailed comparison of the compression and reconstruction performances across different

models (e.g., SD-1.5, SD-2.1, SD-XL, and Flux).

Notably, both Stable Diffusion and Flux are *latent* diffusion models, which operate in the latent space of a pre-trained autoencoder. Although our theoretical results are derived for the original source space, experimental results demonstrate that our scheme remains effective in the latent space. Specifically, it can flexibly navigate the RDP surface in the image space by adjusting the time index $t$ and the score-scaling parameter $\rho$ in the latent space. The influence of these two parameters on RDP metrics is consistent with the trends observed in the CIFAR10 dataset. Corresponding figures and tables are provided in Appendix F.2.2.

**Baselines:** We compare our method against several baselines. For HiFiC (Mentzer et al., 2020), CDC (Yang & Mandt, 2023), and DLF (Xue et al., 2025a), separate models trained with different weighted objectives are required to operate at different points on the RDP plane. We use their official implementations and released pretrained models. StableCodec (Zhang et al., 2025), OneDC (Xue et al., 2025b), and RDEIC (Li et al., 2025b) leverage pretrained diffusion models but require additional trained modules or adapters that must be retrained or tuned for each RDP operating point. For each method, we use the official implementation and corresponding pretrained modules, with the same diffusion backbone when applicable. DDCM (Ohayon et al., 2025) and original DiffC (Theis et al., 2022; Vonderfecht & Liu, 2025) use pretrained diffusion models and can adapt to different rates without retraining, but they lack a mechanism to control the DP tradeoff. Note that the original DiffC is a special case of our framework at $\rho = 1$.

**Achievable RDP Regions:** Figure 5 visualizes the RDP tradeoff traversed by our proposed scheme on the Kodak

*Table 1.* Comparison of different methods in terms of model size and latency on Kodak dataset. Both our approach and DDCM utilize SD-2.1 as the backbone diffusion model. Notably, our framework is *training-free*. By leveraging a single pre-trained model, we can traverse a wide range of RDP tradeoffs, thereby significantly reducing training time and model storage requirements.

|  | Param. | Enc. (s) | Dec. (s) | Total (s) |
|---|---|---|---|---|
| HiFiC(Mentzer et al., 2020) | 181M | 0.67 | 1.53 | 2.20 |
| CDC(Yang & Mandt, 2023) | 53.8M | 0.07 | 3.25 | 3.32 |
| OneDC(Xue et al., 2025b) | 1.4B | 0.31 | 0.43 | 0.74 |
| StableCodec(Zhang et al., 2025) | 1.1B | 0.27 | 0.47 | 0.74 |
| Ours | 950M | 0.22-9.14 | 0.33-2.10 | 2.31-9.47 |
| DDCM(Ohayon et al., 2025) | 950M | 37.76 | 37.91 | 75.67 |

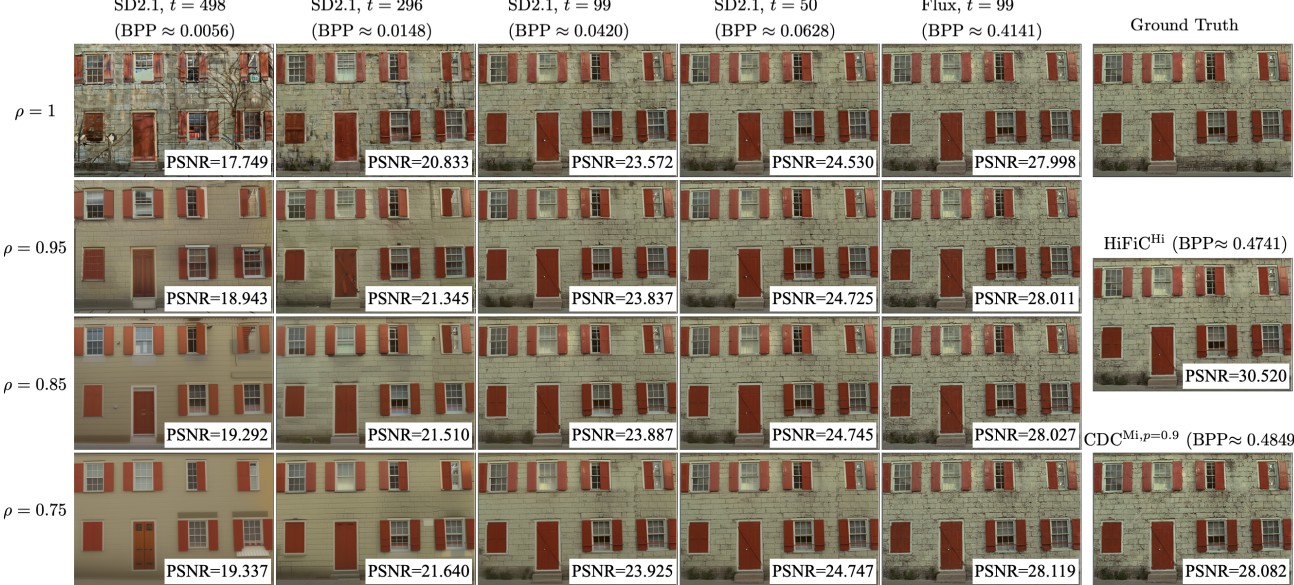

*Figure 6.* Samples from HiFiC, CDC, and our proposed schemes with different $\rho$ and $t$.

and DIV2K datasets. As $t$ decreases, the RDP curves shift downward and shrink, indicating that the tension between distortion and perception is more pronounced at lower bitrates. At higher bitrates, reconstructions converge toward the original images, achieving both low distortion and high perceptual quality. Note that the diffusion process is applied in the latent space, thus the supported $\rho$ ranges may differ from the $[0, 1]$ interval defined in Theorem 3.3.

HiFiC and CDC occasionally outperform in one metric due to their targeted training losses, and DDCM achieves lower LPIPS at the cost of much higher BPP. However, these methods often suffer from performance degradation in other dimensions and lack the flexibility to navigate the RDP tradeoff dynamically.

**Model Sizes and Latencies:** We report the model sizes and encoding/decoding latencies of different methods in Table 1. The encoding/decoding time is measured on Kodak dataset. Note that the running time of the proposed scheme is the same as DiffC (Vonderfecht & Liu, 2025). Although it is slower than some lightweight models like HiFiC, it remains

acceptable. Meanwhile, our scheme is compatible with any RCC coding method. Thus, the encoding time can be reduced by using more efficient RCC coding methods. The decoding time can also be reduced by employing improved diffusion model sampling methods.

Furthermore, our framework is *training-free*. With a single pre-trained model, we can cover a wide range of RDP tradeoffs, thereby saving significant training time and model storage costs. For example, to cover 10 different bitrates and 5 different distortion-perception tradeoffs, HiFiC or CDC would need to train and store 50 distinct models, resulting in a storage cost several times larger than that of our method.

**Visualizations:** We provide visual examples under various settings in Figure 6 to illustrate the RDP tradeoff. *In the low-BPP regime*, a high $\rho$ yields perceptually pleasing images with vivid colors and sharp edges; however, the details may be unfaithful to the original source. Lowering $\rho$ suppresses hallucinated details, producing reconstructions that are smoother with reduced distortion. *At higher bitrates*, reconstructions become both sharp and faithful across all

$\rho$ values. More samples can be found in Appendix F.2.2. We also provide a demo website[2] to compare the details of *high-resolution images* across different $t$ and $\rho$.

We include the experimental details with suggested choices of $\rho$ in Appendix F.2.1. Meanwhile, additional experimental results, including more perceptual metrics (e.g., FID) and more visualizations can be found in Appendix F.2.2.

In summary, our method provides an efficient way to control RDP tradeoff with one pre-trained model and two parameters $t$ and $\rho$. In practice, users can select a suitable bitrate according to resource restrictions and adjust the DP balance according to their specific needs without retraining.

## 6. Discussions

**On Gaussian Assumption:** Our theoretical optimality analysis is based on Gaussian assumptions, which may not hold for complex real-world data. However, analyzing the Gaussian case is the standard, necessary first step to establish theoretical optimality and provides a principled derivation for why the proposed score scaling controls the tradeoff.

As Qian et al. (2025) demonstrated, achieving strict RDP optimality for non-isotropic vector-Gaussian sources requires generalized reverse water-filling. Within our framework, this corresponds to assigning dimension-specific noise levels ($t$) at the encoder and varying scaling parameters ($\rho$) at the decoder. Implementing exact water-filling dimension-by-dimension in practical high-resolution diffusion models is computationally prohibitive. Thus, designing efficient approximations and analyzing their theoretical properties remains an interesting direction for future work.

**On Latent Diffusion:** While our theoretical results are derived in the original source space, our experiments on the Kodak and DIV2K datasets are conducted within the latent space of pre-trained autoencoders. In practice, latent diffusion models (such as SD2.1) use a KL-regularized latent space to approximate a standard Gaussian (Rombach et al., 2022), bridging our theory with empirical practice.

Statistical properties in the latent space correlate with those in pixel space. Thus, the effectiveness of our method in the latent space can be attributed to the fact that preserving the distribution or reconstructing the mean in the latent space effectively achieves the same goals in the pixel space. Still, a rigorous theoretical analysis of our scheme operating explicitly within the latent space remains an open question.

**On Model Size and Latency:** Currently, our method is slower than some lightweight models or one-step diffusion distillation methods, but it remains acceptable. The comparison highlights a fundamental tradeoff in current generative

compression: **highly optimized, task-specific distillation** (e.g., StableCodec and OneDC) versus **training-free, universal flexibility** (Ours). We believe both directions are highly valuable. Moreover, their effective storage cost is much higher if flexibility is required. For example, to cover the same wide range of bitrates and DP tradeoffs that our single model traverses, one would need to retrain and store dozens of separate modules ($N \times 492$M for OneDC, and $N \times 109$M for StableCodec, where $N$ is the number of RDP points to be traversed), making their practical deployment size prohibitively large for adaptive streaming scenarios.

Furthermore, our framework's latency can be mitigated through both engineering and methodological improvements. Because it is derived from the reverse ODE and remains agnostic to the specific solver or RCC method employed, it can seamlessly integrate future advancements. Adopting advanced few-step ODE solvers (e.g., DPM-Solver), diffusion distillation techniques, or highly optimized RCC implementations will directly accelerate our method. We leave the design of efficient solvers and RCC methods for our framework as future work.

## 7. Conclusions

In this paper, we introduced a training-free framework to traverse the complete RDP tradeoff in lossy compression, leveraging the RCC module and the proposed score-scaled PF-ODE decoder. Our theoretical analysis established the optimality of the diffusion decoder for DP tradeoffs in the multivariate Gaussian case, and of the full framework for the RDP function in scalar Gaussian settings. Empirical evaluations on the real-world datasets validated the method's ability to flexibly and effectively balance bitrate, distortion, and perceptual quality using pre-trained diffusion models. This work offers a practical and theoretically principled solution for adaptive compression across the entire RDP surface, using a single pre-trained diffusion model.

## Acknowledgments

The work of Yuhan Wang and Ying-Jun Angela Zhang is supported in part by the General Research Fund (project number 14214122, 14202723, 14207624), Area of Excellence Scheme grant (project number AoE/E-601/22-R), and NSFC/RGC Collaborative Research Scheme (project number CRS_HKUST603/22, CRS_HKU702/24), all from the Research Grants Council of Hong Kong. The work of Suzhi Bi is supported in part by the Guangdong Basic and Applied Basic Research Foundation under Project 2024B1515020089, and in part by Shenzhen Science and Technology Program under Project JCYJ20250604181912016.

---

[2]https://diffrdp.github.io/

## Impact Statement

This paper presents work whose goal is to advance the field of machine learning, in particular by enabling more flexible and efficient lossy compression techniques. There are many potential societal consequences of our work, none of which we feel must be specifically highlighted here.

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

# A. On RDP and DP Tradeoffs

## A.1. Rate-Distortion-Perception Tradeoff in Lossy Compression

**Information RDP function:** Mathematically, the information RDP function (Blau & Michaeli, 2019) for a source $X \sim p_X$ is defined as

$$R(D, P) = \min_{p_{\hat{X}|X}} I(X; \hat{X})$$

$$\text{s.t. } \mathbb{E}[\Delta(X, \hat{X})] \le D, \quad d(p_X, p_{\hat{X}}) \le P,$$

where $\Delta : \mathcal{X} \times \hat{\mathcal{X}} \to \mathbb{R}^+$ is a data distortion measure (e.g., square-error), and $d(\cdot, \cdot)$ is a divergence between probability distributions, such as total variation (TV) divergence or Wasserstein-2 (W2) distance (Panaretos & Zemel, 2020). From an information-theoretic perspective, $R(D, P)$ serves as a *lower bound* on the one-shot achievable rate (Theis & Wagner, 2021) when unlimited common randomness is shared between encoder and decoder. Converse and achievability results have been established under various assumptions on shared randomness (Wagner, 2022) and realism constraints (Chen et al., 2022; Hamdi et al., 2024; Salehkalaibar et al., 2024). Readers may refer to (Niu et al., 2025) for a comprehensive survey on the RDP function.

Closed-form expressions of $R(D, P)$ have been derived for binary sources with Hamming distortion and TV divergence (Blau & Michaeli, 2019), as well as scalar Gaussian sources under MSE distortion and W2 distance (Zhang et al., 2021). For multivariate Gaussian cases, (Qian et al., 2025) solved the RDP function via an extended reverse water-filling algorithm. In this paper, we focus on the practical design to traverse the RDP function for general sources.

**Universal RDP function:** Zhang et al. (2021) further reveal the potential to fix an encoder and only adapt the decoder to meet multiple distortion-perception pairs $(D, P) \in \Theta$. For example, $\Theta$ could be the set of all $(D, P)$ pairs associated with a given rate along the information RDP function. Zhang et al. (2021) demonstrated that for the scalar Gaussian distribution, the theoretical rate required by a fixed encoder to achieve all $(D, P)$ pairs in $\Theta$ is exactly $\sup_{(D,P)\in\Theta} R(D, P)$, where $R(D, P)$ is the information RDP function defined in Eq. (1).

## A.2. Distortion-perception Tradeoff in Image Restoration

A related problem to the RDP function is the distortion-perception (DP) tradeoff in image restoration (Blau & Michaeli, 2018; Freirich et al., 2021). Given a noisy observation $Y$ of the source $X$, the DP function is defined as

$$D = \min_{p_{\hat{X}|Y}} \mathbb{E}[\Delta(X, \hat{X})] \quad \text{s.t.} \quad d(p_X, p_{\hat{X}}) \le P. \tag{9}$$

Note that the denoising problem is fundamentally different from the lossy compression problem, as the observation is fixed and there is no rate constraint.

Regarding the closed-form expressions in Eq. (9), Blau & Michaeli (2018) derived the DP function for scalar Gaussian sources under KL divergence and MSE distortion, assuming the estimator is linear. Freirich et al. (2021) studied multivariate Gaussian sources under W2 distance and MSE distortion, deriving the optimal estimator and the corresponding DP function. Wang et al. (2025) considered a *conditional* DP tradeoff given a fixed observation, assuming the source follows certain conditional distributions; in contrast, the proposed score-scaled ODE module in this paper considers the *unconditional* DP tradeoff, where the stochasticity introduced by the RCC encoder is explicitly accounted for.

Towards traversing the DP tradeoff in general inverse problems, Ohayon et al. (2021) proposed a posterior sampling method based on conditional GANs. The balance between distortion and perception can be controlled by adjusting the noise variance input to the generator or by averaging multiple posterior samples. Freirich et al. (2021) investigated the DP tradeoff in the Wasserstein space and proved that the optimal estimators can be constructed by linear combinations of two extremes: the MMSE estimator and a perfect perception sampler with minimum distortion. However, acquiring these idealized models in practice is often challenging. Wang et al. (2025) also considered a diffusion-based posterior sampling method to traverse the *conditional* DP tradeoff, but those results cannot be directly applied to the *unconditional* DP tradeoff and the lossy compression problem studied in this work.

Note that there are fundamental differences between the DP tradeoff in image restoration and the RDP tradeoff in lossy compression. Some optimality results for the DP tradeoff do not directly extend to the RDP tradeoff due to the presence of rate constraints—for example, the closed-form expressions for multivariate Gaussian sources.

*Table 2.* Comparison of related schemes in image restoration and lossy compression problems. Here the rate or DP control means the ability to adjust the rate or DP tradeoff with a single pre-trained model.

| | | Rate Control | DP Control | Proved Optimality in Gaussian |
|---|---|---|---|---|
| Image Restoration | Ohayon et al. (2021) | / | ✓ | ✗ |
| | Wang et al. (2025) | / | ✓ | ✓ |
| | Freirich et al. (2021) | / | ✓ | ✓ |
| | Our ODE decoder | / | ✓ | ✓ |
| Lossy Compression | HiFiC(Mentzer et al., 2020) | ✗ | ✗ | ✗ |
| | CDC(Yang & Mandt, 2023) | ✗ | ✗ | ✗ |
| | DDCM(Ohayon et al., 2025) | ✓ | ✗ | ✗ |
| | DiffC(Theis et al., 2022) | ✓ | ✗ | at perfect realism |
| | Ours | ✓ | ✓ | scalar Gaussian |

In Table 2, we present a comparison of related works concerning image restoration and lossy compression problems, focusing on DP and RDP tradeoffs. Note that our ODE decoder offers an advantage in flexibility in the lossy compression framework: a single model can adaptively operate across all varying noise levels and solve a series of denoising problems.

## B. Discretization of Score-Scaled PF-ODE

Consider the following time-reversed SDE starting from a particular time index $t \in [0, T_c]$, which is the proposed score-scaled PF-ODE in Eq. (6) with an additional scaled Brownian-motion term:

$$d\overleftarrow{Z}_\tau = \left[ -\frac{1}{2}\beta(\tau)\overleftarrow{Z}_\tau - \frac{1}{2}(2-\rho)\beta(\tau)\nabla \log p_{Z_\tau}(\overleftarrow{Z}_\tau) \right]d\tau + \sqrt{\lambda\beta(\tau)}dW_\tau, \ \overleftarrow{Z}_t \sim \sqrt{\bar{\alpha}_t}X + \sqrt{1-\bar{\alpha}_t}N. \quad (10)$$

To analyze the probabilistic behavior of the proposed score-scaled PF-ODE in Eq. (6), we can study the above SDE at $\lambda \to 0$. By Euler-Maruyama discretization (Särkkä & Solin, 2019), we can divide the whole time interval $[0, T_c]$ into $T$ equal segments with step size $\Delta\tau = \frac{T_c}{T}, \tau = k\Delta\tau$. The noise schedule in the discrete time space is given by $\beta_k = \beta(k\Delta\tau)\Delta\tau$ for $k = 0, 1, \cdots, T$. Following the similar approximation with (Song et al., 2021b, Appendix E), the discretized version of the above SDE can be written as

$$Z_k = (2 - \sqrt{1-\beta_{k+1}})Z_{k+1} + \frac{1}{2}(2-\rho)\beta_{k+1}\nabla \log p_{Z_{k+1}}(Z_{k+1}) + \sqrt{\lambda\beta_{k+1}}\epsilon$$

$$\overset{(a)}{\approx} \left(2 - (1 - \frac{1}{2}\beta_{k+1})\right)Z_{k+1} + \frac{1}{2}(2-\rho)\beta_{k+1}\nabla \log p_{Z_{k+1}}(Z_{k+1}) + \sqrt{\lambda\beta_{k+1}}\epsilon$$

$$\overset{(b)}{\approx} (1 + \frac{1}{2}\beta_{k+1})Z_{k+1} + \frac{1}{2}(2-\rho)\beta_{k+1}\nabla \log p_{Z_{k+1}}(Z_{k+1}) + \frac{1}{2}(2-\rho)\beta_{k+1}^2\nabla \log p_{Z_{k+1}}(Z_{k+1}) + \sqrt{\lambda\beta_{k+1}}\epsilon$$

$$= (1 + \frac{1}{2}\beta_{k+1})\left(Z_{k+1} + \frac{1}{2}(2-\rho)\beta_{k+1}\nabla \log p_{Z_{k+1}}(Z_{k+1})\right) + \sqrt{\lambda\beta_{k+1}}\epsilon$$

$$\overset{(c)}{\approx} \frac{1}{\sqrt{1-\beta_{k+1}}}\left(Z_{k+1} + \frac{1}{2}(2-\rho)\beta_{k+1}\nabla \log p_{Z_{k+1}}(Z_{k+1})\right) + \sqrt{\lambda\beta_{k+1}}\epsilon,$$

where $(a)$ and $(c)$ are due to the equivalent infinitesimal $\sqrt{1-\beta_k} \approx 1 - \frac{1}{2}\beta_k$ and $\frac{1}{\sqrt{1-\beta_k}} \approx 1 + \frac{1}{2}\beta_k$, and $(b)$ is given by eliminating the $\mathcal{O}(\beta_k^2)$ term when $\Delta\tau \to 0$. Here $\epsilon \sim \mathcal{N}(\mathbf{0}, \mathbf{I})$.

## C. Proof of Lemma 3.1

Consider the source $X \sim \mathcal{N}(\boldsymbol{\mu}_0, \boldsymbol{\Sigma}_0)$, and the decoder receives $Z_t = \sqrt{\bar{\alpha}_t}X + \sqrt{1-\bar{\alpha}_t}N$, $N \sim \mathcal{N}(\mathbf{0}, \mathbf{I})$, which has distribution $p_{Z_t}(\mathbf{z}_t) = \mathcal{N}(\sqrt{\bar{\alpha}_t}\boldsymbol{\mu}_0, \bar{\alpha}_t\boldsymbol{\Sigma}_0 + (1 - \bar{\alpha}_t)\mathbf{I})$. Denote the marginal distribution of $Z_k$ for $k = t, t-1, \cdots, 0$ as

$$p_{Z_k}(\mathbf{z}_k) = \mathcal{N}(\underbrace{\sqrt{\bar{\alpha}_k}\boldsymbol{\mu}_0}_{:=\boldsymbol{\mu}_k}, \underbrace{\bar{\alpha}_k\boldsymbol{\Sigma}_0 + (1 - \bar{\alpha}_k)\mathbf{I}}_{:=\boldsymbol{\Sigma}_k}),$$

then $\nabla_{\mathbf{z}_k} \log p_{Z_k}(\mathbf{z}_k) = \boldsymbol{\Sigma}_k^{-1}(\boldsymbol{\mu}_k - \mathbf{z}_k)$. For $k = t-1, \ldots, 0$, the discretized time-reverse score-scaled SDE (10) provides us with

$$
\begin{aligned}
\mathbf{z}_k &= \frac{1}{\sqrt{1-\beta_{k+1}}}\Big(\mathbf{z}_{k+1} + \frac{1}{2}(2-\rho)\beta_{k+1}\nabla_{\mathbf{z}_{k+1}} \log p_{Z_{k+1}}(\mathbf{z}_{k+1})\Big) + \sqrt{\lambda\beta_{k+1}}\epsilon \\
&= \frac{1}{\sqrt{1-\beta_{k+1}}}\Big(\mathbf{z}_{k+1} + \frac{1}{2}(2-\rho)\beta_{k+1}\boldsymbol{\Sigma}_{k+1}^{-1}(\boldsymbol{\mu}_{k+1} - \mathbf{z}_{k+1})\Big) + \sqrt{\lambda\beta_{k+1}}\epsilon \\
&= \frac{1}{\sqrt{\alpha_{k+1}}}\Big(\boldsymbol{\Sigma}_{k+1} - \frac{1}{2}(2-\rho)\beta_{k+1}\mathbf{I}\Big)\boldsymbol{\Sigma}_{k+1}^{-1}\mathbf{z}_{k+1} + \frac{1}{\sqrt{\alpha_{k+1}}}\frac{1}{2}(2-\rho)\beta_{k+1}\boldsymbol{\Sigma}_{k+1}^{-1}\boldsymbol{\mu}_{k+1} + \sqrt{\lambda\beta_{k+1}}\epsilon \\
&= \sqrt{\alpha_{k+1}}\Big(\boldsymbol{\Sigma}_k + \frac{1}{2}\rho\frac{\beta_{k+1}}{\alpha_{k+1}}\mathbf{I}\Big)\boldsymbol{\Sigma}_{k+1}^{-1}\mathbf{z}_{k+1} + \frac{1}{2}(2-\rho)\beta_{k+1}\sqrt{\bar{\alpha}_k}\boldsymbol{\Sigma}_{k+1}^{-1}\boldsymbol{\mu}_0 + \sqrt{\lambda\beta_{k+1}}\epsilon,
\end{aligned}
\tag{11}
$$

which implies the following conditional distribution of $Z_k$ given $Z_{k+1}$:

$$
p_{Z_k|Z_{k+1}}(\mathbf{z}_k|\mathbf{z}_{k+1}) = \mathcal{N}(\mathbf{U}_{k+1}^\rho \mathbf{z}_{k+1} + \mathbf{V}_{k+1}^\rho \boldsymbol{\mu}_0, \ \lambda\beta_{k+1}\mathbf{I}),
\tag{12}
$$

where $\mathbf{U}_{k+1}^\rho := \sqrt{\alpha_{k+1}}\Big(\boldsymbol{\Sigma}_k + \frac{1}{2}\rho\frac{\beta_{k+1}}{\alpha_{k+1}}\mathbf{I}\Big)\boldsymbol{\Sigma}_{k+1}^{-1}$ and $\mathbf{V}_{k+1}^\rho := \frac{1}{2}(2-\rho)\beta_{k+1}\sqrt{\bar{\alpha}_k}\boldsymbol{\Sigma}_{k+1}^{-1}$ for $k = t-1, t-2\ldots, 0$. The proposed score-scaled PF-ODE corresponds to the case $\lambda \to 0$.

### C.1. Conditional Distributions of $Z_0$ given $Z_t = \check{\mathbf{z}}_t$

**Lemma C.1.** *(Bishop, 2006, Section 2.3.3) Given a marginal Gaussian distribution for $X$ and a conditional Gaussian distribution for $Y$ given $X = \mathbf{x}$ in the form*

$$
p_X(\mathbf{x}) = \mathcal{N}\left(\boldsymbol{\mu}, \boldsymbol{\Lambda}^{-1}\right),
$$
$$
p_{Y|X}(\mathbf{y} \mid \mathbf{x}) = \mathcal{N}\left(\mathbf{A}\mathbf{x} + \mathbf{b}, \mathbf{L}^{-1}\right),
$$

*the marginal distribution of $Y$ and the conditional distribution of $X$ given $Y$ are given by*

$$
p_Y(\mathbf{y}) = \mathcal{N}\left(\mathbf{A}\boldsymbol{\mu} + \mathbf{b}, \mathbf{L}^{-1} + \mathbf{A}\boldsymbol{\Lambda}^{-1}\mathbf{A}^\top\right)
$$
$$
p_{X|Y}(\mathbf{x} \mid \mathbf{y}) = \mathcal{N}\left(\boldsymbol{\Sigma}\left\{\mathbf{A}^\top\mathbf{L}(\mathbf{y} - \mathbf{b}) + \boldsymbol{\Lambda}\boldsymbol{\mu}\right\}, \boldsymbol{\Sigma}\right),
$$

*where*

$$
\boldsymbol{\Sigma} = \left(\boldsymbol{\Lambda} + \mathbf{A}^\top\mathbf{L}\mathbf{A}\right)^{-1}.
$$

Starting from a sample $Z_t = \check{\mathbf{z}}_t$, we perform the reverse process in Eq. (11) and compute the conditional distribution of the reconstruction. Considering the above Lemma C.1, together with $p_{Z_{t-1}|Z_t=\check{\mathbf{z}}_t}(\mathbf{z}_{t-1}) = \mathcal{N}(\mathbf{U}_t^\rho \check{\mathbf{z}}_t + \mathbf{V}_t^\rho \boldsymbol{\mu}_0, \ \lambda\beta_t\mathbf{I})$ and $p_{Z_{t-2}|Z_{t-1}}(\mathbf{z}_{t-2}|\mathbf{z}_{t-1}) = \mathcal{N}(\mathbf{U}_{t-1}^\rho \mathbf{z}_{t-1} + \mathbf{V}_{t-1}^\rho \boldsymbol{\mu}_0, \ \lambda\beta_{t-1}\mathbf{I})$, we have

$$
\begin{aligned}
p_{Z_{t-2}|Z_t=\check{\mathbf{z}}_t}(\mathbf{z}_{t-2}) &= \mathcal{N}\Big(\mathbf{U}_{t-1}^\rho\big(\mathbf{U}_t^\rho \check{\mathbf{z}}_t + \mathbf{V}_t^\rho \boldsymbol{\mu}_0\big) + \mathbf{V}_{t-1}^\rho \boldsymbol{\mu}_0, \ \lambda\beta_{t-1}\mathbf{I} + \mathbf{U}_{t-1}^\rho \lambda\beta_t \mathbf{U}_{t-1}^{\rho\top}\Big) \\
&= \mathcal{N}\Big(\mathbf{U}_{t-1}^\rho\mathbf{U}_t^\rho \check{\mathbf{z}}_t + \big(\mathbf{U}_{t-1}^\rho\mathbf{V}_t^\rho + \mathbf{V}_{t-1}^\rho\big)\boldsymbol{\mu}_0, \ \lambda\beta_{t-1}\mathbf{I} + \mathbf{U}_{t-1}^\rho \lambda\beta_t \mathbf{U}_{t-1}^{\rho\top}\Big),
\end{aligned}
$$

where the coefficients can be further expressed as

$$
\begin{aligned}
\mathbf{U}_{t-1}^\rho\mathbf{U}_t^\rho &= \sqrt{\alpha_{t-1}}\Big(\boldsymbol{\Sigma}_{t-2} + \frac{1}{2}\rho\frac{\beta_{t-1}}{\alpha_{t-1}}\mathbf{I}\Big)\boldsymbol{\Sigma}_{t-1}^{-1} \cdot \sqrt{\alpha_t}\Big(\boldsymbol{\Sigma}_{t-1} + \frac{1}{2}\rho\frac{\beta_t}{\alpha_t}\mathbf{I}\Big)\boldsymbol{\Sigma}_t^{-1} \\
&= \sqrt{\alpha_{t-1}}\sqrt{\alpha_t}\Big(\boldsymbol{\Sigma}_{t-2} + \frac{1}{2}\rho\frac{\beta_{t-1}}{\alpha_{t-1}}\mathbf{I}\Big)\Big(\mathbf{I} + \frac{1}{2}\rho\frac{\beta_t}{\alpha_t}\boldsymbol{\Sigma}_{t-1}^{-1}\Big)\boldsymbol{\Sigma}_t^{-1},
\end{aligned}
$$

and

$$
\begin{aligned}
\big(\mathbf{U}_{t-1}^\rho\mathbf{V}_t^\rho + \mathbf{V}_{t-1}^\rho\big) &= \sqrt{\alpha_{t-1}}\Big(\boldsymbol{\Sigma}_{t-2} + \frac{1}{2}\rho\frac{\beta_{t-1}}{\alpha_{t-1}}\mathbf{I}\Big)\boldsymbol{\Sigma}_{t-1}^{-1} \cdot \frac{1}{2}(2-\rho)\beta_t\sqrt{\bar{\alpha}_{t-1}}\boldsymbol{\Sigma}_t^{-1} + \frac{1}{2}(2-\rho)\beta_{t-1}\sqrt{\bar{\alpha}_{t-2}}\boldsymbol{\Sigma}_{t-1}^{-1} \\
&= \frac{1}{2}(2-\rho)\alpha_{t-1}\sqrt{\bar{\alpha}_{t-2}}\beta_t\Big(\boldsymbol{\Sigma}_{t-2} + \frac{1}{2}\rho\frac{\beta_{t-1}}{\alpha_{t-1}}\mathbf{I}\Big)\boldsymbol{\Sigma}_{t-1}^{-1}\boldsymbol{\Sigma}_t^{-1} + \frac{1}{2}(2-\rho)\beta_{t-1}\sqrt{\bar{\alpha}_{t-2}}\boldsymbol{\Sigma}_{t-1}^{-1}.
\end{aligned}
$$

Now we prove the general case by induction. Suppose that in step $k$, $p_{Z_k|Z_t=\check{\mathbf{z}}_t}(\mathbf{z}_k)$ has a Gaussian distribution with mean

$$\mathbf{U}_{k+1}^\rho \mathbf{U}_{k+2}^\rho \cdots \mathbf{U}_t^\rho \check{\mathbf{z}}_t + \Big(\mathbf{U}_{k+1}^\rho \cdots (\mathbf{U}_{t-1}^\rho \mathbf{V}_t^\rho + \mathbf{V}_{t-1}^\rho) \cdots + \mathbf{V}_{k+1}^\rho\Big)\boldsymbol{\mu}_0, \tag{13}$$

and variance

$$\lambda \sum_{i=k+1}^t \beta_i \Big(\prod_{j=k+1}^{i-1} \mathbf{U}_j^\rho\Big)\Big(\prod_{j=k+1}^{i-1} \mathbf{U}_j^\rho\Big)^\top,$$

where the coefficients of $\check{\mathbf{z}}_t$ and $\boldsymbol{\mu}_0$ in Eq. (13) can be expressed as

$$\mathbf{U}_{k+1}^\rho \mathbf{U}_{k+2}^\rho \cdots \mathbf{U}_t^\rho = \Big(\prod_{i=k+1}^t \sqrt{\alpha_i}\Big)\Big(\boldsymbol{\Sigma}_k + \frac{1}{2}\rho\frac{\beta_{k+1}}{\alpha_{k+1}}\mathbf{I}\Big)\prod_{i=k+1}^{t-1}\Big(\mathbf{I} + \frac{1}{2}\rho\frac{\beta_{i+1}}{\alpha_{i+1}}\boldsymbol{\Sigma}_i^{-1}\Big)\boldsymbol{\Sigma}_t^{-1},$$

and

$$\Big(\mathbf{U}_{k+1}^\rho \cdots (\mathbf{U}_{t-1}^\rho \mathbf{V}_t^\rho + \mathbf{V}_{t-1}^\rho) \cdots + \mathbf{V}_{k+1}^\rho\Big)$$
$$=\frac{1}{2}(2-\rho)\sqrt{\bar{\alpha}_k}\Big(\sum_{i=k+2}^t \Big(\prod_{j=k+1}^{i-1}\alpha_j\Big)\beta_i\Big(\boldsymbol{\Sigma}_k + \frac{1}{2}\rho\frac{\beta_{k+1}}{\alpha_{k+1}}\mathbf{I}\Big)\Big(\prod_{j=k+1}^{i-2}\Big(\mathbf{I} + \frac{1}{2}\rho\frac{\beta_{j+1}}{\alpha_{j+1}}\boldsymbol{\Sigma}_j^{-1}\Big)\Big)\boldsymbol{\Sigma}_{i-1}^{-1}\boldsymbol{\Sigma}_i^{-1} + \beta_{k+1}\boldsymbol{\Sigma}_{k+1}^{-1}\Big).$$

Again, with Lemma C.1 and $p_{Z_{k-1}|Z_k}(\mathbf{z}_{k-1}|\mathbf{z}_k) = \mathcal{N}(\mathbf{U}_k^\rho \mathbf{z}_k + \mathbf{V}_k^\rho \boldsymbol{\mu}_0, \lambda\beta_k\mathbf{I})$, we can obtain that $p_{Z_{k-1}|Z_t=\check{\mathbf{z}}_t}(\mathbf{z}_{k-1})$ has mean

$$\mathbf{U}_k^\rho\Big(\mathbf{U}_{k+1}^\rho \mathbf{U}_{k+2}^\rho \cdots \mathbf{U}_t^\rho \check{\mathbf{z}}_t + \Big(\mathbf{U}_{k+1}^\rho \cdots (\mathbf{U}_{t-1}^\rho \mathbf{V}_t^\rho + \mathbf{V}_{t-1}^\rho) \cdots + \mathbf{V}_{k+1}^\rho\Big)\boldsymbol{\mu}_0\Big) + \mathbf{V}_k^\rho\boldsymbol{\mu}_0$$
$$=\mathbf{U}_k^\rho \mathbf{U}_{k+1}^\rho \mathbf{U}_{k+2}^\rho \cdots \mathbf{U}_t^\rho \check{\mathbf{z}}_t + \Big(\mathbf{U}_k^\rho\Big(\mathbf{U}_{k+1}^\rho \cdots (\mathbf{U}_{t-1}^\rho \mathbf{V}_t^\rho + \mathbf{V}_{t-1}^\rho) \cdots + \mathbf{V}_{k+1}^\rho\Big) + \mathbf{V}_k^\rho\Big)\boldsymbol{\mu}_0,$$

and variance

$$\lambda\beta_k + \mathbf{U}_k^\rho\lambda\sum_{i=k+1}^t \beta_i\Big(\prod_{j=k+1}^{i-1}\mathbf{U}_j^\rho\Big)\Big(\prod_{j=k+1}^{i-1}\mathbf{U}_j^\rho\Big)^\top\mathbf{U}_k^{\rho\top} = \lambda\sum_{i=k}^t \beta_i\Big(\prod_{j=k}^{i-1}\mathbf{U}_j^\rho\Big)\Big(\prod_{j=k}^{i-1}\mathbf{U}_j^\rho\Big)^\top,$$

where the coefficients of $\check{\mathbf{z}}_t$ and $\boldsymbol{\mu}_0$ in the mean can be further simplified as

$$\mathbf{U}_k^\rho \mathbf{U}_{k+1}^\rho \mathbf{U}_{k+2}^\rho \cdots \mathbf{U}_t^\rho$$
$$= \sqrt{\alpha_k}\Big(\boldsymbol{\Sigma}_{k-1} + \frac{1}{2}\rho\frac{\beta_k}{\alpha_k}\mathbf{I}\Big)\boldsymbol{\Sigma}_k^{-1}\cdot\Big(\prod_{i=k+1}^t \sqrt{\alpha_i}\Big)\Big(\boldsymbol{\Sigma}_k + \frac{1}{2}\rho\frac{\beta_{k+1}}{\alpha_{k+1}}\mathbf{I}\Big)\prod_{i=k+1}^{t-1}\Big(\mathbf{I} + \frac{1}{2}\rho\frac{\beta_{i+1}}{\alpha_{i+1}}\boldsymbol{\Sigma}_i^{-1}\Big)\boldsymbol{\Sigma}_t^{-1}$$
$$= \Big(\prod_{i=k}^t \sqrt{\alpha_i}\Big)\Big(\boldsymbol{\Sigma}_{k-1} + \frac{1}{2}\rho\frac{\beta_k}{\alpha_k}\mathbf{I}\Big)\Big(\mathbf{I} + \frac{1}{2}\rho\frac{\beta_{k+1}}{\alpha_{k+1}}\boldsymbol{\Sigma}_k^{-1}\Big)\prod_{i=k+1}^{t-1}\Big(\mathbf{I} + \frac{1}{2}\rho\frac{\beta_{i+1}}{\alpha_{i+1}}\boldsymbol{\Sigma}_i^{-1}\Big)\boldsymbol{\Sigma}_t^{-1}$$
$$= \Big(\prod_{i=k}^t \sqrt{\alpha_i}\Big)\Big(\boldsymbol{\Sigma}_{k-1} + \frac{1}{2}\rho\frac{\beta_k}{\alpha_k}\mathbf{I}\Big)\prod_{i=k}^{t-1}\Big(\mathbf{I} + \frac{1}{2}\rho\frac{\beta_{i+1}}{\alpha_{i+1}}\boldsymbol{\Sigma}_i^{-1}\Big)\boldsymbol{\Sigma}_t^{-1},$$

and

$$\mathbf{U}_k^\rho\Big(\mathbf{U}_{k+1}^\rho \cdots (\mathbf{U}_{t-1}^\rho \mathbf{V}_t^\rho + \mathbf{V}_{t-1}^\rho) \cdots + \mathbf{V}_{k+1}^\rho\Big) + \mathbf{V}_k^\rho$$
$$=\frac{1}{2}(2-\rho)\sqrt{\bar{\alpha}_k}\Big(\sum_{i=k+1}^t \Big(\prod_{j=k}^{i-1}\alpha_j\Big)\beta_i\Big(\boldsymbol{\Sigma}_{k-1} + \frac{1}{2}\rho\frac{\beta_k}{\alpha_k}\mathbf{I}\Big)\Big(\prod_{j=k}^{i-2}\Big(\mathbf{I} + \frac{1}{2}\rho\frac{\beta_{j+1}}{\alpha_{j+1}}\boldsymbol{\Sigma}_j^{-1}\Big)\Big)\boldsymbol{\Sigma}_{i-1}^{-1}\boldsymbol{\Sigma}_i^{-1} + \beta_k\boldsymbol{\Sigma}_k^{-1}\Big).$$

At the last step, the conditional distribution of the reconstruction $p_{Z_0|Z_t=\check{\mathbf{z}}_t}(\mathbf{z}_0)$ is given by

$$p_{Z_0|Z_t=\check{\mathbf{z}}_t}(\mathbf{z}_0) = \mathcal{N}(\mathbf{A}_t^\rho \check{\mathbf{z}}_t + \mathbf{B}_t^\rho \boldsymbol{\mu}_0, \lambda\boldsymbol{\Lambda}_t^\rho), \tag{14}$$

where

$$\mathbf{A}_t^\rho := \mathbf{U}_1^\rho \mathbf{U}_2^\rho \cdots \mathbf{U}_t^\rho = \sqrt{\bar{\alpha}_t} \mathbf{\Sigma}_0 \prod_{i=0}^{t-1} \big(\mathbf{I} + \frac{1}{2}\rho \frac{\beta_{i+1}}{\alpha_{i+1}} \mathbf{\Sigma}_i^{-1}\big) \mathbf{\Sigma}_t^{-1},$$

$$\mathbf{B}_t^\rho := \mathbf{U}_1^\rho \Big(\mathbf{U}_2^\rho \cdots \big(\mathbf{U}_{t-1}^\rho \mathbf{V}_t^\rho + \mathbf{V}_{t-1}^\rho\big) \cdots + \mathbf{V}_2^\rho\Big) + \mathbf{V}_1^\rho$$

$$= \frac{1}{2}(2-\rho)\bigg(\sum_{i=2}^{t} \bar{\alpha}_{i-1}\beta_i \mathbf{\Sigma}_0 \prod_{j=0}^{i-2}\big(\mathbf{I} + \frac{1}{2}\rho\frac{\beta_{j+1}}{\alpha_{j+1}}\mathbf{\Sigma}_j^{-1}\big)\mathbf{\Sigma}_{i-1}^{-1}\mathbf{\Sigma}_i^{-1} + \beta_1 \mathbf{\Sigma}_1^{-1}\bigg),$$

$$\mathbf{\Lambda}_t^\rho := \sum_{i=1}^{t} \beta_i \Big(\prod_{j=1}^{i-1}\mathbf{U}_j^\rho\Big)\Big(\prod_{j=1}^{i-1}\mathbf{U}_j^\rho\Big)^\top.$$

For our score-scaled ODE (which corresponds to $\lambda \to 0$), the variance will be zero, regardless of the value of $\rho$. Thus, the reconstruction $\hat{X}^\rho(\check{\mathbf{z}}_t)$ is given by the mean in Eq. (14), i.e.,

$$\hat{X}^\rho(\check{\mathbf{z}}_t) = \mathbf{A}_t^\rho \check{\mathbf{z}}_t + \mathbf{B}_t^\rho \boldsymbol{\mu}_0. \tag{15}$$

**At the point of** $\rho = 0$, $\mathbf{B}_t^\rho$ has a simplified expression,

$$\mathbf{B}_t^\rho = \sum_{i=2}^{t} \bar{\alpha}_{i-1}\beta_i \mathbf{\Sigma}_0 \mathbf{\Sigma}_{i-1}^{-1}\mathbf{\Sigma}_i^{-1} + \beta_1\mathbf{\Sigma}_1^{-1} = (1-\bar{\alpha}_t)\mathbf{\Sigma}_t^{-1},$$

and the mean of $p_{Z_0|Z_t = \mathbf{z}_t}(\mathbf{z}_0)$ is

$$\hat{\boldsymbol{\mu}}_0(\check{\mathbf{z}}_t) = \sqrt{\bar{\alpha}_t}\mathbf{\Sigma}_0\mathbf{\Sigma}_t^{-1}\check{\mathbf{z}}_t + (1-\bar{\alpha}_t)\mathbf{\Sigma}_t^{-1}\boldsymbol{\mu}_0.$$

By Tweedie's formula (Robbins, 1956) and the relationship $Z_t = \sqrt{\bar{\alpha}_t}X + \sqrt{1-\bar{\alpha}_t}N$, we have

$$\mathbb{E}[X|Z_t = \check{\mathbf{z}}_t] = \frac{1}{\sqrt{\bar{\alpha}_t}}\big(\check{\mathbf{z}}_t + (1-\bar{\alpha}_t)\nabla_{\mathbf{z}_t}\log p_{Z_t}(\check{\mathbf{z}}_t)\big)$$

$$= \frac{1}{\sqrt{\bar{\alpha}_t}}\big(\check{\mathbf{z}}_t + (1-\bar{\alpha}_t)\mathbf{\Sigma}_t^{-1}(\boldsymbol{\mu}_t - \check{\mathbf{z}}_t)\big)$$

$$= \frac{1}{\sqrt{\bar{\alpha}_t}}\big(\mathbf{\Sigma}_t - (1-\bar{\alpha}_t)\mathbf{I}\big)\mathbf{\Sigma}_t^{-1}\check{\mathbf{z}}_t + \frac{1}{\sqrt{\bar{\alpha}_t}}(1-\bar{\alpha}_t)\mathbf{\Sigma}_t^{-1}\boldsymbol{\mu}_t$$

$$= \sqrt{\bar{\alpha}_t}\mathbf{\Sigma}_0\mathbf{\Sigma}_t^{-1}\check{\mathbf{z}}_t + (1-\bar{\alpha}_t)\mathbf{\Sigma}_t^{-1}\boldsymbol{\mu}_0 = \hat{\boldsymbol{\mu}}_0(\check{\mathbf{z}}_t),$$

which implies that the reverse sampling of the proposed score-scaled PF-ODE can reach the MMSE $\mathbb{E}[X|Z_t = \check{\mathbf{z}}_t]$.

## C.2. An Alternative Expression of $\mathbf{A}_t^\rho$ and $\mathbf{B}_t^\rho$

Since $\mathbf{\Sigma}_k$ is a covariance matrix, it is symmetric and positive semi-definite. Further suppose $\mathbf{\Sigma}_k$ is invertible. Then, for any $\mathbf{\Sigma}_i$ and $\mathbf{\Sigma}_j$, we have

$$\mathbf{\Sigma}_i\mathbf{\Sigma}_j = (\bar{\alpha}_i\mathbf{\Sigma}_0 + (1-\bar{\alpha}_i)\mathbf{I})(\bar{\alpha}_j\mathbf{\Sigma}_0 + (1-\bar{\alpha}_j)\mathbf{I})$$

$$= \bar{\alpha}_i\bar{\alpha}_j\mathbf{\Sigma}_0^2 + \big(\bar{\alpha}_i + \bar{\alpha}_j - 2\bar{\alpha}_i\bar{\alpha}_j\big)\mathbf{\Sigma}_0 + (1-\bar{\alpha}_i)(1-\bar{\alpha}_j)\mathbf{I},$$

which is symmetric. Thus, $\mathbf{\Sigma}_i\mathbf{\Sigma}_j = (\mathbf{\Sigma}_i\mathbf{\Sigma}_j)^\top = \mathbf{\Sigma}_j\mathbf{\Sigma}_i$, which implies that $\mathbf{\Sigma}_i$ and $\mathbf{\Sigma}_j$ commute with each other.

**Lemma C.2.** *(Friedberg et al., 2003, Section 5.2) Two diagonalizable matrices* $\mathbf{A}$ *and* $\mathbf{B}$ *commute if and only if they can be simultaneously diagonalized, i.e., there exists a nonsingular matrix* $\mathbf{P}$ *such that both* $\mathbf{PAP}^{-1}$ *and* $\mathbf{PBP}^{-1}$ *are diagonal.*

First, we show an approximation which will be used frequently in the following proof. For $i \in \{0, \ldots, T-1\}$, we have

$$\left(\alpha_{i+1}\boldsymbol{\Sigma}_i + \frac{\rho}{2}\beta_{i+1}\mathbf{I}\right)^2$$

$$= \left(\alpha_{i+1}(\bar{\alpha}_i\boldsymbol{\Sigma}_0 + (1-\bar{\alpha}_i)\mathbf{I}) + \frac{\rho}{2}\beta_{i+1}\mathbf{I}\right)^2$$

$$= \bar{\alpha}_{i+1}\bar{\alpha}_{i+1}\boldsymbol{\Sigma}_0\boldsymbol{\Sigma}_0 + 2\bar{\alpha}_{i+1}(\alpha_{i+1} - \bar{\alpha}_{i+1} + \frac{\rho}{2}\beta_{i+1})\boldsymbol{\Sigma}_0 + (\alpha_{i+1} - \bar{\alpha}_{i+1} + \frac{\rho}{2}\beta_{i+1})^2\mathbf{I}$$

$$= \bar{\alpha}_{i+1}\bar{\alpha}_{i+1}\boldsymbol{\Sigma}_0\boldsymbol{\Sigma}_0 + \bar{\alpha}_{i+1}(2\alpha_{i+1} - 2\bar{\alpha}_{i+1} + \rho\beta_{i+1})\boldsymbol{\Sigma}_0 + (\alpha_{i+1} - \bar{\alpha}_{i+1} + \rho\beta_{i+1})(\alpha_{i+1} - \bar{\alpha}_{i+1})\mathbf{I} + \frac{\rho}{4}\beta_{i+1}^2\mathbf{I}$$

$$= \alpha_{i+1}\left(\bar{\alpha}_i\bar{\alpha}_{i+1}\boldsymbol{\Sigma}_0\boldsymbol{\Sigma}_0 + ((2-\rho)\bar{\alpha}_{i+1} + \rho\bar{\alpha}_i - 2\bar{\alpha}_{i+1}\bar{\alpha}_i)\boldsymbol{\Sigma}_0 + ((1-\rho)\alpha_{i+1} - \bar{\alpha}_{i+1} + \rho)(1-\bar{\alpha}_i)\mathbf{I}\right) + \mathcal{O}(\beta_{i+1}^2)$$

$$\overset{(a)}{=} \alpha_{i+1}\left(\bar{\alpha}_i\boldsymbol{\Sigma}_0 + (1-\bar{\alpha}_i)\mathbf{I}\right)\left(\bar{\alpha}_{i+1}\boldsymbol{\Sigma}_0 + (\rho + (1-\rho)\alpha_{i+1} - \bar{\alpha}_{i+1})\mathbf{I}\right)$$

$$= \alpha_{i+1}\boldsymbol{\Sigma}_i(\rho\boldsymbol{\Sigma}_{i+1} + (1-\rho)\alpha_{i+1}\boldsymbol{\Sigma}_i),$$

where $(a)$ is approximated by neglecting the $\mathcal{O}(\beta_{i+1}^2)$ term as $T \to \infty$. Furthermore, since $\boldsymbol{\Sigma}_i$ and $(\rho\boldsymbol{\Sigma}_{i+1} + (1-\rho)\alpha_{i+1}\boldsymbol{\Sigma}_i)$ commute, together with Lemma C.2, we have

$$\alpha_{i+1}\boldsymbol{\Sigma}_i + \frac{\rho}{2}\beta_{i+1}\mathbf{I} = \sqrt{\alpha_{i+1}}\boldsymbol{\Sigma}_i^{\frac{1}{2}}(\rho\boldsymbol{\Sigma}_{i+1} + (1-\rho)\alpha_{i+1}\boldsymbol{\Sigma}_i)^{\frac{1}{2}}, \text{ as } T \to \infty. \tag{16}$$

In particular, when $\rho = 1$, $\alpha_{i+1}\boldsymbol{\Sigma}_i + \frac{1}{2}\beta_{i+1}\mathbf{I} = \sqrt{\alpha_{i+1}}\boldsymbol{\Sigma}_i^{\frac{1}{2}}\boldsymbol{\Sigma}_{i+1}^{\frac{1}{2}}$ as $T \to \infty$.

Then, we can derive the alternative expression of $\mathbf{A}_t^\rho$ and $\mathbf{B}_t^\rho$. For $t \in \{1, \ldots, T\}$, we have

$$\mathbf{A}_t^\rho = \sqrt{\bar{\alpha}_t}\boldsymbol{\Sigma}_0 \prod_{i=0}^{t-1}\left(\mathbf{I} + \frac{1}{2}\rho\frac{\beta_{i+1}}{\alpha_{i+1}}\boldsymbol{\Sigma}_i^{-1}\right)\boldsymbol{\Sigma}_t^{-1}$$

$$= \sqrt{\bar{\alpha}_t}\boldsymbol{\Sigma}_0 \prod_{i=0}^{t-1}\left(\frac{1}{\alpha_{i+1}}(\alpha_{i+1}\boldsymbol{\Sigma}_i + \frac{\rho}{2}\beta_{i+1}\mathbf{I})\boldsymbol{\Sigma}_i^{-1}\right)\boldsymbol{\Sigma}_t^{-1}$$

$$= \sqrt{\bar{\alpha}_t}\boldsymbol{\Sigma}_0 \prod_{i=0}^{t-1}\left(\frac{1}{\sqrt{\alpha_{i+1}}}\boldsymbol{\Sigma}_i^{-\frac{1}{2}}(\rho\boldsymbol{\Sigma}_{i+1} + (1-\rho)\alpha_{i+1}\boldsymbol{\Sigma}_i)^{\frac{1}{2}}\right)\boldsymbol{\Sigma}_t^{-1} \quad \text{[By Eq. (16)]}$$

$$= \boldsymbol{\Sigma}_0 \prod_{i=0}^{t-1}\left(\boldsymbol{\Sigma}_i^{-\frac{1}{2}}(\rho\mathbf{I} + (1-\rho)\alpha_{i+1}\boldsymbol{\Sigma}_i\boldsymbol{\Sigma}_{i+1}^{-1})^{\frac{1}{2}}\boldsymbol{\Sigma}_{i+1}^{\frac{1}{2}}\right)\boldsymbol{\Sigma}_t^{-1}$$

$$= \boldsymbol{\Sigma}_0^{\frac{1}{2}} \prod_{i=0}^{t-1}\left(\rho\mathbf{I} + (1-\rho)\alpha_{i+1}\boldsymbol{\Sigma}_i\boldsymbol{\Sigma}_{i+1}^{-1}\right)^{\frac{1}{2}}\boldsymbol{\Sigma}_t^{-\frac{1}{2}}, \tag{17}$$

as $T \to \infty$, and

$$\mathbf{B}_t^\rho = \frac{1}{2}(2-\rho)\left(\sum_{i=2}^{t}\bar{\alpha}_{i-1}\beta_i\boldsymbol{\Sigma}_0 \prod_{j=0}^{i-2}\left(\mathbf{I} + \frac{1}{2}\rho\frac{\beta_{j+1}}{\alpha_{j+1}}\boldsymbol{\Sigma}_j^{-1}\right)\boldsymbol{\Sigma}_{i-1}^{-1}\boldsymbol{\Sigma}_i^{-1} + \beta_1\boldsymbol{\Sigma}_1^{-1}\right)$$

$$= \frac{1}{2}(2-\rho)\left(\sum_{i=2}^{t}\bar{\alpha}_{i-1}\beta_i\frac{1}{\sqrt{\bar{\alpha}_{i-1}}}\boldsymbol{\Sigma}_0^{\frac{1}{2}} \prod_{j=0}^{i-2}\left(\rho\mathbf{I} + (1-\rho)\alpha_{j+1}\boldsymbol{\Sigma}_j\boldsymbol{\Sigma}_{j+1}^{-1}\right)^{\frac{1}{2}}\boldsymbol{\Sigma}_{i-1}^{-\frac{1}{2}}\boldsymbol{\Sigma}_i^{-1} + \beta_1\boldsymbol{\Sigma}_1^{-1}\right) \quad \text{[By Eq. (17)]}$$

$$\overset{(a)}{=} (2-\rho)\left(\mathbf{I} - \sqrt{\bar{\alpha}_t}\boldsymbol{\Sigma}_0^{\frac{1}{2}}\boldsymbol{\Sigma}_t^{-\frac{1}{2}} - \sum_{i=1}^{t-1}\frac{1}{2}\sqrt{\bar{\alpha}_i}\beta_{i+1}\boldsymbol{\Sigma}_0^{\frac{1}{2}}\left(\mathbf{I} - \prod_{j=0}^{i-1}\left(\rho\mathbf{I} + (1-\rho)\alpha_{j+1}\boldsymbol{\Sigma}_j\boldsymbol{\Sigma}_{j+1}^{-1}\right)^{\frac{1}{2}}\right)\boldsymbol{\Sigma}_{i+1}^{-1}\boldsymbol{\Sigma}_i^{-\frac{1}{2}}\right), \tag{18}$$

as $T \to \infty$, where (a) can be proved by induction. Note that the alternative expressions Eq. (17) and Eq. (18) will be used in the following sections to derive the marginal distributions of $Z_0$ and prove the achievability results of DP tradeoffs.

Now we prove (a) in Eq. (18) by induction. For $t = 1$, we have

$$\mathbf{B}_1^\rho = \frac{1}{2}(2-\rho)\beta_1\boldsymbol{\Sigma}_1^{-1} = (2-\rho)\left(\alpha_1\boldsymbol{\Sigma}_0 - \alpha\boldsymbol{\Sigma}_0 + (1-\alpha_1)\mathbf{I} - \frac{1}{2}\beta_1\mathbf{I}\right)\boldsymbol{\Sigma}_1^{-1}$$

$$= (2-\rho)\left(\boldsymbol{\Sigma}_1 - \sqrt{\alpha_1}\boldsymbol{\Sigma}_0^{\frac{1}{2}}\boldsymbol{\Sigma}_1^{\frac{1}{2}}\right)\boldsymbol{\Sigma}_1^{-1} = (2-\rho)\left(\mathbf{I} - \sqrt{\alpha_1}\boldsymbol{\Sigma}_0^{\frac{1}{2}}\boldsymbol{\Sigma}_1^{-\frac{1}{2}}\right),$$

as $T \to \infty$. Suppose we have

$$\mathbf{B}_{t-1}^{\rho} = (2-\rho)\Big(\mathbf{I} - \sqrt{\bar{\alpha}_{t-1}}\boldsymbol{\Sigma}_0^{\frac{1}{2}}\boldsymbol{\Sigma}_{t-1}^{-\frac{1}{2}} - \underbrace{\sum_{i=1}^{t-2}\frac{1}{2}\sqrt{\bar{\alpha}_i}\beta_{i+1}\boldsymbol{\Sigma}_0^{\frac{1}{2}}\big(\mathbf{I} - \prod_{j=0}^{i-1}(\rho\mathbf{I} + (1-\rho)\alpha_{j+1}\boldsymbol{\Sigma}_j\boldsymbol{\Sigma}_{j+1}^{-1})^{\frac{1}{2}}\big)\boldsymbol{\Sigma}_{i+1}^{-1}\boldsymbol{\Sigma}_i^{-\frac{1}{2}}}_{:=\mathbf{C}_i^{\rho}}\Big).$$

Then, by induction, we have

$$\begin{aligned}
\mathbf{B}_t^{\rho} =& \mathbf{B}_{t-1}^{\rho} + \frac{1}{2}(2-\rho)\sqrt{\bar{\alpha}_{t-1}}\beta_t\boldsymbol{\Sigma}_0^{\frac{1}{2}}\prod_{j=0}^{t-2}\big(\rho\mathbf{I} + (1-\rho)\alpha_{j+1}\boldsymbol{\Sigma}_j\boldsymbol{\Sigma}_{j+1}^{-1}\big)^{\frac{1}{2}}\boldsymbol{\Sigma}_t^{-1}\boldsymbol{\Sigma}_{t-1}^{-\frac{1}{2}} \\
=& (2-\rho)\Big(\mathbf{I} - \sqrt{\bar{\alpha}_{t-1}}\boldsymbol{\Sigma}_0^{\frac{1}{2}}\boldsymbol{\Sigma}_{t-1}^{-\frac{1}{2}} - \sum_{i=1}^{t-2}\mathbf{C}_i^{\rho} + \frac{1}{2}\sqrt{\bar{\alpha}_{t-1}}\beta_t\boldsymbol{\Sigma}_0^{\frac{1}{2}}\prod_{j=0}^{t-2}\big(\rho\mathbf{I} + (1-\rho)\alpha_{j+1}\boldsymbol{\Sigma}_j\boldsymbol{\Sigma}_{j+1}^{-1}\big)^{\frac{1}{2}}\boldsymbol{\Sigma}_t^{-1}\boldsymbol{\Sigma}_{t-1}^{-\frac{1}{2}}\Big) \\
=& (2-\rho)\Big(\mathbf{I} - \sqrt{\bar{\alpha}_{t-1}}\boldsymbol{\Sigma}_0^{\frac{1}{2}}\big(\boldsymbol{\Sigma}_t - \beta_t\frac{1}{2}\prod_{j=0}^{t-2}\big(\rho\mathbf{I} + (1-\rho)\alpha_{j+1}\boldsymbol{\Sigma}_j\boldsymbol{\Sigma}_{j+1}^{-1}\big)^{\frac{1}{2}}\big)\boldsymbol{\Sigma}_t^{-1}\boldsymbol{\Sigma}_{t-1}^{-\frac{1}{2}} - \sum_{i=1}^{t-2}\mathbf{C}_i^{\rho}\Big) \\
\overset{(b)}{=}& (2-\rho)\Big(\mathbf{I} - \sqrt{\bar{\alpha}_{t-1}}\boldsymbol{\Sigma}_0^{\frac{1}{2}}\big(\sqrt{\alpha_t}\boldsymbol{\Sigma}_{t-1}^{\frac{1}{2}}\boldsymbol{\Sigma}_t^{\frac{1}{2}} + \frac{1}{2}\beta_t\big(\mathbf{I} - \prod_{j=0}^{t-2}\big(\rho\mathbf{I} + (1-\rho)\alpha_{j+1}\boldsymbol{\Sigma}_j\boldsymbol{\Sigma}_{j+1}^{-1}\big)^{\frac{1}{2}}\big)\big)\boldsymbol{\Sigma}_t^{-1}\boldsymbol{\Sigma}_{t-1}^{-\frac{1}{2}} - \sum_{i=1}^{t-2}\mathbf{C}_i^{\rho}\Big) \\
=& (2-\rho)\Big(\mathbf{I} - \sqrt{\bar{\alpha}_t}\boldsymbol{\Sigma}_0^{\frac{1}{2}}\boldsymbol{\Sigma}_t^{-\frac{1}{2}} - \sum_{i=1}^{t-1}\frac{1}{2}\sqrt{\bar{\alpha}_i}\beta_{i+1}\boldsymbol{\Sigma}_0^{\frac{1}{2}}\big(\mathbf{I} - \prod_{j=0}^{i-1}(\rho\mathbf{I} + (1-\rho)\alpha_{j+1}\boldsymbol{\Sigma}_j\boldsymbol{\Sigma}_{j+1}^{-1})^{\frac{1}{2}}\big)\boldsymbol{\Sigma}_{i+1}^{-1}\boldsymbol{\Sigma}_i^{-\frac{1}{2}}\Big),
\end{aligned}$$

as $T \to \infty$, where (b) follows from $\boldsymbol{\Sigma}_t = \alpha_t\boldsymbol{\Sigma}_{t-1} + \beta_t\mathbf{I}$ and Eq. (16).

**At the point of $\rho = 1$**, $\mathbf{A}_t^{\rho}$ and $\mathbf{B}_t^{\rho}$ have simplified expressions, and the solution becomes

$$Z_0 = \boldsymbol{\Sigma}_0^{\frac{1}{2}}\boldsymbol{\Sigma}_t^{-\frac{1}{2}}Z_t + (\mathbf{I} - \sqrt{\bar{\alpha}_t}\boldsymbol{\Sigma}_0^{\frac{1}{2}}\boldsymbol{\Sigma}_t^{-\frac{1}{2}})\boldsymbol{\mu}_0,$$

as $T \to \infty$. This matches the optimal transport solution $\hat{X}_0$ in Theorem 2 of (Freirich et al., 2021), where $\hat{X}_0 = \boldsymbol{\Sigma}_X^{\frac{1}{2}}\boldsymbol{\Sigma}_{X^\star}^{-\frac{1}{2}}X^\star = \boldsymbol{\Sigma}_0^{\frac{1}{2}}\boldsymbol{\Sigma}_t^{-\frac{1}{2}}Z_t$ assuming zero mean Gaussian source. Here $X^\star$ is the MMSE solution with covariance matrix $\boldsymbol{\Sigma}_{X^\star} = \bar{\alpha}_t\boldsymbol{\Sigma}_0^2\boldsymbol{\Sigma}_t^{-1}$.

### C.3. Marginal Distributions of $p_{Z_0}(\cdot)$

In Section C.1, we have derived the conditional distribution of $p_{Z_0|Z_t=\check{\mathbf{z}}_t}(\mathbf{z}_0)$ in Eq. (14). With Lemma C.1 and $p_{Z_t}(\mathbf{z}_t) = \mathcal{N}(\sqrt{\bar{\alpha}_t}\boldsymbol{\mu}_0, \bar{\alpha}_t\boldsymbol{\Sigma}_0 + (1-\bar{\alpha}_t)\mathbf{I})$, we can obtain that $Z_0 \sim \mathcal{N}(\hat{\boldsymbol{\mu}}_0, \hat{\boldsymbol{\Sigma}}_0)$, where

$$\hat{\boldsymbol{\mu}}_0 := \mathbf{A}_t^{\rho}\sqrt{\bar{\alpha}_t}\boldsymbol{\mu}_0 + \mathbf{B}_t^{\rho}\boldsymbol{\mu}_0 = (\mathbf{A}_t^{\rho}\sqrt{\bar{\alpha}_t} + \mathbf{B}_t^{\rho})\boldsymbol{\mu}_0, \tag{19}$$

and the variance is

$$\hat{\boldsymbol{\Sigma}}_0 := \lambda\boldsymbol{\Lambda}_t^{\rho} + \mathbf{A}_t^{\rho}\big(\bar{\alpha}_t\boldsymbol{\Sigma}_0 + (1-\bar{\alpha}_t)\mathbf{I}\big)\mathbf{A}_t^{\rho\top}. \tag{20}$$

First, we show that the mean of $p_{Z_0}(\mathbf{z}_0)$ is $\boldsymbol{\mu}_0$ regardless of the value of $\rho$. For any $\rho \in [0,1]$ and $t \in \{1, \ldots, T\}$, we have

$$
\mathbf{A}_t^\rho \sqrt{\bar{\alpha}_t} + \mathbf{B}_t^\rho
$$

$$
= \bar{\alpha}_t \Big( \boldsymbol{\Sigma}_0 + \frac{1}{2}\rho\frac{\beta_1}{\alpha_1}\mathbf{I} \Big) \Big( \prod_{i=1}^{t-1} \big( \mathbf{I} + \frac{1}{2}\rho\frac{\beta_{i+1}}{\alpha_{i+1}}\boldsymbol{\Sigma}_i^{-1} \big) \Big) \boldsymbol{\Sigma}_t^{-1}
$$

$$
\quad + \frac{1}{2}(2-\rho) \Big( \sum_{i=2}^{t} \bar{\alpha}_{i-1}\beta_i \Big( \boldsymbol{\Sigma}_0 + \frac{1}{2}\rho\frac{\beta_1}{\alpha_1}\mathbf{I} \Big) \Big( \prod_{j=1}^{i-2} \big( \mathbf{I} + \frac{1}{2}\rho\frac{\beta_{j+1}}{\alpha_{j+1}}\boldsymbol{\Sigma}_j^{-1} \big) \Big) \boldsymbol{\Sigma}_{i-1}^{-1}\boldsymbol{\Sigma}_i^{-1} + \beta_1\boldsymbol{\Sigma}_1^{-1} \Big)
$$

$$
= \bar{\alpha}_t \Big( \boldsymbol{\Sigma}_0 + \frac{1}{2}\rho\frac{\beta_1}{\alpha_1}\mathbf{I} \Big) \big( \mathbf{I} + \frac{1}{2}\rho\frac{\beta_2}{\alpha_2}\boldsymbol{\Sigma}_1^{-1} \big) \cdots \big( \mathbf{I} + \frac{1}{2}\rho\frac{\beta_t}{\alpha_t}\boldsymbol{\Sigma}_{t-1}^{-1} \big) \boldsymbol{\Sigma}_t^{-1} \qquad \text{①}
$$

$$
\quad + \frac{1}{2}(2-\rho)\bar{\alpha}_{t-1}\beta_t \Big( \boldsymbol{\Sigma}_0 + \frac{1}{2}\rho\frac{\beta_1}{\alpha_1}\mathbf{I} \Big) \big( \mathbf{I} + \frac{1}{2}\rho\frac{\beta_2}{\alpha_2}\boldsymbol{\Sigma}_1^{-1} \big) \cdots \big( \mathbf{I} + \frac{1}{2}\rho\frac{\beta_{t-1}}{\alpha_{t-1}}\boldsymbol{\Sigma}_{t-2}^{-1} \big) \boldsymbol{\Sigma}_{t-1}^{-1}\boldsymbol{\Sigma}_t^{-1} \qquad \text{②}
$$

$$
\quad + \frac{1}{2}(2-\rho)\bar{\alpha}_{t-2}\beta_{t-1} \Big( \boldsymbol{\Sigma}_0 + \frac{1}{2}\rho\frac{\beta_1}{\alpha_1}\mathbf{I} \Big) \big( \mathbf{I} + \frac{1}{2}\rho\frac{\beta_2}{\alpha_2}\boldsymbol{\Sigma}_1^{-1} \big) \cdots \big( \mathbf{I} + \frac{1}{2}\rho\frac{\beta_{t-2}}{\alpha_{t-2}}\boldsymbol{\Sigma}_{t-3}^{-1} \big) \boldsymbol{\Sigma}_{t-2}^{-1}\boldsymbol{\Sigma}_{t-1}^{-1} \qquad \text{③}
$$

$$
\quad + \cdots
$$

$$
\quad + \frac{1}{2}(2-\rho)\bar{\alpha}_1\beta_2 \Big( \boldsymbol{\Sigma}_0 + \frac{1}{2}\rho\frac{\beta_1}{\alpha_1}\mathbf{I} \Big) \boldsymbol{\Sigma}_1^{-1}\boldsymbol{\Sigma}_2^{-1} \qquad \text{ⓣ}
$$

$$
\quad + \frac{1}{2}(2-\rho)\beta_1\boldsymbol{\Sigma}_1^{-1}.
$$

By first considering ① + ②, we have

$$
\text{①} + \text{②} = \bar{\alpha}_{t-1} \Big( \boldsymbol{\Sigma}_0 + \frac{1}{2}\rho\frac{\beta_1}{\alpha_1}\mathbf{I} \Big) \big( \mathbf{I} + \frac{1}{2}\rho\frac{\beta_2}{\alpha_2}\boldsymbol{\Sigma}_1^{-1} \big) \cdots \big( \mathbf{I} + \frac{1}{2}\rho\frac{\beta_{t-1}}{\alpha_{t-1}}\boldsymbol{\Sigma}_{t-2}^{-1} \big) \Big[ \alpha_t \big( \mathbf{I} + \frac{1}{2}\rho\frac{\beta_t}{\alpha_t}\boldsymbol{\Sigma}_{t-1}^{-1} \big) + \frac{1}{2}(2-\rho)\beta_t\boldsymbol{\Sigma}_{t-1}^{-1} \Big] \boldsymbol{\Sigma}_t^{-1}
$$

$$
= \bar{\alpha}_{t-1} \Big( \boldsymbol{\Sigma}_0 + \frac{1}{2}\rho\frac{\beta_1}{\alpha_1}\mathbf{I} \Big) \big( \mathbf{I} + \frac{1}{2}\rho\frac{\beta_2}{\alpha_2}\boldsymbol{\Sigma}_1^{-1} \big) \cdots \big( \mathbf{I} + \frac{1}{2}\rho\frac{\beta_{t-1}}{\alpha_{t-1}}\boldsymbol{\Sigma}_{t-2}^{-1} \big) \boldsymbol{\Sigma}_{t-1}^{-1} \Big[ \alpha_t\boldsymbol{\Sigma}_{t-1} + \frac{1}{2}\rho\beta_t\mathbf{I} + \frac{1}{2}(2-\rho)\beta_t\mathbf{I} \Big] \boldsymbol{\Sigma}_t^{-1}
$$

$$
= \bar{\alpha}_{t-1} \Big( \boldsymbol{\Sigma}_0 + \frac{1}{2}\rho\frac{\beta_1}{\alpha_1}\mathbf{I} \Big) \big( \mathbf{I} + \frac{1}{2}\rho\frac{\beta_2}{\alpha_2}\boldsymbol{\Sigma}_1^{-1} \big) \cdots \big( \mathbf{I} + \frac{1}{2}\rho\frac{\beta_{t-1}}{\alpha_{t-1}}\boldsymbol{\Sigma}_{t-2}^{-1} \big) \boldsymbol{\Sigma}_{t-1}^{-1} \Big[ \bar{\alpha}_t\boldsymbol{\Sigma}_0 + (\alpha_t - \bar{\alpha}_t + \beta_t)\mathbf{I} \Big] \boldsymbol{\Sigma}_t^{-1}
$$

$$
= \bar{\alpha}_{t-1} \Big( \boldsymbol{\Sigma}_0 + \frac{1}{2}\rho\frac{\beta_1}{\alpha_1}\mathbf{I} \Big) \big( \mathbf{I} + \frac{1}{2}\rho\frac{\beta_2}{\alpha_2}\boldsymbol{\Sigma}_1^{-1} \big) \cdots \big( \mathbf{I} + \frac{1}{2}\rho\frac{\beta_{t-1}}{\alpha_{t-1}}\boldsymbol{\Sigma}_{t-2}^{-1} \big) \boldsymbol{\Sigma}_{t-1}^{-1}\boldsymbol{\Sigma}_t\boldsymbol{\Sigma}_t^{-1}
$$

$$
= \bar{\alpha}_{t-1} \Big( \boldsymbol{\Sigma}_0 + \frac{1}{2}\rho\frac{\beta_1}{\alpha_1}\mathbf{I} \Big) \big( \mathbf{I} + \frac{1}{2}\rho\frac{\beta_2}{\alpha_2}\boldsymbol{\Sigma}_1^{-1} \big) \cdots \big( \mathbf{I} + \frac{1}{2}\rho\frac{\beta_{t-1}}{\alpha_{t-1}}\boldsymbol{\Sigma}_{t-2}^{-1} \big) \boldsymbol{\Sigma}_{t-1}^{-1},
$$

which has a similar structure to ① with $t$ replaced by $t-1$. Thus, when proceeding to the next line, we have a similar induction structure

$$
\text{①} + \text{②} + \text{③}
$$

$$
= \bar{\alpha}_{t-2} \Big( \boldsymbol{\Sigma}_0 + \frac{1}{2}\rho\frac{\beta_1}{\alpha_1}\mathbf{I} \Big) \big( \mathbf{I} + \frac{1}{2}\rho\frac{\beta_2}{\alpha_2}\boldsymbol{\Sigma}_1^{-1} \big) \cdots \big( \mathbf{I} + \frac{1}{2}\rho\frac{\beta_{t-2}}{\alpha_{t-2}}\boldsymbol{\Sigma}_{t-3}^{-1} \big) \Big[ \alpha_{t-1} \big( \mathbf{I} + \frac{1}{2}\rho\frac{\beta_{t-1}}{\alpha_{t-1}}\boldsymbol{\Sigma}_{t-2}^{-1} \big) + \frac{1}{2}(2-\rho)\beta_{t-1}\boldsymbol{\Sigma}_{t-2}^{-1} \Big] \boldsymbol{\Sigma}_{t-1}^{-1}
$$

$$
= \bar{\alpha}_{t-2} \Big( \boldsymbol{\Sigma}_0 + \frac{1}{2}\rho\frac{\beta_1}{\alpha_1}\mathbf{I} \Big) \big( \mathbf{I} + \frac{1}{2}\rho\frac{\beta_2}{\alpha_2}\boldsymbol{\Sigma}_1^{-1} \big) \cdots \big( \mathbf{I} + \frac{1}{2}\rho\frac{\beta_{t-2}}{\alpha_{t-2}}\boldsymbol{\Sigma}_{t-3}^{-1} \big) \boldsymbol{\Sigma}_{t-2}^{-1} \Big[ \alpha_{t-1}\boldsymbol{\Sigma}_{t-2} + \frac{1}{2}\rho\beta_{t-1}\mathbf{I} + \frac{1}{2}(2-\rho)\beta_{t-1}\mathbf{I} \Big] \boldsymbol{\Sigma}_{t-1}^{-1}
$$

$$
= \bar{\alpha}_{t-2} \Big( \boldsymbol{\Sigma}_0 + \frac{1}{2}\rho\frac{\beta_1}{\alpha_1}\mathbf{I} \Big) \big( \mathbf{I} + \frac{1}{2}\rho\frac{\beta_2}{\alpha_2}\boldsymbol{\Sigma}_1^{-1} \big) \cdots \big( \mathbf{I} + \frac{1}{2}\rho\frac{\beta_{t-2}}{\alpha_{t-2}}\boldsymbol{\Sigma}_{t-3}^{-1} \big) \boldsymbol{\Sigma}_{t-2}^{-1}\boldsymbol{\Sigma}_{t-1}\boldsymbol{\Sigma}_{t-1}^{-1}
$$

$$
= \bar{\alpha}_{t-2} \Big( \boldsymbol{\Sigma}_0 + \frac{1}{2}\rho\frac{\beta_1}{\alpha_1}\mathbf{I} \Big) \big( \mathbf{I} + \frac{1}{2}\rho\frac{\beta_2}{\alpha_2}\boldsymbol{\Sigma}_1^{-1} \big) \cdots \big( \mathbf{I} + \frac{1}{2}\rho\frac{\beta_{t-2}}{\alpha_{t-2}}\boldsymbol{\Sigma}_{t-3}^{-1} \big) \boldsymbol{\Sigma}_{t-2}^{-1}.
$$

By induction, the summation over $\textcircled{1} \cdots \textcircled{t}$ is

$$
\begin{aligned}
&\textcircled{1} + \cdots + \textcircled{t}\\
=&\bar{\alpha}_2\Big(\boldsymbol{\Sigma}_0 + \frac{1}{2}\rho\frac{\beta_1}{\alpha_1}\mathbf{I}\Big)\big(\mathbf{I} + \frac{1}{2}\rho\frac{\beta_2}{\alpha_2}\boldsymbol{\Sigma}_1^{-1}\big)\boldsymbol{\Sigma}_2^{-1} + \frac{1}{2}(2-\rho)\bar{\alpha}_1\beta_2\Big(\boldsymbol{\Sigma}_0 + \frac{1}{2}\rho\frac{\beta_1}{\alpha_1}\mathbf{I}\Big)\boldsymbol{\Sigma}_1^{-1}\boldsymbol{\Sigma}_2^{-1}\\
=&\alpha_1\Big(\boldsymbol{\Sigma}_0 + \frac{1}{2}\rho\frac{\beta_1}{\alpha_1}\mathbf{I}\Big)\Big[\alpha_2\big(\mathbf{I} + \frac{1}{2}\rho\frac{\beta_2}{\alpha_2}\boldsymbol{\Sigma}_1^{-1}\big) + \frac{1}{2}(1-\rho)\beta_2\boldsymbol{\Sigma}_1^{-1}\Big]\boldsymbol{\Sigma}_2^{-1}\\
=&\alpha_1\Big(\boldsymbol{\Sigma}_0 + \frac{1}{2}\rho\frac{\beta_1}{\alpha_1}\mathbf{I}\Big)\boldsymbol{\Sigma}_1^{-1}\Big[\alpha_2\boldsymbol{\Sigma}_1 + \frac{1}{2}\rho\beta_2 + \frac{1}{2}(1-\rho)\beta_2\Big]\boldsymbol{\Sigma}_2^{-1}\\
=&\alpha_1\Big(\boldsymbol{\Sigma}_0 + \frac{1}{2}\rho\frac{\beta_1}{\alpha_1}\mathbf{I}\Big)\boldsymbol{\Sigma}_1^{-1}.
\end{aligned}
$$

Thus, we have

$$
\begin{aligned}
\mathbf{A}_t^\rho\sqrt{\bar{\alpha}_t} + \mathbf{B}_t^\rho &= \alpha_1\Big(\boldsymbol{\Sigma}_0 + \frac{1}{2}\rho\frac{\beta_1}{\alpha_1}\mathbf{I}\Big)\boldsymbol{\Sigma}_1^{-1} + \frac{1}{2}(2-\rho)\beta_1\boldsymbol{\Sigma}_1^{-1}\\
&= \big(\alpha_1\boldsymbol{\Sigma}_0 + \frac{1}{2}\rho\beta_1\mathbf{I} + \frac{1}{2}(2-\rho)\beta_1\mathbf{I}\big)\boldsymbol{\Sigma}_1^{-1}\\
&= \big(\alpha_1\boldsymbol{\Sigma}_0 + (1-\alpha_1)\mathbf{I}\big)\boldsymbol{\Sigma}_1^{-1}\\
&= \boldsymbol{\Sigma}_1\boldsymbol{\Sigma}_1^{-1} = \mathbf{I}.
\end{aligned}
$$

By plugging it into Eq. (19), we can obtain that the mean of $p_{Z_0}(\mathbf{z}_0)$ is $\hat{\boldsymbol{\mu}}_0 = (\mathbf{A}_t^\rho\sqrt{\bar{\alpha}_t} + \mathbf{B}_t^\rho)\boldsymbol{\mu}_0 = \boldsymbol{\mu}_0$.

Now, let's compute the variance of $p_{Z_0}(\mathbf{z}_0)$ with the help of the second expression of $\mathbf{A}_t^\rho$ in Eq. (17). By substituting Eq. (17) into Eq. (20), we have

$$
\begin{aligned}
\hat{\boldsymbol{\Sigma}}_0 =&\lambda\boldsymbol{\Lambda}_t^\rho + \mathbf{A}_t^\rho\big(\bar{\alpha}_t\boldsymbol{\Sigma}_0 + (1-\bar{\alpha}_t)\mathbf{I}\big)\mathbf{A}_t^{\rho\top}\\
=&\lambda\boldsymbol{\Lambda}_t^\rho + \boldsymbol{\Sigma}_0^{\frac{1}{2}}\prod_{i=0}^{t-1}\Big(\rho\mathbf{I} + (1-\rho)\alpha_{i+1}\boldsymbol{\Sigma}_i\boldsymbol{\Sigma}_{i+1}^{-1}\Big)^{\frac{1}{2}}\boldsymbol{\Sigma}_t^{-\frac{1}{2}}\boldsymbol{\Sigma}_t\Big(\boldsymbol{\Sigma}_0^{\frac{1}{2}}\prod_{i=0}^{t-1}\Big(\rho\mathbf{I} + (1-\rho)\alpha_{i+1}\boldsymbol{\Sigma}_i\boldsymbol{\Sigma}_{i+1}^{-1}\Big)^{\frac{1}{2}}\boldsymbol{\Sigma}_t^{-\frac{1}{2}}\Big)^\top\\
=&\lambda\boldsymbol{\Lambda}_t^\rho + \boldsymbol{\Sigma}_0\prod_{i=0}^{t-1}\Big(\rho\mathbf{I} + (1-\rho)\alpha_{i+1}\boldsymbol{\Sigma}_i\boldsymbol{\Sigma}_{i+1}^{-1}\Big).
\end{aligned}
\tag{21}
$$

In summary, the marginal distribution of the reconstruction given by our score-scaled PF-ODE is (by setting $\lambda \to 0$)

$$
p_{\hat{X}^\rho}(\hat{\mathbf{x}}^\rho) = \mathcal{N}\Big(\boldsymbol{\mu}_0,\ \boldsymbol{\Sigma}_0\prod_{i=0}^{t-1}\Big(\rho\mathbf{I} + (1-\rho)\alpha_{i+1}\boldsymbol{\Sigma}_i\boldsymbol{\Sigma}_{i+1}^{-1}\Big)\Big),\ \text{as } T \to \infty.
$$

**When $\rho = 0$**, the variance is

$$
\boldsymbol{\Sigma}_0\prod_{i=0}^{t-1}\Big(\alpha_{i+1}\boldsymbol{\Sigma}_i\boldsymbol{\Sigma}_{i+1}^{-1}\Big) = \bar{\alpha}_t\boldsymbol{\Sigma}_0\boldsymbol{\Sigma}_0\boldsymbol{\Sigma}_1^{-1}\cdots\boldsymbol{\Sigma}_{t-1}\boldsymbol{\Sigma}_t^{-1} = \bar{\alpha}_t\boldsymbol{\Sigma}_0^2\boldsymbol{\Sigma}_t^{-1},
$$

and the marginal distribution of the reconstruction is $p_{\hat{X}^\rho}(\hat{\mathbf{x}}^\rho) = \mathcal{N}(\boldsymbol{\mu}_0, \bar{\alpha}_t\boldsymbol{\Sigma}_0^2\boldsymbol{\Sigma}_t^{-1})$.

**When $\rho = 1$**, the variance is $\boldsymbol{\Sigma}_0$, and the marginal distribution is $p_{\hat{X}^\rho}(\hat{\mathbf{x}}^\rho) = \mathcal{N}(\boldsymbol{\mu}_0, \boldsymbol{\Sigma}_0)$, which match the marginal distribution of $p_X(\mathbf{x})$.

## D. Proof of Theorem 3.3

### D.1. Scalar Gaussian Case

Let's first prove the scalar Gaussian case, which will be part of the basis for the achievability proof of the vector Gaussian case.

### D.1.1. CONVERSE PROOF

Consider the source $X \sim \mathcal{N}(\mu_0, \sigma_0^2)$, and the decoder receives $Z_t = \sqrt{\bar{\alpha}_t} X + \sqrt{1 - \bar{\alpha}_t} N$, $N \sim \mathcal{N}(0, 1)$, which has distribution

$$p_{Z_t}(z_t) = \mathcal{N}(\underbrace{\sqrt{\bar{\alpha}_t} \mu_0}_{\mu_t}, \underbrace{\bar{\alpha}_t \sigma_0^2 + 1 - \bar{\alpha}_t}_{\sigma_t^2}).$$

The DP function is defined as:

$$D_t(P) = \min_{p_{\hat{X}|Z_t}(\hat{x}|z_t)} \mathbb{E}[(X - \hat{X})^2]$$

$$\text{s.t. } W_2^2(p_X, p_{\hat{X}}) \leq P.$$

To be proved in Lemma D.2 in the multivariate case, we can restrict the reconstruction to follow the form of $p_{\hat{X}|Z_t}(\hat{x}|z_t) = \mathcal{N}(az_t + b, c^2)$ for $a, b, c$ to be optimized. Then, the marginal distribution can be expressed as $p_{\hat{X}}(\hat{x}) = \mathcal{N}(a\sqrt{\bar{\alpha}_t}\mu_0 + b, c^2 + a^2\sigma_t^2)$, and the covariance of $X$ and $\hat{X}$ is $\mathrm{Cov}[X, \hat{X}] = a\sqrt{\bar{\alpha}_t}\sigma_0^2$. Define $\hat{\mu}_0 := a\sqrt{\bar{\alpha}_t}\mu_0 + b$.

Then, we can express the distortion and perception term as

$$\begin{aligned}
\mathbb{E}[(X - \hat{X})^2] &= \mathbb{E}[X^2] - 2\mathbb{E}[X\hat{X}] + \mathbb{E}[\hat{X}^2] \\
&= \sigma_0^2 + \mu_0^2 - 2(a\sqrt{\bar{\alpha}_t}\sigma_0^2 + \mu_0\hat{\mu}_0) + \hat{\mu}_0^2 + (c^2 + a^2\sigma_t^2) \\
&= (\mu_0 - \hat{\mu}_0)^2 + \sigma_0^2 + (c^2 + a^2\sigma_t^2) - 2a\sqrt{\bar{\alpha}_t}\sigma_0^2,
\end{aligned}$$

and

$$W_2^2(p_X, p_{\hat{X}}) = (\mu_0 - \hat{\mu}_0)^2 + (\sigma_0 - \sqrt{c^2 + a^2\sigma_t^2})^2.$$

Without loss of optimality, we can set $\hat{\mu}_0 = \mu_0$, which leads to $b = \mu_0 - a\sqrt{\bar{\alpha}_t}\mu_0$. Then, the optimization problem can be simplified as

$$D_t(P) = \min_{a,c} \sigma_0^2 + (c^2 + a^2\sigma_t^2) - 2a\sqrt{\bar{\alpha}_t}\sigma_0^2 \tag{22}$$

$$\text{s.t. } (\sigma_0 - \sqrt{c^2 + a^2\sigma_t^2})^2 \leq P.$$

Denoting $\hat{\sigma}_0^2 := c^2 + a^2\sigma_t^2$, the problem is equivalent to

$$D_t(P) = \min_{a,\hat{\sigma}_0} \sigma_0^2 + \hat{\sigma}_0^2 - 2a\sqrt{\bar{\alpha}_t}\sigma_0^2$$

$$\text{s.t. } (\sigma_0 - \hat{\sigma}_0)^2 \leq P$$

$$a^2\sigma_t^2 \leq \hat{\sigma}_0^2.$$

For any fixed $\hat{\sigma}_0$ (which leads to fixed value of $(\sigma_0 - \hat{\sigma}_0)^2$), we can find the optimal $a$ that minimizes the distortion level to $\frac{\hat{\sigma}_0}{\sigma_t}$, which corresponds to $c = 0$. Then, the problem is just to minimize a quadratic function $\hat{\sigma}_0^2 - 2\frac{\hat{\sigma}_0}{\sigma_t}\sqrt{\bar{\alpha}_t}\sigma_0^2 + \sigma_0^2 = (\hat{\sigma}_0 - \frac{\sqrt{\bar{\alpha}_t}\sigma_0^2}{\sigma_t})^2 + (1 - \bar{\alpha}_t)\frac{\sigma_0^2}{\sigma_t^2}$, with respect to $\sigma_0 - \sqrt{P} \leq \hat{\sigma}_0 \leq \sigma_0 + \sqrt{P}$.

$1^o$ When $\sqrt{P} < \sigma_0 - \frac{\sqrt{\bar{\alpha}_t}\sigma_0^2}{\sigma_t}$, the optimal $\hat{\sigma}_0$ is $\sigma_0 - \sqrt{P}$, and the optimal distortion is $\sigma_0^2 + (\sigma_0 - \sqrt{P})^2 - 2\frac{\sigma_0 - \sqrt{P}}{\sigma_t}\sqrt{\bar{\alpha}_t}\sigma_0^2 = \frac{(1-\bar{\alpha}_t)\sigma_0^2}{\sigma_t^2} + (\sigma_0 - \sqrt{P} - \frac{\sigma_0^2\sqrt{\bar{\alpha}_t}}{\sigma_t})^2$. The corresponding reconstruction distribution is

$$p_{\hat{X}|Z_t}(\hat{x}|z_t) = \mathcal{N}\left(\frac{\sigma_0 - \sqrt{P}}{\sigma_t}z_t + \left(1 - \frac{(\sigma_0 - \sqrt{P})\sqrt{\bar{\alpha}_t}}{\sigma_t}\right)\mu_0, \ 0\right).$$

$2^o$ When $\sqrt{P} \geq \sigma_0 - \frac{\sqrt{\bar{\alpha}_t}\sigma_0^2}{\sigma_t}$, the optimal $\hat{\sigma}_0$ is $\frac{\sqrt{\bar{\alpha}_t}\sigma_0^2}{\sigma_t}$, and the optimal distortion is $\frac{(1-\bar{\alpha}_t)\sigma_0^2}{\sigma_t^2}$. The corresponding reconstruction distribution is

$$p_{\hat{X}|Z_t}(\hat{x}|z_t) = \mathcal{N}\left(\frac{\sqrt{\bar{\alpha}_t}\sigma_0^2}{\sigma_t^2}z_t + \frac{1 - \bar{\alpha}_t}{\sigma_t^2}\mu_0, \ 0\right).$$

The resulting DP tradeoff is

$$D_t(P) = \begin{cases} \frac{(1-\bar{\alpha}_t)\sigma_0^2}{\sigma_t^2} + \left(\sigma_0 - \sqrt{P} - \frac{\sigma_0^2\sqrt{\bar{\alpha}_t}}{\sigma_t}\right)^2 & \sqrt{P} < \sigma_0 - \frac{\sqrt{\bar{\alpha}_t}\sigma_0^2}{\sigma_t}, \\ \frac{(1-\bar{\alpha}_t)\sigma_0^2}{\sigma_t^2} & \sqrt{P} \geq \sigma_0 - \frac{\sqrt{\bar{\alpha}_t}\sigma_0^2}{\sigma_t}. \end{cases}$$

*Remark* D.1. We can observe that $c^2$, which is the conditional variance of the reconstruction, is always zero. This coincides with the fact that the reconstruction is a deterministic function of $z_t$.

### D.1.2. ACHIEVABILITY PROOF

As shown in Eq. (15), the reconstruction $\hat{X}^\rho$ given by our proposed score-scaled PF-ODE is

$$\hat{X}^\rho = a_t^\rho Z_t + b_t^\rho \mu_0,$$

where

$$a_t^\rho = \sqrt{\bar{\alpha}_t}\frac{\sigma_0^2}{\sigma_t^2}\prod_{i=0}^{t-1}\left(1 + \frac{1}{2}\rho\frac{\beta_{i+1}}{\alpha_{i+1}}\frac{1}{\sigma_i^2}\right) = \frac{\sigma_0}{\sigma_t}\prod_{i=0}^{t-1}\left(\rho + (1-\rho)\alpha_{i+1}\frac{\sigma_i^2}{\sigma_{i+1}^2}\right)^{\frac{1}{2}},$$

$$b_t^\rho = \frac{1}{2}(2-\rho)\left(\sum_{i=2}^{t}\bar{\alpha}_{i-1}\beta_i\frac{\sigma_0^2}{\sigma_{i-1}^2\sigma_i^2}\prod_{j=0}^{i-2}\left(1 + \frac{1}{2}\rho\frac{\beta_{j+1}}{\alpha_{j+1}}\frac{1}{\sigma_j}\right) + \beta_1\frac{1}{\sigma_1^2}\right)$$

$$= (2-\rho)\left(1 - \sqrt{\bar{\alpha}_t}\frac{\sigma_0}{\sigma_t} - \sum_{i=1}^{t-1}\frac{1}{2}\sqrt{\bar{\alpha}_i}\beta_{i+1}\frac{\sigma_0}{\sigma_{i+1}^2\sigma_i}\left(\mathbf{I} - \prod_{j=0}^{i-1}\left(\rho\mathbf{I} + (1-\rho)\alpha_{j+1}\mathbf{\Sigma}_j\mathbf{\Sigma}_{j+1}^{-1}\right)^{\frac{1}{2}}\right)\right).$$

The covariance between $\hat{X}^\rho$ and $X$ is

$$\text{Cov}[X, \hat{X}^\rho] = \text{Cov}[X, a_t^\rho Z_t + b_t^\rho \mu_0] = \text{Cov}[X, a_t^\rho(\sqrt{\bar{\alpha}_t}X + \sqrt{1-\bar{\alpha}_t}N)] = a_t^\rho\sqrt{\bar{\alpha}_t}\sigma_0^2.$$

According to Eq. (21), the marginal distribution of the reconstruction is

$$p_{\hat{X}^\rho}(\hat{x}) = \mathcal{N}(\mu_0, \hat{\sigma}_0^2),$$

where

$$\hat{\sigma}_0^2 = \bar{\alpha}_t\frac{\sigma_0^4}{\sigma_t^2}\prod_{i=0}^{t-1}\left(1 + \frac{1}{2}\rho\frac{\beta_{i+1}}{\alpha_{i+1}}\frac{1}{\sigma_i^2}\right)^2 = \sigma_0^2\prod_{i=0}^{t-1}\left(\rho + (1-\rho)\alpha_{i+1}\frac{\sigma_i^2}{\sigma_{i+1}^2}\right).$$

We can now compute the achievable distortion and perception levels of our proposed score-scaled ODE as functions of $\rho$

$$D_t^\rho = \mathbb{E}[(X - \hat{X}^\rho)^2] = \sigma_0^2 + \sigma_0^2\prod_{i=0}^{t-1}\left(\rho + (1-\rho)\alpha_{i+1}\frac{\sigma_i^2}{\sigma_{i+1}^2}\right) - 2\frac{\sigma_0}{\sigma_t}\prod_{i=0}^{t-1}\left(\rho + (1-\rho)\alpha_{i+1}\frac{\sigma_i^2}{\sigma_{i+1}^2}\right)^{\frac{1}{2}}\sqrt{\bar{\alpha}_t}\sigma_0^2$$

$$= \sigma_0^2\left(\prod_{i=0}^{t-1}\left(\rho + (1-\rho)\alpha_{i+1}\frac{\sigma_i^2}{\sigma_{i+1}^2}\right)^{\frac{1}{2}} - \frac{\sqrt{\bar{\alpha}_t}\sigma_0}{\sigma_t}\right)^2 + \sigma_0^2 - \frac{\bar{\alpha}_t\sigma_0^4}{\sigma_t^2}, \tag{23}$$

$$P_t^\rho = W_2^2(p_X, p_{\hat{X}^\rho}) = \left(\sigma_0 - \sigma_0\prod_{i=0}^{t-1}\left(\rho + (1-\rho)\alpha_{i+1}\frac{\sigma_i^2}{\sigma_{i+1}^2}\right)^{\frac{1}{2}}\right)^2. \tag{24}$$

From Eq. (24) together with the fact that $\rho + (1-\rho)\alpha_{i+1}\frac{\sigma_i^2}{\sigma_{i+1}^2} \leq 1, \forall i$, we can obtain $\prod_{i=0}^{t-1}\left(\rho + (1-\rho)\alpha_{i+1}\frac{\sigma_i^2}{\sigma_{i+1}^2}\right)^{\frac{1}{2}} = \frac{\sigma_0 - \sqrt{P_t^\rho}}{\sigma_0}$. Since $0 \leq \rho \leq 1$, we have $\prod_{i=0}^{t-1}\left(\rho + (1-\rho)\alpha_{i+1}\frac{\sigma_i^2}{\sigma_{i+1}^2}\right)^{\frac{1}{2}} \in \left[\frac{\sqrt{\bar{\alpha}_t}\sigma_0}{\sigma_t}, 1\right]$, thus $P_t^\rho \in [0, (\sigma_0 - \frac{\sqrt{\bar{\alpha}_t}\sigma_0^2}{\sigma_t})^2]$. Plugging into Eq. (23) we can get the achievable DP tradeoff is

$$D_t^\rho(P_t^\rho) = \sigma_0^2\left(\frac{\sigma_0 - \sqrt{P_t^\rho}}{\sigma_0} - \frac{\sqrt{\bar{\alpha}_t}\sigma_0}{\sigma_t}\right)^2 + \frac{\sigma_0^2(\sigma_t^2 - \bar{\alpha}_t\sigma_0^2)}{\sigma_t^2}$$

$$= \left(\sigma_0 - \sqrt{P_t^\rho} - \frac{\sqrt{\bar{\alpha}_t}\sigma_0^2}{\sigma_t}\right)^2 + \frac{(1-\bar{\alpha}_t)\sigma_0^2}{\sigma_t^2}, \text{ for } 0 \leq P_t^\rho \leq \left(\sigma_0 - \frac{\sqrt{\bar{\alpha}_t}\sigma_0^2}{\sigma_t}\right)^2,$$

which matches the optimal DP tradeoff derived in the last section.

## D.2. Converse Proof for Multivariate Gaussian Case

For a $d$-dimensional source $X = (X_1, \ldots, X_d) \sim \mathcal{N}(\boldsymbol{\mu}_0, \boldsymbol{\Sigma}_0)$, consider the eigen-decomposition of the covariance matrix

$$\boldsymbol{\Sigma}_0 = \mathbf{Q}\boldsymbol{\Lambda}_0\mathbf{Q}^\top,$$

where $\mathbf{Q}$ is orthogonal and $\boldsymbol{\Lambda}_0$ is a diagonal matrix with positive eigenvalues $\boldsymbol{\Lambda}_0 = \mathrm{diag}(\lambda_1, \ldots, \lambda_d)$. We then define

$$Y = \mathbf{Q}^\top X,$$

which implies $Y = (Y_1, \ldots, Y_d) \sim \mathcal{N}(\mathbf{Q}^\top \boldsymbol{\mu}_0, \boldsymbol{\Lambda}_0)$. The components of $Y$ are mutually independent.

Given the received $Z_t = \sqrt{\bar{\alpha}_t}X + \sqrt{1 - \bar{\alpha}_t}N$, $N \sim \mathcal{N}(\mathbf{0}, \mathbf{I})$, the DP function is defined as

$$D_t(P) = \min_{p_{\hat{X}|Z_t}(\hat{\mathbf{x}}|\mathbf{z}_t)} \mathbb{E}[\|X - \hat{X}\|_2^2] \tag{25}$$

$$\text{s.t. } W_2^2(p_X, p_{\hat{X}}) \leq P.$$

**Lemma D.2.** *Without loss of optimality, for the optimization problem in Eq. (25), we can restrict the conditional distribution $p_{\hat{X}|Z_t}(\hat{\mathbf{x}}|\mathbf{z}_t)$ as the following form: Let $\tilde{Z}_t = \mathbf{Q}^\top Z_t = \sqrt{\bar{\alpha}_t}Y + \sqrt{1 - \bar{\alpha}_t}N_1$ and $\hat{Y} = \tilde{\mathbf{A}}\tilde{Z}_t + \tilde{\mathbf{b}} + \tilde{\mathbf{C}}N_2$, where $N_1, N_2 \overset{i.i.d.}{\sim} \mathcal{N}(\mathbf{0}, \mathbf{I})$, $\tilde{\mathbf{A}} = \mathrm{diag}(\tilde{a}_1, \ldots, \tilde{a}_d)$ and $\tilde{\mathbf{C}} = \mathrm{diag}(\tilde{c}_1, \ldots, \tilde{c}_d)$ are diagonal matrices with $\tilde{c}_\ell \geq 0$ for $1 \leq \ell \leq d$. Then $\hat{X} = \mathbf{Q}\hat{Y}$.*

*Proof.* For any $Y = \mathbf{Q}^\top X$ and $\hat{X} = \mathbf{Q}\hat{Y}$, we have

$$\mathbb{E}[\|X - \hat{X}\|_2^2] \overset{(a)}{=} \mathbb{E}[\|Y - \hat{Y}\|_2^2] = \sum_{\ell=1}^d \mathbb{E}[(Y_\ell - \hat{Y}_\ell)_2^2], \tag{26}$$

where $(a)$ follows from the invariance of Euclidean distance under orthogonal matrix. Meanwhile,

$$W_2^2(p_X, p_{\hat{X}}) \overset{(b)}{=} W_2^2(p_Y, p_{\hat{Y}}) \overset{(c)}{\geq} \sum_{i=1}^d W_2^2(p_{Y_\ell}, p_{\hat{Y}_\ell}), \tag{27}$$

where $(b)$ follows from the invariance of Wasserstein-2 under unitary transformations; $(c)$ follows from the tensorization property of Wasserstein-2 distance and the equality holds if $(Y_i, \hat{Y}_i)$ and $(Y_j, \hat{Y}_j)$ are independent for any $i \neq j$ (Panaretos & Zemel, 2020). Thus, we can optimize on $p_{\hat{Y}|\tilde{Z}_t}(\hat{\mathbf{y}}|\tilde{\mathbf{z}}_t)$ instead of $p_{\hat{X}|Z_t}(\hat{\mathbf{x}}|\mathbf{z}_t)$ and assume that $\hat{Y} = (\hat{Y}_1, \ldots, \hat{Y}_d)$ has independent components without loss of optimality (Freirich et al., 2021).

Furthermore, as discussed in (Wang et al., 2025) and (Qian et al., 2025), the optimal $\hat{Y}$ must be jointly Gaussian with $Y$, which implies that $\tilde{Z}_t$ and $\hat{Y}$ are jointly Gaussian. Together with the independence of components within each $\tilde{Z}_t$ and $\hat{Y}$, we have $(\tilde{Z}_{t,\ell}, Y_\ell)$ are jointly Gaussian for each $\ell$ and $\{(\tilde{Z}_{t,\ell}, Y_\ell)\}_{\ell=1}^d$ are mutually independent with each other.

For any bivariate Gaussian $U \sim \mathcal{N}(\mu_u, \sigma_u^2)$, $V \sim \mathcal{N}(\mu_v, \sigma_v^2)$, and given covariance $\mathrm{Cov}[U, V] = \theta$, one can express $U$ and $V$ as

$$U \sim \mathcal{N}(\mu_u, \sigma_u^2)$$

$$V = \frac{\theta}{\sigma_u^2}U - \frac{\sigma_v}{\sigma_u}\mu_u + \mu_v + \sqrt{\sigma_v^2 - \frac{\theta^2}{\sigma_u^2}}N, \ N \sim \mathcal{N}(0, 1).$$

Thus, for any possible first and second moments induced by a conditional distribution $p_{\hat{Y}_\ell|Z_{t,\ell}}$ with $(\tilde{Z}_{t,\ell}, \hat{Y}_\ell)$ being jointly Gaussian, we can express the conditional relationship with a linear expression $\hat{Y}_\ell = \tilde{a}_\ell \tilde{Z}_{t,\ell} + \tilde{b}_\ell + \tilde{c}_\ell N$, $N \sim \mathcal{N}(0, 1)$ with adjustable $\tilde{a}_\ell$, $\tilde{b}_\ell$, and $\tilde{c}_\ell$.

In summary, it is sufficient to consider the reconstruction of $\hat{Y}$ as $\hat{Y} = \tilde{\mathbf{A}}\tilde{Z}_t + \tilde{\mathbf{b}} + \tilde{\mathbf{C}}N$, where $N \sim \mathcal{N}(\mathbf{0}, \mathbf{I})$, $\tilde{\mathbf{A}} = \mathrm{diag}(\tilde{a}_1, \ldots, \tilde{a}_d)$ and $\tilde{\mathbf{C}} = \mathrm{diag}(\tilde{c}_1, \ldots, \tilde{c}_d)$ are diagonal matrices with $\tilde{c}_\ell \geq 0$ for $1 \leq \ell \leq d$. Note that $\tilde{Z}_t = \mathbf{Q}^\top Z_t = \sqrt{\bar{\alpha}_t}\mathbf{Q}^\top X + \sqrt{1 - \bar{\alpha}_t}\mathbf{Q}^\top N = \sqrt{\bar{\alpha}_t}Y + \sqrt{1 - \bar{\alpha}_t}N'$, $N' \sim \mathcal{N}(\mathbf{0}, \mathbf{I})$.

$\square$

Now we proceed to derive the optimal solution to the DP tradeoff Eq. (25). For $Y = (Y_1, \ldots, Y_d) \sim \mathcal{N}(\mathbf{Q}^\top \boldsymbol{\mu}_0, \boldsymbol{\Lambda}_0)$, and $\boldsymbol{\Lambda}_0 = \text{diag}(\lambda_1, \ldots, \lambda_d)$, denote

- $\lambda_\ell^{(t)} := \bar{\alpha}_t \lambda_\ell + (1 - \bar{\alpha}_t)$ representing the variance of $\tilde{Z}_{t,\ell}$;

- $\mu_{y,\ell}$ as the mean of $Y_\ell$;

- $\hat{\mu}_{y,\ell} := \tilde{a}_\ell \sqrt{\bar{\alpha}_t} \mu_{y,\ell} + \tilde{b}_\ell$ as the mean of $\hat{Y}_\ell$.

Now, the optimization problem can be written as

$$D_t(P) = \min_{\tilde{\mathbf{A}}, \tilde{\mathbf{b}}, \tilde{\mathbf{C}}} \sum_{\ell=1}^d \mathbb{E}\left[(Y_\ell - \hat{Y}_\ell)_2^2\right] = \sum_{\ell=1}^d \left(\mu_{y,\ell} - \hat{\mu}_{y,\ell}\right)^2 + \lambda_\ell + (\tilde{c}_\ell^2 + \tilde{a}_\ell^2 \lambda_\ell^{(t)}) - 2\tilde{a}\sqrt{\bar{\alpha}_t} \lambda_\ell$$

$$\text{s.t. } \sum_{\ell=1}^d W_2^2(p_{Y_\ell}, p_{\hat{Y}_\ell}) = \sum_{\ell=1}^d \left(\mu_{y,\ell} - \hat{\mu}_{y,\ell}\right)^2 + \left(\sqrt{\lambda_\ell} - \sqrt{\tilde{c}_\ell^2 + \tilde{a}_\ell^2 \lambda_\ell^{(t)}}\right)^2 \leq P.$$

We can set $\mu_{y,\ell} = \hat{\mu}_{y,\ell}$ (i.e., $\tilde{b}_\ell = \mu_{y,\ell} - \tilde{a}_\ell \sqrt{\bar{\alpha}_t} \mu_{y,\ell}$) and $\tilde{c}_\ell = 0$ without loss of optimality. Let $f_\ell = \sqrt{\frac{\tilde{a}_\ell^2 \lambda_\ell^{(t)}}{\lambda_\ell}}$. The optimization problem (25) now becomes

$$D_t(P) = \min_{\{f_\ell\}_{\ell=1}^d} \sum_{\ell=1}^d \lambda_\ell \left(f_\ell - \sqrt{\frac{\bar{\alpha}_t \lambda_\ell}{\lambda_\ell^{(t)}}}\right)^2 + \frac{(1 - \bar{\alpha}_t)\lambda_\ell}{\lambda_\ell^{(t)}}$$

$$\text{s.t. } \sum_{\ell=1}^d \lambda_\ell (1 - f_\ell)^2 \leq P,$$

$$f_\ell \geq 0.$$

Consider the following KKT conditions:

$$\frac{\partial}{\partial f_\ell} \left[\sum_{\ell=1}^d \lambda_\ell \left(f_\ell - \sqrt{\frac{\bar{\alpha}_t \lambda_\ell}{\lambda_\ell^{(t)}}}\right)^2 + \frac{(1 - \bar{\alpha}_t)\lambda_\ell}{\lambda_\ell^{(t)}} + \nu_0 \left(\sum_{\ell=1}^d \lambda_\ell (1 - f_\ell)^2 - P\right) - \sum_{\ell=1}^d \nu_\ell f_\ell\right],$$

$$= 2\lambda_\ell \left(f_\ell - \sqrt{\frac{\bar{\alpha}_t \lambda_\ell}{\lambda_\ell^{(t)}}}\right) + 2\nu_0 \lambda_\ell (f_\ell - 1) - \nu_\ell = 0, \quad \text{for } \ell = 1, \ldots, d, \tag{28}$$

$$\nu_\ell \geq 0, \quad \text{for } \ell = 1, \ldots, d, \tag{29}$$

$$\nu_0 \left(\sum_{\ell=1}^d \lambda_\ell (1 - f_\ell)^2 - P\right) = 0, \tag{30}$$

$$\nu_\ell f_\ell = 0, \quad \text{for } \ell = 1, \ldots, d. \tag{31}$$

From Eq. (28), we can solve $f_\ell = \frac{\sqrt{\bar{\alpha}_t}\sqrt{\lambda_\ell} + \nu_0 \sqrt{\lambda_\ell^{(t)}}}{(1 + \nu_0)\sqrt{\lambda_\ell^{(t)}}} + \frac{\nu_\ell}{2\lambda(1 + \nu_0)}$.

Since $\lambda_\ell > 0$, by Eq. (29) and Eq. (31), we have $\nu_\ell$ must be zero for $\ell = 1, \cdots, d$. Thus, $f_\ell = \frac{\sqrt{\bar{\alpha}_t}\sqrt{\lambda_\ell} + \nu_0 \sqrt{\lambda_\ell^{(t)}}}{(1 + \nu_0)\sqrt{\lambda_\ell^{(t)}}} > 0$. Plugging the value of $f_\ell$ into Eq. (30), we have

$$\nu_0 \left(\sum_{\ell=1}^d \lambda_\ell \left(1 - \frac{\sqrt{\bar{\alpha}_t}\sqrt{\lambda_\ell} + \nu_0 \sqrt{\lambda_\ell^{(t)}}}{(1 + \nu_0)\sqrt{\lambda_\ell^{(t)}}}\right)^2 - P\right) = \nu_0 \left(\frac{1}{(1 + \nu_0)^2} \sum_{\ell=1}^d \frac{\lambda_\ell}{\lambda_\ell^{(t)}} \left(\sqrt{\lambda_\ell^{(t)}} - \sqrt{\bar{\alpha}_t}\sqrt{\lambda_\ell}\right)^2 - P\right) = 0. \tag{32}$$

If $\nu_0 = 0$, we have $f_\ell = \sqrt{\frac{\bar{\alpha}_t \lambda_\ell}{\lambda_\ell^{(t)}}}$, and $P$ should be larger than $\sum_{\ell=1}^{d} \frac{\lambda_\ell}{\lambda_\ell^{(t)}} \left( \sqrt{\lambda_\ell^{(t)}} - \sqrt{\bar{\alpha}_t} \sqrt{\lambda_\ell} \right)^2$ to satisfy primal feasibility. Then, the distortion level is $D_t = \sum_{\ell=1}^{d} \frac{(1-\bar{\alpha}_t)\lambda_\ell}{\lambda_\ell^{(t)}}$.

Here $\tilde{a}_\ell = \frac{\sqrt{\bar{\alpha}_t}\lambda_\ell}{\lambda_\ell^{(t)}}$, $\tilde{b}_\ell = \frac{1-\bar{\alpha}_t}{\lambda_\ell^{(t)}} \mu_{y,\ell}$, $\tilde{c}_\ell = 0$, and the distribution of $\tilde{Y}_\ell$ is $\mathcal{N}(\mu_{y,\ell}, \frac{\bar{\alpha}_t \lambda_\ell^2}{\lambda_\ell^{(t)}})$.

If $\nu_0 > 0$, by Eq. (32), we have $\frac{1}{(1+\nu_0)^2} \sum_{\ell=1}^{d} \frac{\lambda_\ell}{\lambda_\ell^{(t)}} \left( \sqrt{\lambda_\ell^{(t)}} - \sqrt{\bar{\alpha}_t} \sqrt{\lambda_\ell} \right)^2 = P$, which gives us

$$\nu_0 = \sqrt{\frac{1}{P} \sum_{\ell=1}^{d} \frac{\lambda_\ell}{\lambda_\ell^{(t)}} \left( \sqrt{\lambda_\ell^{(t)}} - \sqrt{\bar{\alpha}_t} \sqrt{\lambda_\ell} \right)^2} - 1. \tag{33}$$

In this case, $P < \sum_{\ell=1}^{d} \frac{\lambda_\ell}{\lambda_\ell^{(t)}} \left( \sqrt{\lambda_\ell^{(t)}} - \sqrt{\bar{\alpha}_t} \sqrt{\lambda_\ell} \right)^2$ to ensure $\nu > 0$. With Eq. (33), we can obtain the value of $f_\ell$ as

$$f_\ell = 1 - \frac{\left( 1 - \sqrt{\frac{\bar{\alpha}_t \lambda_\ell}{\lambda_\ell^{(t)}}} \right) \cdot \sqrt{P}}{\sqrt{\sum_{i=1}^{d} \frac{\lambda_i}{\lambda_i^{(t)}} \left( \sqrt{\lambda_i^{(t)}} - \sqrt{\bar{\alpha}_t} \sqrt{\lambda_i} \right)^2}}. \tag{34}$$

Then, plugging Eq. (34) into the expression of $D_t$, we can get the optimal DP curve for multivariate Gaussian case:

$$\begin{aligned}
D_t &= \sum_{\ell=1}^{d} \lambda_\ell \left( 1 + \frac{\left( \sqrt{\frac{\alpha_t \lambda_\ell}{\lambda_\ell^{(t)}}} - 1 \right) \cdot \sqrt{P}}{\sqrt{\sum_{i=1}^{d} \frac{\lambda_i}{\lambda_i^{(t)}} \left( \sqrt{\lambda_i^{(t)}} - \sqrt{\bar{\alpha}_t} \sqrt{\lambda_i} \right)^2}} - \sqrt{\frac{\bar{\alpha}_t \lambda_\ell}{\lambda_\ell^{(t)}}} \right)^2 + \sum_{\ell=1}^{d} \frac{(1-\bar{\alpha}_t)\lambda_\ell}{\lambda_\ell^{(t)}} \\
&= \sum_{\ell=1}^{d} \lambda_\ell \frac{\left( 1 - \sqrt{\frac{\bar{\alpha}_t \lambda_\ell}{\lambda_\ell^{(t)}}} \right)^2 \cdot \left( \sqrt{\sum_{i=1}^{d} \frac{\lambda_i}{\lambda_i^{(t)}} \left( \sqrt{\lambda_i^{(t)}} - \sqrt{\bar{\alpha}_t} \sqrt{\lambda_i} \right)^2} - \sqrt{P} \right)^2}{\sum_{i=1}^{d} \frac{\lambda_i}{\lambda_i^{(t)}} \left( \sqrt{\lambda_i^{(t)}} - \sqrt{\bar{\alpha}_t} \sqrt{\lambda_i} \right)^2} + \sum_{\ell=1}^{d} \frac{(1-\bar{\alpha}_t)\lambda_\ell}{\lambda_\ell^{(t)}} \\
&= \frac{\left( \sqrt{\sum_{i=1}^{d} \frac{\lambda_i}{\lambda_i^{(t)}} \left( \sqrt{\lambda_i^{(t)}} - \sqrt{\bar{\alpha}_t} \sqrt{\lambda_i} \right)^2} - \sqrt{P} \right)^2}{\sum_{i=1}^{d} \frac{\lambda_i}{\lambda_i^{(t)}} \left( \sqrt{\lambda_i^{(t)}} - \sqrt{\bar{\alpha}_t} \sqrt{\lambda_i} \right)^2} \sum_{\ell=1}^{d} \frac{\lambda_\ell}{\lambda_\ell^{(t)}} \left( \sqrt{\lambda_\ell^{(t)}} - \sqrt{\bar{\alpha}_t \lambda_\ell} \right)^2 + \sum_{\ell=1}^{d} \frac{(1-\bar{\alpha}_t)\lambda_\ell}{\lambda_\ell^{(t)}} \\
&= \left( \sqrt{\sum_{i=1}^{d} \frac{\lambda_i}{\lambda_i^{(t)}} \left( \sqrt{\lambda_i^{(t)}} - \sqrt{\bar{\alpha}_t} \sqrt{\lambda_i} \right)^2} - \sqrt{P} \right)^2 + \sum_{\ell=1}^{d} \frac{(1-\bar{\alpha}_t)\lambda_\ell}{\lambda_\ell^{(t)}}.
\end{aligned} \tag{35}$$

### D.3. Achievability

For $Y \sim \mathcal{N}(\boldsymbol{\mu}_y, \boldsymbol{\Lambda}_0)$ and $\tilde{Z}_t = \sqrt{\bar{\alpha}_t} Y + \sqrt{1 - \bar{\alpha}_t} N', N' \sim \mathcal{N}(\mathbf{0}, \mathbf{I})$, let

$$\boldsymbol{\Lambda}_k = \bar{\alpha}_k \boldsymbol{\Lambda}_0 + (1 - \bar{\alpha}_k) \mathbf{I} = \operatorname{diag}(\lambda_1^{(k)}, \dots, \lambda_d^{(k)}), \quad \text{for } k \in \{0, \dots, t\},$$

where $\lambda_\ell^{(k)} = \bar{\alpha}_k \lambda_\ell + (1 - \bar{\alpha}_k)$.

Similar to (Theis et al., 2022), we can decompose the score-scaled PF-ODE into $d$ separate ODEs, each of which is given by

$$d\overleftarrow{Z}_{\tau,\ell} = \left[ -\frac{1}{2}\beta(\tau)\overleftarrow{Z}_{\tau,\ell} - \frac{1}{2}(2-\rho_\ell)\beta(\tau)\nabla_{z_{\tau,\ell}} \log p_{Z_{\tau,\ell}}(\overleftarrow{Z}_{\tau,\ell}) \right] d\tau, \quad \overleftarrow{Z}_{t,\ell} = \tilde{Z}_{t,\ell} \sim \sqrt{\bar{\alpha}_t} Y_\ell + \sqrt{1-\bar{\alpha}_t} N, \tag{36}$$
$$\text{for } \ell = 1, \dots, d.$$

For each $\ell \in \{1, \dots, d\}$, we can always find a $\rho_\ell \in [0, 1]$ such that

$$\prod_{i=0}^{t-1} \sqrt{\rho_\ell + (1 - \rho_\ell)\alpha_{i+1} \frac{\lambda_\ell^{(i)}}{\lambda_\ell^{(i+1)}}} = 1 - \frac{\left( 1 - \sqrt{\frac{\bar{\alpha}_t \lambda_\ell}{\lambda_\ell^{(t)}}} \right) \cdot \sqrt{P}}{\sqrt{\sum_{i=1}^{d} \lambda_i \left( 1 - \sqrt{\frac{\bar{\alpha}_t \lambda_i}{\lambda_i^{(t)}}} \right)^2}}. \tag{37}$$

Then, according to Eq. (23) and Eq. (24), the achievable distortion and Wasserstein levels in each dimension $\ell$ given by the per-dimensional ODE in Eq. (36) is

$$D_\ell = \lambda_\ell \left( 1 - \frac{\left(1 - \sqrt{\frac{\bar{\alpha}_t \lambda_\ell}{\lambda_\ell^{(t)}}}\right) \cdot \sqrt{P}}{\sqrt{\sum_{i=1}^d \lambda_i \left(1 - \sqrt{\frac{\bar{\alpha}_t \lambda_i}{\lambda_i^{(t)}}}\right)^2}} - \sqrt{\frac{\bar{\alpha}_t \lambda_\ell}{\lambda_\ell^{(t)}}} \right)^2 + \frac{(1 - \bar{\alpha}_t)\lambda_\ell}{\lambda_\ell^{(t)}},$$

$$P_\ell = \frac{\lambda_\ell \left(1 - \sqrt{\frac{\bar{\alpha}_t \lambda_\ell}{\lambda_\ell^{(t)}}}\right)^2 \cdot P}{\sum_{i=1}^d \lambda_i \left(1 - \sqrt{\frac{\bar{\alpha}_t \lambda_i}{\lambda_i^{(t)}}}\right)^2}.$$

Denote the reconstruction in dimension $\ell$ as $\hat{Y}_\ell$. Overall, we can obtain that the achievable MSE and Wasserstein-2 divergence is

$$\mathbb{E}[||X - \hat{X}||_2^2] = \mathbb{E}[||Y - \hat{Y}||_2^2] = \sum_{\ell=1}^d \mathbb{E}[(Y_\ell - \hat{Y}_\ell)^2] = \sum_{\ell=1}^d D_\ell$$

$$= \left( \sqrt{\sum_{i=1}^d \frac{\lambda_i}{\lambda_i^{(t)}} \left(\sqrt{\lambda_i^{(t)}} - \sqrt{\bar{\alpha}_t}\sqrt{\lambda_i}\right)^2} - \sqrt{P} \right)^2 + \sum_{\ell=1}^d \frac{(1 - \bar{\alpha}_t)\lambda_\ell}{\lambda_\ell^{(t)}},$$

$$W_2^2(p_X, p_{\hat{X}}) = W_2^2(p_Y, p_{\hat{Y}}) \stackrel{(c)}{=} \sum_{i=1}^d W_2^2(p_{Y_\ell}, p_{\hat{Y}_\ell}) = \sum_{\ell=1}^d P_\ell = P.$$

Thus, the achievable DP tradeoff is

$$D_t^\rho = \left( \sqrt{\sum_{i=1}^d \frac{\lambda_i}{\lambda_i^{(t)}} \left(\sqrt{\lambda_i^{(t)}} - \sqrt{\bar{\alpha}_t}\sqrt{\lambda_i}\right)^2} - \sqrt{P_t^\rho} \right)^2 + \sum_{\ell=1}^d \frac{(1 - \bar{\alpha}_t)\lambda_\ell}{\lambda_\ell^{(t)}},$$

which coincide with the optimal DP tradeoff derived in Eq. (35). The optimal DP tradeoff can be achieved by component-wise reconstruction with delicate design of $\rho_\ell$ as in Eq. (37).

## E. Proof of Theorem 4.1

Let $I_t = I(X; \sqrt{\bar{\alpha}_t}X + \sqrt{1 - \bar{\alpha}_t}N)$. We have

$$\begin{aligned}
I_t &= I(X; \sqrt{\bar{\alpha}_t}X + \sqrt{1 - \bar{\alpha}_t}N) \\
&= h(\sqrt{\bar{\alpha}_t}X + \sqrt{1 - \bar{\alpha}_t}N) - h(\sqrt{\bar{\alpha}_t}X + \sqrt{1 - \bar{\alpha}_t}N|X) \\
&= \frac{1}{2}\log\left(2\pi e(\bar{\alpha}_t \sigma_0^2 + 1 - \bar{\alpha}_t)\right) - \frac{1}{2}\log\left(2\pi e(1 - \bar{\alpha}_t)\right) \\
&= \frac{1}{2}\log\left(\frac{\bar{\alpha}_t}{1 - \bar{\alpha}_t}\sigma_0^2 + 1\right),
\end{aligned}$$

where $h(\cdot)$ denotes the differential entropy for continuous random variables. Given a noising level $t$, the RCC encoder transmits the codeword $M$, and the RCC decoder produces $Z_t = \sqrt{\bar{\alpha}_t}X + \sqrt{1 - \bar{\alpha}_t}N$. Subsequently, the score-scaled PF-ODE reconstructs $\hat{X}^\rho$ for a chosen $\rho$ from $Z_t$.

According to the strong functional representation lemma (Li & Gamal, 2018), the one-shot achievable rate $R_t^1$ is bounded by the cross-entropy between the distribution of $M$ and the Zipf distribution $\text{Zipf}(1 + 1/(I(X; Z_t) + 1))$, i.e.,

$$H(M) \leq I(X; Z_t) + \log(I(X; Z_t) + 1) + 4 \text{ bits}.$$

Thus, the one-shot and asymptotic *achievable* rates (denoted as $R_t^1$ and $R_t^\infty$) are (Li & Gamal, 2018)

$$\begin{aligned}
I_t &\leq R_t^1 \leq I_t + \log(I_t + 1) + 4, \\
R_t^\infty &= I_t.
\end{aligned}$$

According to Eq. (23) and Eq. (24), the *achievable* distortion and perception levels by adjusting compression parameter $t$ and score-scaling parameter $\rho$ are

$$D_t^\rho = \sigma_0^2 + \sigma_0^2 \prod_{i=0}^{t-1} \left( \rho + (1-\rho)\alpha_{i+1} \frac{\sigma_i^2}{\sigma_{i+1}^2} \right) - 2\frac{\sigma_0}{\sigma_t} \prod_{i=0}^{t-1} \left( \rho + (1-\rho)\alpha_{i+1} \frac{\sigma_i^2}{\sigma_{i+1}^2} \right)^{\frac{1}{2}} \sqrt{\bar{\alpha}_t} \sigma_0^2$$

$$= \sigma_0^2 \Big( \prod_{i=0}^{t-1} \big( \rho + (1-\rho)\alpha_{i+1} \frac{\sigma_i^2}{\sigma_{i+1}^2} \big)^{\frac{1}{2}} - \frac{\sqrt{\bar{\alpha}_t}\sigma_0}{\sigma_t} \Big)^2 + \sigma_0^2 - \frac{\bar{\alpha}_t \sigma_0^4}{\sigma_t^2} \tag{23}$$

$$= \sigma_0^2 \Big( f_t^\rho - \frac{\sqrt{\bar{\alpha}_t}\sigma_0}{\sigma_t} \Big)^2 + \sigma_0^2 - \frac{\bar{\alpha}_t \sigma_0^4}{\sigma_t^2}, \tag{38}$$

$$P_t^\rho = \Big( \sigma_0 - \sigma_0 \prod_{i=0}^{t-1} \big( \rho + (1-\rho)\alpha_{i+1} \frac{\sigma_i^2}{\sigma_{i+1}^2} \big)^{\frac{1}{2}} \Big)^2 \tag{24}$$

$$= \sigma_0^2 \big( 1 - f_t^\rho \big)^2, \tag{39}$$

where $f_t^\rho := \prod_{i=0}^{t-1} \big( \rho + (1-\rho)\alpha_{i+1} \frac{\sigma_i^2}{\sigma_{i+1}^2} \big)^{\frac{1}{2}}$.

From (Zhang et al., 2021), the optimal RDP tradeoff for the scalar Gaussian source $X \sim \mathcal{N}(\mu_0, \sigma_0^2)$ is

$$R(D, P) = \begin{cases} \frac{1}{2} \log \frac{\sigma_0^2 (\sigma_0 - \sqrt{P})^2}{\sigma_0^2 (\sigma_0 - \sqrt{P})^2 - (\sigma_0^2 + (\sigma_0 - \sqrt{P})^2 - D)^2/4} & \text{if } \sqrt{P} < \sigma_0 - \sqrt{|\sigma_0^2 - D|}, \\ \max\{\frac{1}{2} \log \frac{\sigma_0^2}{D}, 0\} & \text{if } \sqrt{P} \geq \sigma_0 - \sqrt{|\sigma_0^2 - D|}. \end{cases}$$

First, for $0 < \rho \leq 1$, we have $f_t^\rho = \prod_{i=0}^{t-1} \big( \rho + (1-\rho)\alpha_{i+1} \frac{\sigma_i^2}{\sigma_{i+1}^2} \big)^{\frac{1}{2}}$ falls in $(\frac{\sqrt{\bar{\alpha}_t}\sigma_0}{\sigma_t}, \ 1]$, which implies that $\sqrt{P_t^\rho} < \sigma_0 - \sqrt{|\sigma_0^2 - D_t^\rho|}$. Plugging Eq. (38) and Eq. (39) into the first case of $R(D, P)$ function, we have

$$R(D_t^\rho, P_t^\rho) = \frac{1}{2} \log \left( \frac{\sigma_0^2 \cdot \sigma_0^2 (f_t^\rho)^2}{\sigma_0^2 \cdot \sigma_0^2 (f_t^\rho)^2 - \big( \sigma_0^2 + \sigma_0^2 (f_t^\rho)^2 - \sigma_0^2 (f_t^\rho - \frac{\sqrt{\bar{\alpha}_t}\sigma_0}{\sigma_t})^2 - \frac{1-\bar{\alpha}_t}{\sigma_0} \big)^2/4} \right)$$

$$= \frac{1}{2} \log \left( \frac{4\sigma_0^4 (f_t^\rho)^2}{4\sigma_0^4 (f_t^\rho)^2 - 4\frac{\bar{\alpha}_t \sigma_0^2}{\sigma_t^2} (f_t^\rho)^2 \sigma_0^4} \right)$$

$$= \frac{1}{2} \log \left( \frac{\bar{\alpha}_t}{1 - \bar{\alpha}_t} \sigma_0^2 + 1 \right).$$

For the second case, when $\rho = 0$, $\sqrt{P_t^\rho} = \sigma_0 - \sqrt{|\sigma_0^2 - D_t^\rho|}$, the distortion now become the MMSE value

$$D_t^0 = \sigma_0^2 - \frac{\bar{\alpha}_t \sigma_0^4}{\sigma_t^2},$$

and the optimal rate is

$$R(D_t^0, P_t^0) = \frac{1}{2} \log \frac{\sigma_0^2}{D_t^0} = \frac{1}{2} \log \left( \frac{\bar{\alpha}_t}{1 - \bar{\alpha}_t} \sigma_0^2 + 1 \right).$$

We can observe that in both cases, the optimal rate $R(D, P)$ given distortion level $D_t^\rho$ and perception level $P_t^\rho$ is $I_t = \frac{1}{2} \log \left( \frac{\bar{\alpha}_t}{1 - \bar{\alpha}_t} \sigma_0^2 + 1 \right)$, which coincides with the asymptotic achievable rate $I_t$ provided by PFR when transmitting $Z_t \sim \sqrt{\bar{\alpha}_t} X + \sqrt{1 - \bar{\alpha}_t} N$.

# F. More Experimental Results

## F.1. CIFAR-10 Dataset

**Experimental details:** We use a pre-trained diffusion model provided by a third-party repository[3], which follows the original DDPM setup (Ho et al., 2020). The training details can be found in the original repository. For FID computation,

[3] https://github.com/w86763777/pytorch-ddpm

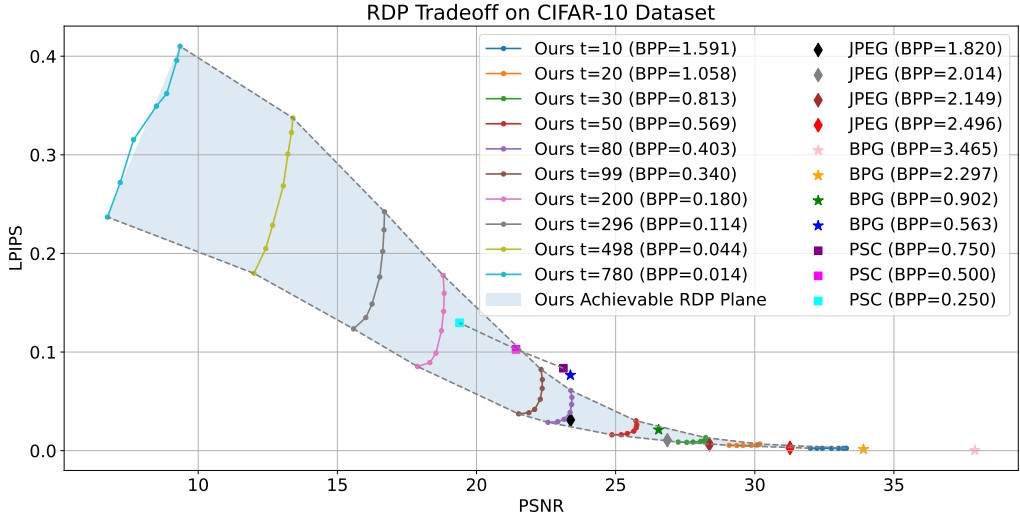

*Figure 7.* RDP curves on CIFAR-10 using PSNR vs. LPIPS.

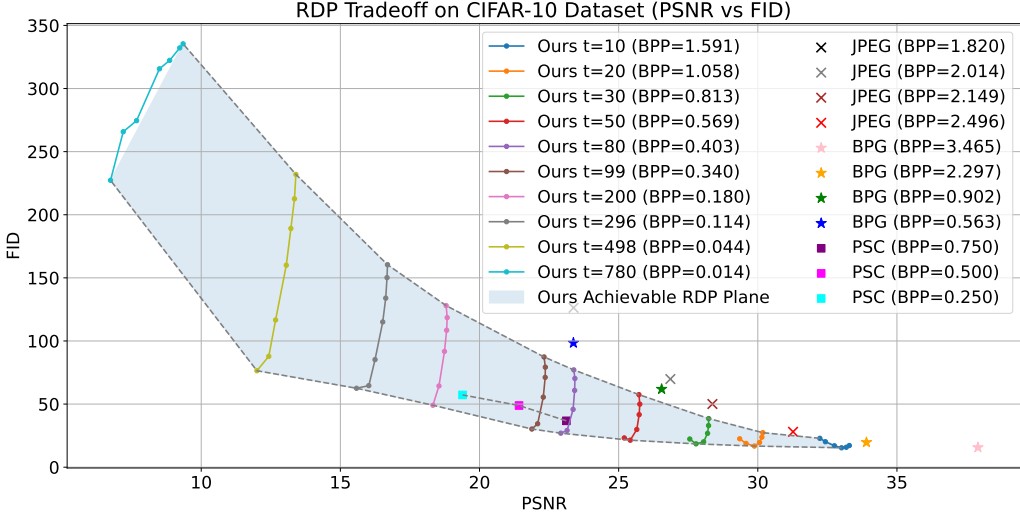

*Figure 8.* RDP curves on CIFAR-10 using PSNR vs. FID.

we compress and reconstruct 2,000 samples, extract features using the pre-trained Inception network, and compute the Fréchet distance using the empirical means and covariances of real and generated features. To generate the RDP curves in Figure 4, we vary the score-scaling parameter $\rho \in \{0.5, 0.6, 0.7, 0.8, 0.9, 0.95, 1\}$. All experiments are conducted on a single NVIDIA A100 GPU.

**More results:** In Figures 7 and 8, we present additional RDP results using PSNR as the distortion metric, and LPIPS and FID as the perception metrics. The tradeoff behavior remains consistent with our theoretical analysis: a higher $\rho$ leads to better perceptual quality but lower PSNR. We also provide qualitative examples in Figure 9 showcasing reconstructions under different $t$ and $\rho$. These results further demonstrate the smooth and controllable tradeoff enabled by our method.

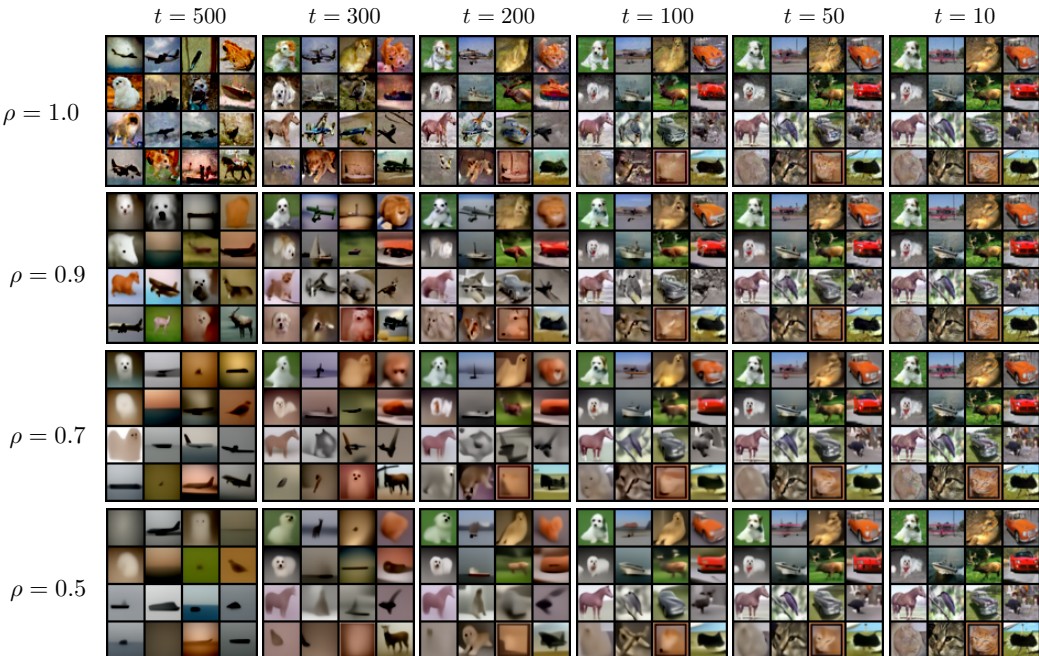

*Figure 9.* Sample reconstructions on CIFAR-10 under varying $t$ and $\rho$. Higher $\rho$ produces more vivid but less faithful images.

## F.2. Kodak and DIV2K Datasets

### F.2.1. EXPERIMENTAL DETAILS

We evaluate our method using two high-resolution datasets: Kodak (24 images of size $768 \times 512 \times 3$) and DIV2K validation (100 images at 2K resolution). We use pre-trained latent diffusion models, including Stable Diffusion (versions 1.5, 2.1, and SDXL)(Rombach et al., 2022) and Flux (Black-Forest-Labs et al., 2025). We follow the DiffC implementation (Vonderfecht & Liu, 2025) for the diffusion noise schedule and the CUDA-accelerated implementation of the PFR algorithm, specifically,

- **Solver, steps, and guidance**: We use a standard 50-step DDIM scheduler (Song et al., 2021a). Actual denoising steps scale with the bitrate (e.g., 13 steps for $t = 200$ at bpp=0.024). Since step counts are fixed, tolerances do not apply. No guidance is used.

- **RCC/PFR candidates**: Following the design in (Vonderfecht & Liu, 2025), we transmit several chunks of 12-16 bits (about 2-10 chunks for $t > 450$ and about 50 chunks for $t = 20$), resulting in a bounded candidate pool of $2^{12}$ to $2^{16}$. Generating and encoding a 16-bit chunk takes $< 3$ms.

- **Wall-clock time**: Latency scales with bitrate due to ODE steps. At low bitrates, encoding/decoding is comparable to or faster than HiFiC and CDC. At high bitrates, it is slower but remains significantly faster than DDCM (which uses 1000 DDPM steps). All experiments are conducted on a single NVIDIA A100 GPU.

Note that we evaluate the coding rate using the length of the actually compressed bitstream from an implementable encoder, not theoretical predictions.

For DDCM (Ohayon et al., 2025), we follow their official implementation and configurations on the rate control. Specifically, we set $(K, M, C) = (256, 1, 1), (8192, 1, 1), (2048, 2, 3), (2048, 3, 3)$ according to their paper to obtain the reported points.

In Figure 5, we choose different timesteps $t \in \{498, 296, 200, 99, 80, 50, 30, 20, 10\}$. Note that other choices of $t$ along the process of reverse diffusion sampling are also possible. The score-scaling parameter $\rho$ is varied on the latent space to explore the distortion-perception tradeoff under different compression levels $t$. Specifically, we use the following $\rho$ values for each model:

- Stable Diffusion 2.1: $\rho \in \{0.75, 0.83, 0.85, 0.88, 0.9, 0.92, 0.93, 0.95, 1\}$,

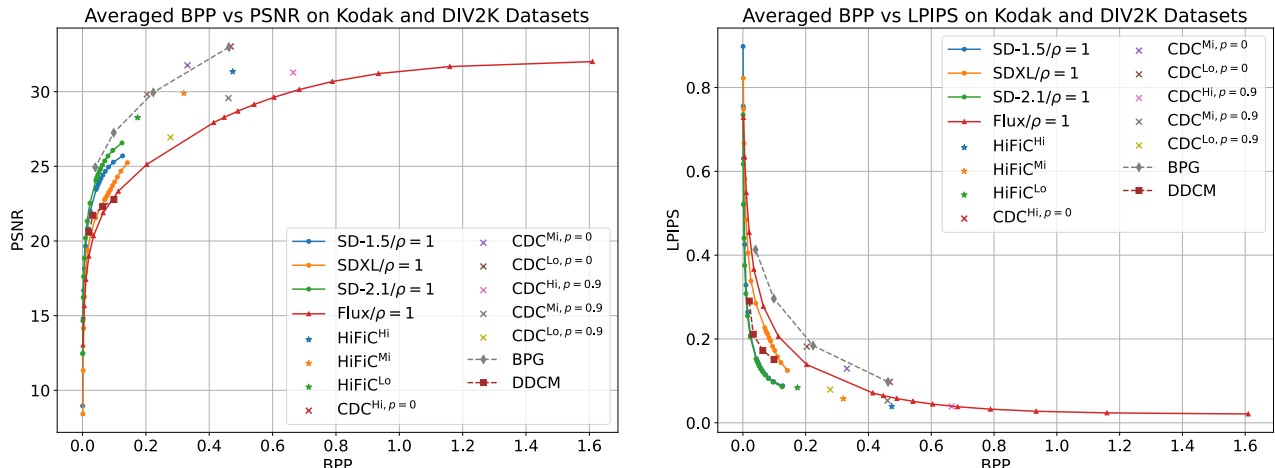

*Figure 10.* RDP metrics under $\rho = 1$ for SD1.5/2.1/XL and Flux.

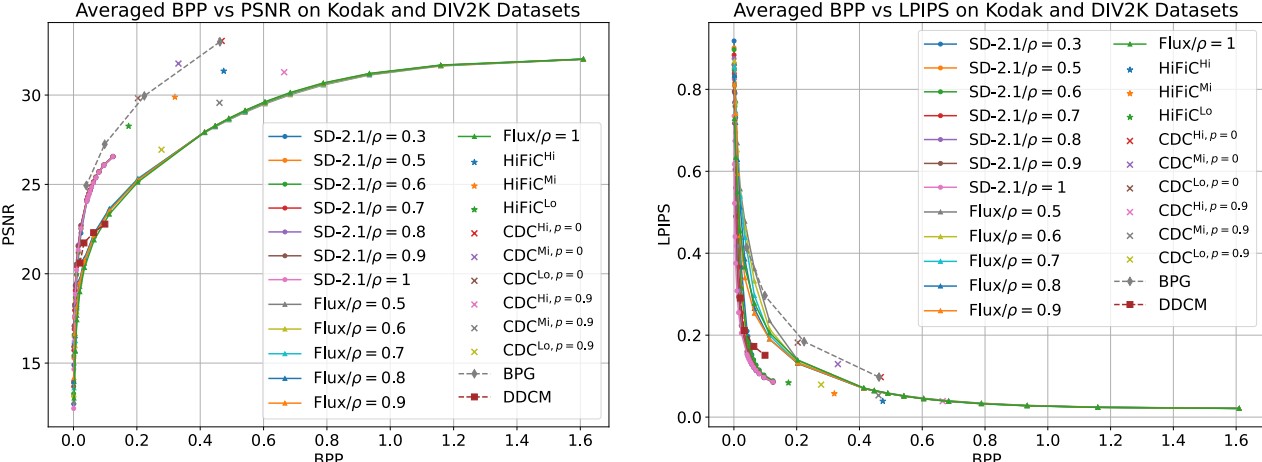

*Figure 11.* Effect of controlling $t$ and $\rho$ on different metrics for the Kodak and DIV2K datasets. Stable Diffusion 2.1 and Flux depict different rate-distortion (R-D) and rate-perception (R-P) curves.

- Flux: $\rho \in \{0.7, 0.75, 0.8, 0.85, 0.88, 0.9, 0.92, 0.95\}$.

The general rule is that increasing $\rho$ consistently improves perception (lower FID/LPIPS) at the cost of MSE. Because different models (e.g., SD vs. Flux) and datasets have distinct baseline statistical distributions, a per-model/per-dataset calibration is recommended. *This is a one-time calibration*; then the target $\rho$ values can be reused during inference on this dataset. Importantly, our scheme only utilizes pre-trained diffusion models and requires no additional training for specific $t$ or $\rho$ values.

### F.2.2. MORE RESULTS

**Effect of RDP Control Parameters $t$ and $\rho$ on Different Models:** We include the distortion and perception performance results for SD1.5/2.1/XL and Flux under $\rho = 1$ in Figure 10. We can observe that the R-D and R-P performances of SD1.5 and SDXL are inferior to SD2.1. Thus, we choose SD2.1 in our experiments when comparing with the benchmark.

Figure 11 shows the PSNR and LPIPS results across different $\rho$ and $t$ values for both SD 2.1 and Flux. While SD 2.1 performs better at low bitrates, Flux supports broader RDP traversal at higher bitrates. In both cases, increasing $\rho$ improves perception but leads to worse distortion, consistent with theoretical predictions.

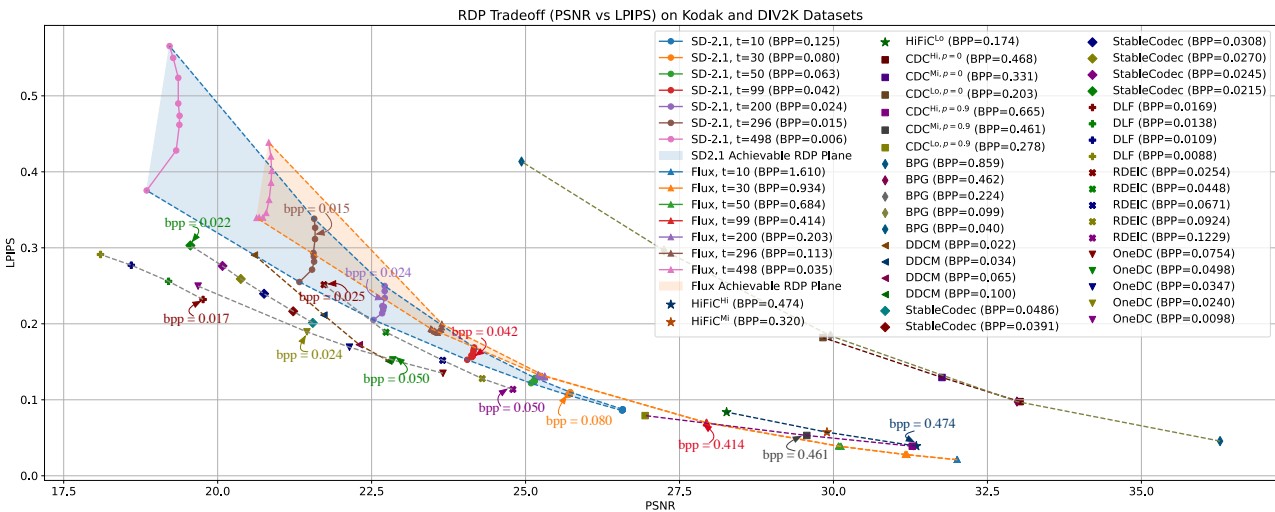

*Figure 12.* RDP curves for Kodak and DIV2K using PSNR vs. LPIPS under SD2.1 and Flux.

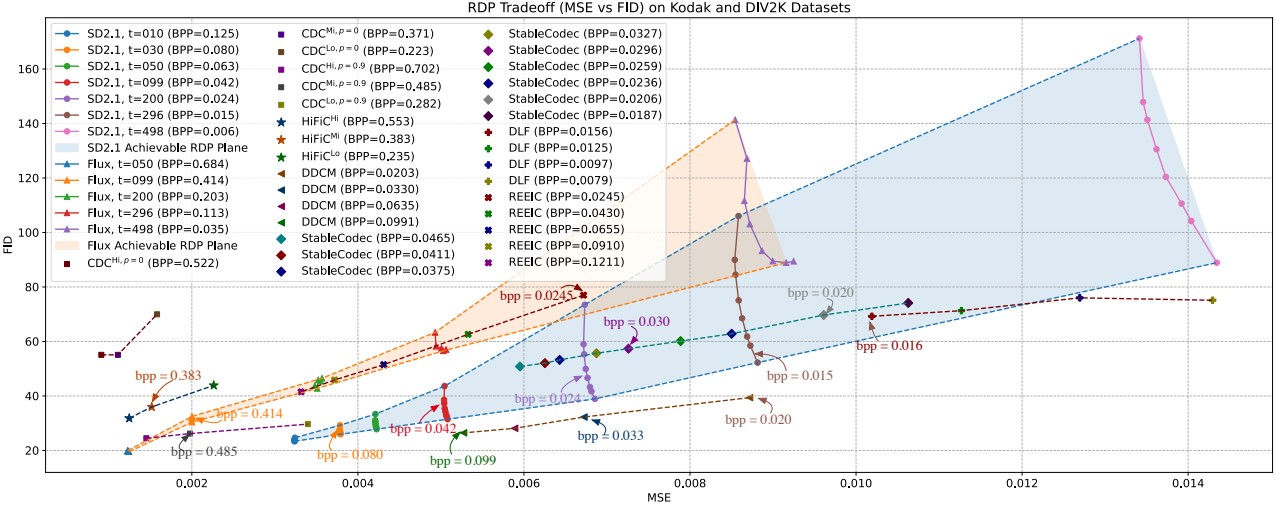

*Figure 13.* RDP curves for Kodak using MSE vs. FID under SD2.1 and Flux.

**More DP metrics:** We provide additional RDP results using PSNR-LPIPS curves for SD2.1 and Flux across both datasets in Figure 12. We also report the FID on the Kodak dataset and plot the rate-MSE-FID tradeoff in Figure 13. Similar trends are observed as in MSE-LPIPS results. Note that the computation of FID here follows Mentzer et al. (2020); Ohayon et al. (2025), wherein $64 \times 64$ patches are extracted from the high resolution images to compute the FID scores on these patches. FID is mathematically defined as the squared Wasserstein-2 distance between two multivariate Gaussian distributions fitted to the Inception feature representations of the real and generated images. We also provide tables of numerical results in Table 3.

**More samples:** Figures 14 and 15 present more sample reconstructions on Kodak and DIV2K datasets with diverse $\rho$ selections and bitrate levels against baselines. Figure 16 depicts the visual changes in reconstructions provided by Flux under different $t$ and $\rho$. Figures 17 and 18 show samples with high-resolution details. Note that we also provide an interactive demo online [4] to help the reader compare the details of images for different values of $t$ and $\rho$. These qualitative results further illustrate the smooth and controllable RDP tradeoff achieved by our method.

---

[4]https://diffrdp.github.io/

*Table 3.* MSE and FID values across different $\rho$ and $t$ values on Kodak dataset (SD 2.1)

| MSE ($\times 10^{-3}$) /FID | $t$ values | | | | | | |
|---|---|---|---|---|---|---|---|
| | 10 | 30 | 50 | 99 | 200 | 296 | 498 |
| $\rho = 0.83$ | 3.239/ 24.158 | 3.782/ 27.854 | 4.209/ 31.040 | 5.038/ 38.579 | 6.718/ 59.002 | 8.539/ 89.957 | 13.458/ 147.872 |
| $\rho = 0.85$ | 3.238/ 24.026 | 3.783/ 27.529 | 4.210/ 30.425 | 5.040/ 37.295 | 6.725/ 55.286 | 8.549/ 84.503 | 13.509/ 141.376 |
| $\rho = 0.88$ | 3.238/ 23.823 | 3.784/ 27.026 | 4.213/ 29.601 | 5.046/ 35.397 | 6.744/ 49.927 | 8.586/ 75.094 | 13.619/ 130.549 |
| $\rho = 0.90$ | 3.237/ 23.692 | 3.786/ 26.734 | 4.216/ 29.067 | 5.054/ 34.241 | 6.766/ 46.616 | 8.628/ 68.504 | 13.732/ 120.375 |
| $\rho = 0.92$ | 3.237/ 23.544 | 3.788/ 26.405 | 4.219/ 28.509 | 5.062/ 33.085 | 6.795/ 43.259 | 8.688/ 61.814 | 13.921/ 110.608 |
| $\rho = 0.93$ | 3.238/ 23.477 | 3.789/ 26.250 | 4.222/ 28.266 | 5.068/ 32.538 | 6.813/ 41.707 | 8.727/ 58.462 | 14.038/ 104.204 |
| $\rho = 0.95$ | 3.237/ 23.358 | 3.791/ 25.933 | 4.226/ 27.784 | 5.080/ 31.470 | 6.855/ 38.907 | 8.815/ 52.233 | 14.345/ 88.912 |

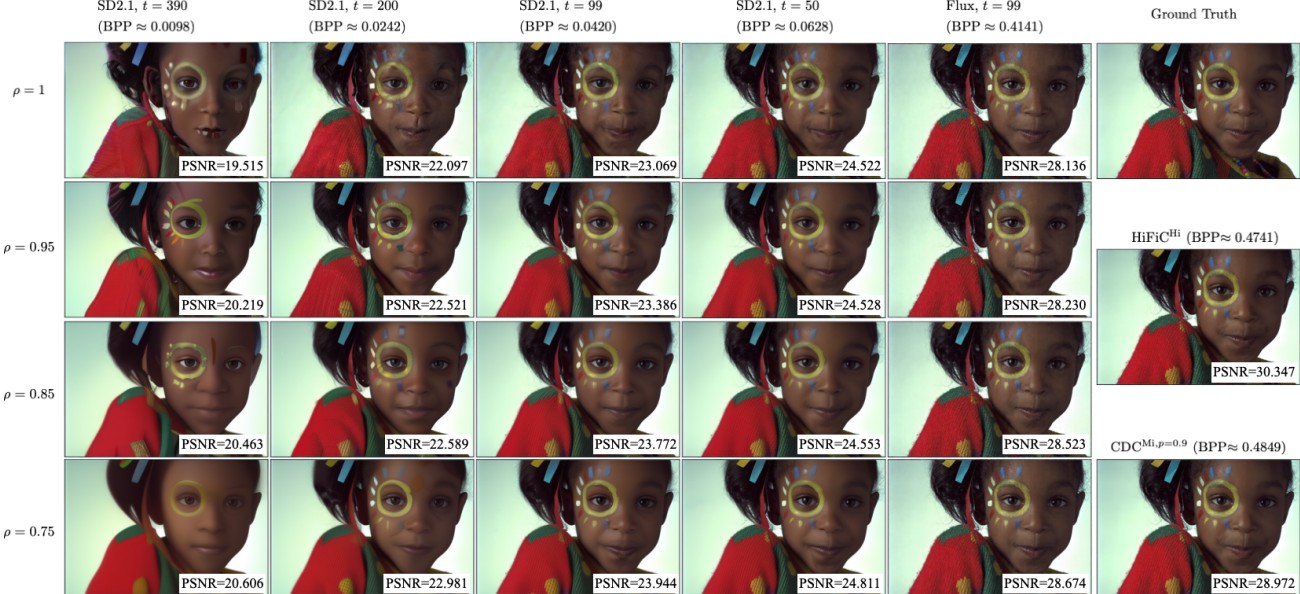

*Figure 14.* Sample reconstructions on Kodak dataset under different $t$ and $\rho$.

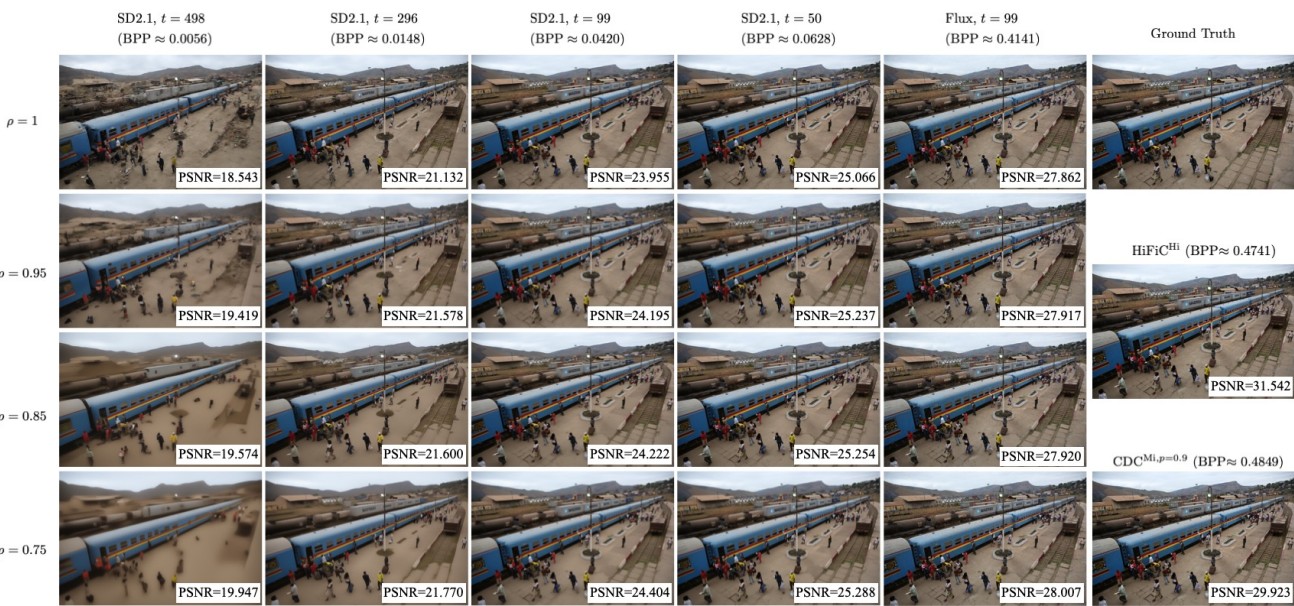

*Figure 15.* Sample reconstructions on DIV2K dataset under different $t$ and $\rho$.

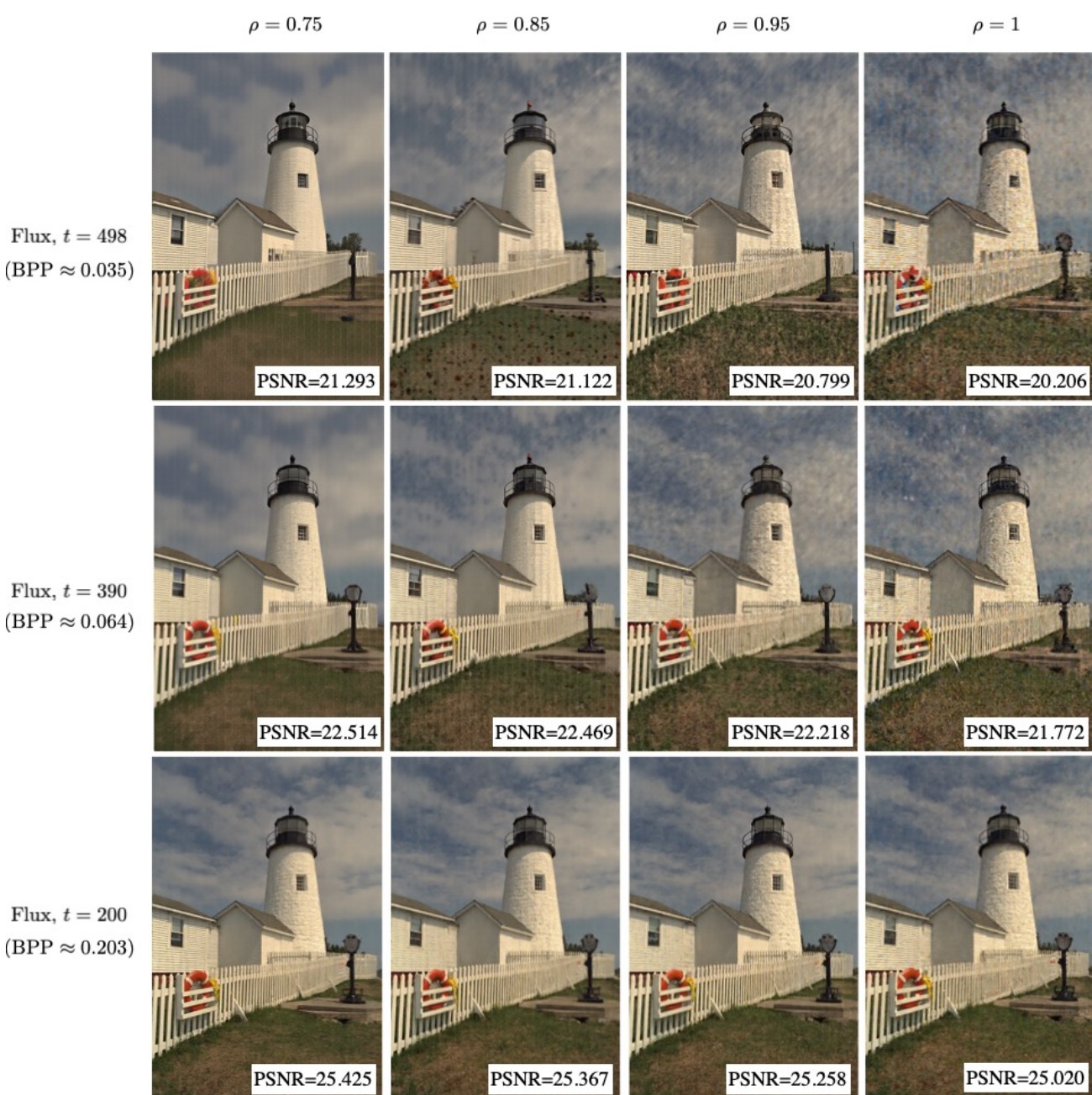

*Figure 16.* Sample reconstructions provided by Flux under different $t$ and $\rho$.

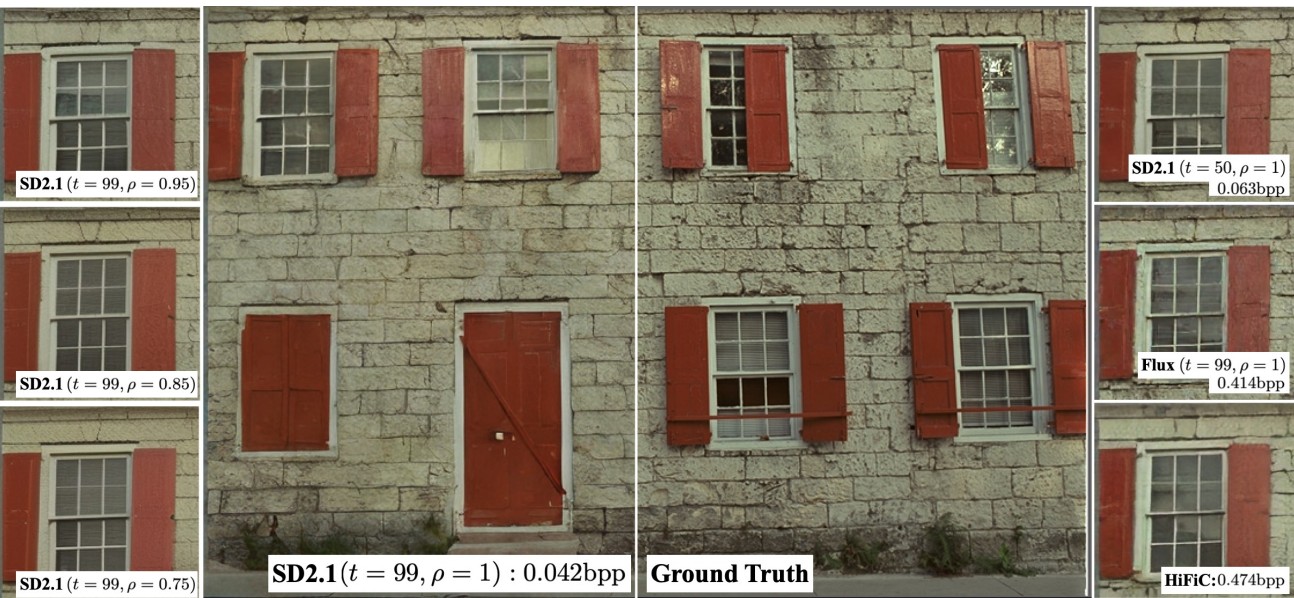

*Figure 17.* Sample reconstructions with high resolution details under different $t$ and $\rho$.

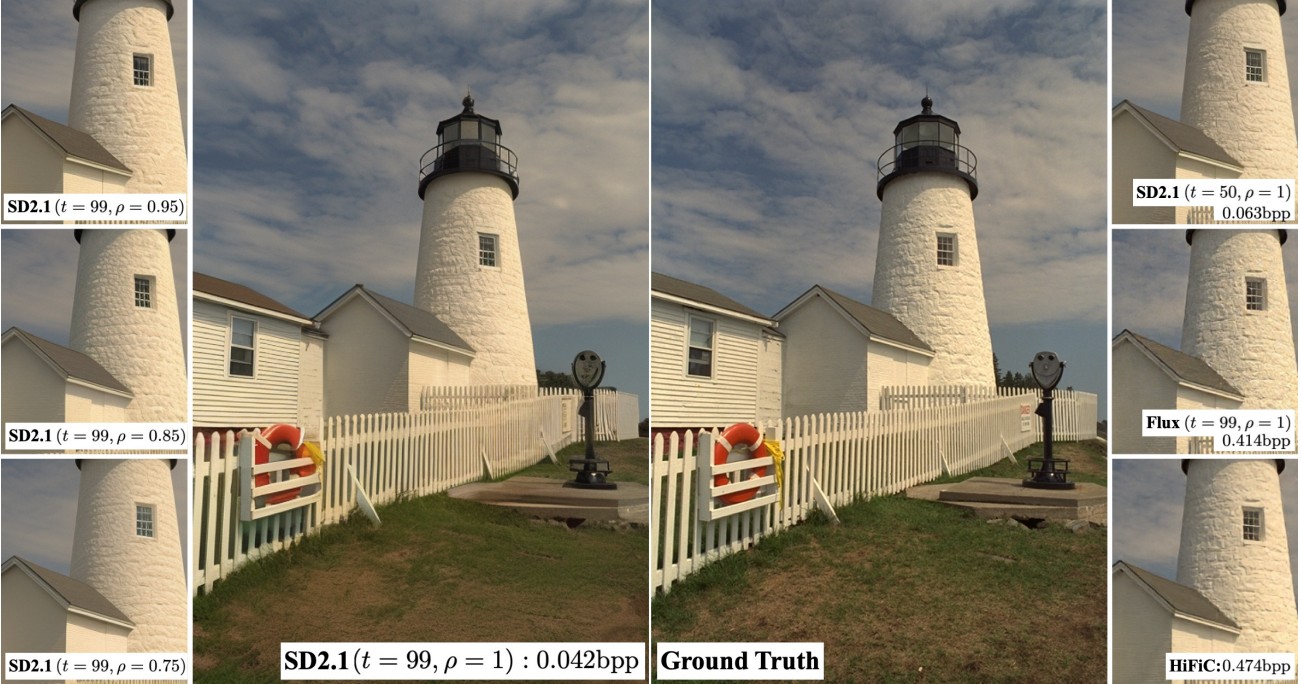

*Figure 18.* Sample reconstructions with high resolution details under different $t$ and $\rho$.

