# OpenReview forum: "Training-Free Rate-Distortion-Perception Traversal With Diffusion"
_ICML.cc/2026/Conference — ICML 2026 regular_

### Official Review · Reviewer_5bmn · 2026-03-02

**Soundness:** 4
**Presentation:** 4
**Significance:** 4
**Originality:** 4
**Overall Recommendation:** 5
**Confidence:** 4

**Summary:**

This paper proposes a training-free framework for traversing the full rate–distortion–perception (RDP) tradeoff using a single pre-trained diffusion model. Building on DiffC, it combines reverse channel coding (RCC) to control bitrate via the diffusion time index with a score-scaled probability flow ODE decoder that introduces a continuous parameter to navigate the distortion–perception axis, interpolating between MMSE-like reconstructions and high-realism samples. The authors provide information-theoretic guarantees, showing optimal distortion–perception performance under AWGN for Gaussian sources and achieving the scalar-Gaussian RDP function when RCC is included. Experiments on CIFAR-10, Kodak, and DIV2K (including latent diffusion backbones such as Stable Diffusion and Flux) demonstrate flexible, continuous control over bitrate, distortion, and perceptual quality, often outperforming classical codecs and diffusion-based baselines at comparable rates—all without retraining.

**Compliance With Llm Reviewing Policy:**

Affirmed.

**Final Justification:**

The rebuttal was helpful and increased my confidence in the paper. I appreciate the added implementation details and rebuttal figures, which improve the paper’s transparency and clarify several practical aspects of the method. The paper’s main contribution remains strong: a training-free framework for traversing the full rate-distortion-perception tradeoff with a single pretrained diffusion model, supported by a clear control mechanism through t and ρ and by a meaningful theoretical analysis in Gaussian settings.

Some concerns remain partially unresolved. In particular, the response does not fully establish compute-matched fairness against baselines, and the connection between the theoretical perception notion and empirical metrics such as FID/LPIPS, while improved, remains somewhat indirect. I also think the strongest theoretical claims should be interpreted primarily in the idealized Gaussian regime.

Overall, I remain positive on the paper. The work is original, well executed, and addresses an important problem with a compelling combination of flexibility, theory, and experimental breadth.

**Key Questions For Authors:**

1. Compute/latency matching across methods: For every point on your RDP curves, what are the exact inference settings—ODE/SDE solver type, step count, tolerances, guidance settings (if any), and (critically) RCC/PFR candidate counts—and were these matched to baselines under an equal compute budget (time or FLOPs) at both encoding and decoding?
2. Calibration and transferability of the $\rho$ knob: Is $\rho$ consistently interpretable across datasets, resolutions, and backbones (e.g., SD vs Flux), or does it require per-model/per-dataset calibration? Can you provide a recommended calibration procedure and evidence of stability (e.g., a mapping from $\rho$ to target LPIPS/FID at fixed rate)?
3. Alignment between theoretical "perception" and experimental metrics: Your theory frames perception using distributional distances (e.g., W_2/TV-style notions), while experiments report FID/LPIPS. Do you have either (i) empirical evidence that improvements under your method are consistent across multiple perception metrics, including ones closer to the theory, or (ii) a clear argument for why your theoretical perception notion meaningfully supports the FID/LPIPS results?
4.  Other questions in the Weaknesses.

**Limitations:**

No. Please refer to Weakness.

**Strengths And Weaknesses:**

Strengths
1. Clear, unified objective: Frames the contribution as training-free traversal of the full rate-distortion-perception (RDP) tradeoff, rather than optimizing a few isolated operating points.
2. Interpretable control knobs: Uses $t$ primarily for bitrate control and $\rho$ for distortion-perception steering, with experimental curves that qualitatively match this separation.
3. Theory provides a solid anchor in idealized settings: Offers information-theoretic analysis for Gaussian sources under AWGN, supporting the interpretation of $\rho$ as a principled DP control rather than a purely heuristic hyperparameter.
4. Solid experiments validation: Demonstrates the traversal behavior on multiple datasets (e.g., CIFAR-10, Kodak, DIV2K) and on modern latent diffusion backbones (e.g., SD/Flux-style models), including  both classical codecs (e.g., JPEG/BPG) and learned/generative compression baselines.

Weaknesses
1. Insufficient detail on computational settings: provide a standardized table of (i) sampling steps, (ii) solver type/tolerances, (iii) RCC/PFR candidate counts, (iv) hardware, (v) wall-clock measurement protocol. Without this, "training-free" may still hide large inference costs.
2. Clarify the semantics/calibration of $\rho$: since the effective $\rho$ range may differ in latent diffusion settings, readers need guidance on how to choose $\rho$ in practice and how stable its effect is across models/datasets.
3. Large theory-practice gap: Optimality guarantees rely on Gaussian/AWGN assumptions; real images and learned diffusion scores deviate significantly, so "optimal RDP traversal" is not fully justified beyond the ideal regime.
4. Perception metric mismatch: Theory often uses distributional notions (e.g., W_2/TV-style distances), while experiments rely on FID/LPIPS; the link between these is informal, weakening the "theory validates experiments" narrative.

---

> ### Author Rebuttal · Authors · 2026-03-31
>
> _**Response to W1 & Q1: Computational Settings and Latency**_
>
> We thank the reviewer for the constructive comments. We follow the DiffC implementation (Vonderfecht & Liu, 2025) and will include a standardized table of these settings in the revision:
>
> - **Solver, steps, and guidance:** We use a standard 50-step DDIM scheduler. Actual denoising steps scale with the bitrate (e.g., 13 steps for $t=200$ at bpp=0.024). Since step counts are fixed, tolerances do not apply. No guidance is used.
>
> - **RCC/PFR candidate**: Following the design in Vonderfecht & Liu (2025), we transmit several chunks of 12–16 bits (about 2~10 chunks for $t>450$ and about 50 chunks for $t=20$), resulting in a bounded candidate pool of $2^{12}$ to $2^{16}$. Generating and encoding a 16-bit chunk takes <3ms.
>
> - **Wall-clock time**: As shown in Table 1, latency scales with bitrate due to ODE steps. At low bitrates, encoding/decoding is comparable to or faster than HiFiC and CDC. At high bitrates, it is slower but remains significantly faster than DDCM (which uses 1000 DDPM steps). All experiments are conducted on a single NVIDIA A100 GPU.
>
> While we mainly focus on the initial exploration of the potential to traverse the RDP plane with pre-trained diffusion models, **the latency can be reduced through both engineering and methodological improvements**. Because our method is agnostic to the specific solver and RCC backend, future improvements in fast ODE solvers and RCC implementations will directly reduce this latency.
>
> _**Response to W2 & Q2: On calibration and transferability of $\rho$**_
>
> We thank the reviewer for the insightful comment and question. The general rule is that increasing $\rho$ consistently improves perception (lower FID/LPIPS) at the cost of MSE. Because different models (e.g., SD vs. Flux) and datasets have distinct baseline statistical distributions, a per-model/per-dataset calibration is recommended. **This is a one-time calibration**; then the target $\rho$'s can be reused during the inference on this dataset. Importantly, our scheme only utilizes pre-trained diffusion models and requires no additional training for specific $t$ or $\rho$ values.
>
> However, when using the same model, $\rho$ transfers stably across datasets with similar characteristics. For example, using Stable Diffusion 2.1 at a fixed compression rate, the mapping from $\rho$ to DP performance is similar between the Kodak and Div2K datasets (see Figure R3 in https://diffrdprebuttal.github.io/).
>
> _**Response to W3: Theory-practice gap**_
>
> Deriving closed-form RDP functions for non-Gaussian sources remains an open problem in information theory. **Analyzing the Gaussian case is the standard, necessary first step to establish theoretical optimality** (e.g., Zhang et al., 2021) and provides a principled derivation for *why* score scaling controls the tradeoff. Practically, latent diffusion models (like SD2.1) explicitly use a KL-regularized latent space to approximate a standard Gaussian (Rombach et al., 2022), bridging our theory with empirical practice.
>
> Furthermore, **the core intuition of our method holds for non-Gaussian data**: the variance of the reconstruction distribution scales with $\rho$. A smaller $\rho$ yields lower variance (favoring distortion), while a larger $\rho$ yields higher variance (favoring perception).  We demonstrate this using a Gaussian mixture source $p_X = w_1*\mathcal{N}(u_1, \sigma_1^2) + w_2*\mathcal{N}(u_2, \sigma_2^2)$ and $Y=\sqrt{\bar{\alpha}_t}X + \sqrt{1-\bar{\alpha}_t}N$ with $N\sim\mathcal{N}(\mathbf 0, \mathbf I)$. As shown in Figure R1 (https://diffrdprebuttal.github.io/), the ODE sampling trajectories correctly reflect the intended effect of $\rho$ and $t$. **We will discuss this limitation and potential extensions via interpolating ODEs in the revision.**
>
> _**Response to W4 & Q3: On perception metric**_
>
> We thank the reviewer for the comment. FID is mathematically defined as the Wasserstein-2 distance between two multivariate Gaussian distributions fitted to the Inception-v3 feature representations of the real and generated images. Therefore, our theoretical framework supports the empirical improvements measured by FID. Since exact high-dimensional image distributions are intractable, FID and LPIPS serve as the standard, theoretically grounded proxies for human perception, which have been adopted by numerous existing works.

---

> > ### Author Rebuttal · Reviewer_5bmn · 2026-04-03
> >
> > The rebuttal is helpful and increases my confidence in the paper. I appreciate the added implementation details and the new rebuttal figures. Some concerns remain partially unresolved. The response improves transparency on runtime settings, but it still does not fully establish compute-matched fairness against baselines. The discussion connecting the theoretical perception notion to empirical metrics such as FID/LPIPS is also improved, but still somewhat indirect. Overall, I remain positive on the paper.

---

> > > ### Author Response · Authors · 2026-04-06
> > >
> > > We sincerely thank the reviewer for the continued support, constructive feedback, and for acknowledging the improvements in our rebuttal. We are glad that the added details and figures increased your confidence in our work.
> > >
> > > Regarding your remaining concerns on compute-matched fairness and perception metrics, we would like to offer a few follow-up clarifications:
> > >
> > > - **_Compute-matched fairness and storage costs_**: We acknowledge that distilled, one-step models are inherently faster at decoding. As shown in our updated Table 1 (Figure R4 at https://diffrdprebuttal.github.io/), StableCodec and OneDC achieve decoding times of 0.47s and 0.43s, respectively, compared to our 0.33s–2.10s (depending on the chosen timestep $t$).
> > >
> > >   However, when evaluating fairness, we must also consider the effective storage cost required for flexibility. Our method utilizes a single, pre-trained SD-2.1 model (950M parameters). In contrast, the base models of StableCodec (1.1B) and OneDC (1.4B) are larger, and specific modules must be retrained for different bitrates. To cover the same wide range of bitrates and DP tradeoffs that our single model can traverse, one would need to retrain and store dozens of separate StableCodec or OneDC modules ($N \times 492\text{M}$ for OneDC, and $N \times 109\text{M}$ for StableCodec, where $N$ is the number of RDP points to be traversed). This makes their practical deployment size prohibitively large for adaptive streaming scenarios.
> > >
> > >   This comparison highlights a fundamental tradeoff in current generative compression: **highly optimized, task-specific distillation** (StableCodec/OneDC) versus **training-free, universal flexibility** (Ours). Both directions are highly valuable. Because our score-scaled ODE framework is theoretically grounded, future work could potentially integrate our traversal mechanisms with distilled one-step models to achieve both ultra-fast inference and RDP flexibility.
> > >
> > > - **_Bridging theoretical perception and empirical metrics:_** To better connect these concepts, we have added a clarifying paragraph in the main text. As noted in previous responses, we will highlight that FID is mathematically defined as the Wasserstein-2 distance between two multivariate Gaussian distributions fitted to the Inception feature representations of the real and generated images.
> > >
> > > Thanks again for your insightful reviews throughout this process. Your feedback has been instrumental in improving the transparency, fairness, and theoretical clarity of our paper.

---

### Official Review · Reviewer_ncaM · 2026-03-05

**Soundness:** 1
**Presentation:** 2
**Significance:** 1
**Originality:** 2
**Overall Recommendation:** 3
**Confidence:** 5

**Summary:**

The authors considered the problem of traversing the whole RDP tradeoff using a diffusion model without training. For this purpose, they proposed using the score-scaled probability flow ODE, in the DiffC framework with the RCC module. Theoretically, they claim the score-scaled PF-ODE can achieve the full vector-Gaussian DP tradeoff with a noisy version of the source, and also for the scalar Gaussian RDP. Experimental results on images are also provided, and exhibit some tradeoff behavior.

**Compliance With Llm Reviewing Policy:**

Affirmed.

**Final Justification:**

The rebuttal partially addressed my concern, though there are still issues that are unlikely to be resolved. I've decided to raise the score by 1.

**Key Questions For Authors:**

See weakness comments

**Limitations:**

I did not find a discussion on the limitations. No particular concern on potential negative societal impact.

**Strengths And Weaknesses:**

Strengths:

1. The motivation is well articulated, and I feel diffusion is a good tool for this problem setting.
2. Experiments on images are promising and led to some interesting observations on the behaviors of the proposed approach.

Weaknesses:

1. The work by Freirick et al. 2021 and the work by Qu et al. [1] below established the optimal tradeoff structure for RDP coding under MSE and Wasserstein-2. The optimal solution is essentially a linear combination of two extremes: the MMSE solution and a perfect-perception solution obtained through optimal transport from the MMSE solution. The authors may have a fundamental misunderstanding of the structure of this optimal solution, which led to several issues discussed below.
2. Lemma 1 is either incorrect or not meaningful. For optimal DP tradeoff, the extreme case of \rho=1, which corresponds to perfect realism, should give the optimal transport solution for the MMSE solution. Instead, Lemma 1 currently states when \rho=1, the solution matches the source distribution, without any coupling with the noisy observation. In other words, the claimed result in Lemma 1 only shows distribution realism, not the required coupling as required by optimal DP. As a result, the lemma does not justify the interpolation structure used later.
3. The authors are essentially claiming that the same compression is a "universal" representation, which provides the optimal DP tradeoff. This universality is in fact not theoretically possible for vector Gaussian source in general; see the discussion in [1].
4. Given the linear structure of the optimal DP solution, once the two extreme estimators are available (MMSE and optimal-transport reconstruction), intermediate points follow by linear interpolation. The PF-ODE dynamics are linear, so obtaining intermediate tradeoffs would then be straightforward. However, the paper does not prove that the score-scaled PF-ODE actually recovers these two required extremes, particularly the optimal-transport reconstruction.
5. The theory suggests linear interpolation between two reconstructions, but for natural images, such interpolations will typically produce artifacts. While the diffusion process avoids explicit pixel-wise averaging, it is not clear that the resulting trajectory corresponds to the theoretically optimal DP interpolation for general image distributions. There is a disconnect between the theoretical framework the authors wish to rely on and the practice in image coding.
6. As the authors stated, the RCC module relies on the PFR (Poisson Functional Representation) algorithm to generate the channel output. However, PFR is primarily a theoretical construction and may require an unbounded number of trials in general, making it impractical for real implementations. As a result, the proposed framework currently lacks a computationally feasible encoding mechanism. The authors should clarify whether they evaluate the coding rate using the theoretical prediction or the length of an actually compressed bitstream via an implementable encoder.


In summary,  I believe the paper has some fundamental flaws, even though the PF-ODE approach itself is interesting.

[1] Qu X, Chen J, Yu L, Xu X. Rate-distortion-perception theory for the quadratic Wasserstein space. IEEE Transactions on Information Theory. 2025 Sep 19.

---

> ### Author Rebuttal · Authors · 2026-03-31
>
> _**Responses to Weaknesses 1, 2, and 4: Optimal tradeoff structure, coupling, and extreme points**_
>
> We agree that the optimal DP solution is theoretically constructed via the MMSE and optimal-transport extremes, as established by Freirich et al. (2021) and [1]. **Our framework does not contradict this**; it provides a generative mechanism to recover this structure without manual pixel-space interpolation.
>
> - **On Lemma 3.1**: We would like to clarify that our $\rho=1$ solution is indeed coupled with the noisy observation. As stated in Lemma 3.1 (Page 4, Line 175), the reconstruction is explicitly coupled with the noisy observation $\check{\mathbf{z}}_t$ via the relation $Z\_0(\check{\mathbf{z}}\_t) = \mathbf{A}\_t^{\rho} \check{\mathbf{z}}\_{t} + \mathbf{B}\_t^{\rho} \boldsymbol{\mu}\_0$ for $\rho\in[0,1]$. Specifically, $\mathbf{A}\_t^\rho = \boldsymbol{\Sigma}\_0^{\frac{1}{2}} \boldsymbol{\Sigma}\_t^{-\frac{1}{2}}$ when $\rho=1$. **We will make this coupling result and its specification at $\rho=1$ more explicit in the revision.**
> - **On extreme estimators**: Our score-scaled PF-ODE **exactly recovers the two extreme estimators** identified in Freirich et al. (2021). Assuming a zero-mean Gaussian source:
>   - For $\rho=0$, Appendix C.1 (Page 15) shows $\mathbf{A}\_t^{0} = \sqrt{\bar{\alpha}\_t} \boldsymbol{\Sigma}\_0 \boldsymbol{\Sigma}\_t^{-1}$. Thus, $Z\_0 = \sqrt{\bar{\alpha}\_t} \boldsymbol{\Sigma}\_0 \boldsymbol{\Sigma}\_t^{-1} Z\_t$, which matches the MMSE solution $X^\star$ in Freirich et al. for the case of $Y=Z\_t=\sqrt{\bar{\alpha}}X+\sqrt{1-\bar{\alpha}}N$.
>   - For $\rho=1$, Appendix C.2 (Page 16) shows the solution is $Z_0 = \boldsymbol{\Sigma}_0^{\frac{1}{2}} \boldsymbol{\Sigma}\_t^{-\frac{1}{2}} Z\_t$. This exactly matches the optimal transport solution in Theorem 2 of Freirich et al.: $\hat{X}\_0 = \boldsymbol{\Sigma}\_{X}^{\frac{1}{2}}\boldsymbol{\Sigma}\_{X^\star}^{-\frac{1}{2}}X^{\star} = \boldsymbol{\Sigma}\_{0}^{\frac{1}{2}}\boldsymbol{\Sigma}\_{t}^{-\frac{1}{2}}Z\_t$. Here $X^\star$ is the MMSE solution with covariance matrix $\boldsymbol{\Sigma}\_{X^{\star}} = \bar{\alpha}\_t \boldsymbol{\Sigma}\_0^2 \boldsymbol{\Sigma}\_t^{-1}$.
> - **Optimality of the Trajectory:** We do not assume the intermediate ODE points are optimal simply because the ODE is linear. Instead, Theorem 3.3 and Appendix D explicitly prove that, by adjusting $\rho$, the achievable distortion and perception of our ODE's reconstruction *exactly match* the optimal DP tradeoff curve.
>
> _**Response to W3: Universality**_
>
> We thank the reviewer for the insightful comments. We do not claim our scheme achieves the optimal RDP tradeoff for general vector Gaussian sources (as noted on Page 5, Line 225). Our use of the term "universal" in Remark 4.2 simply meant that there is no need to adjust the encoder but only adapt the value of $\rho$ to achieve $(D,P)$ pairs associated with a fixed rate. **We recognize this terminology caused confusion and will revise Remark 4.2 to clarify this and discuss [1]**.
>
> _**Response to W5: Interpolation artifacts and theory-practice gap**_
>
> The reviewer notes that linear interpolation produces artifacts in images. We completely agree. This is why our method is effective: we interpolate along the data manifold learned by the diffusion prior. While we only claim theoretical optimality for Gaussians, practically, latent diffusion models (like SD2.1) explicitly use a KL-regularized latent space to approximate a standard Gaussian (Rombach et al., 2022), bridging our theory with empirical practice.
>
> Meanwhile, **the core intuition of our method holds for sources beyond Gaussian**: the variance of the reconstruction scales with $\rho$. We illustrate this phenomenon using a mixture of Gaussian sources where $p_X = w_1*\mathcal{N}(u_1, \sigma_1^2) + w_2*\mathcal{N}(u_2, \sigma_2^2)$ and $Y=\sqrt{\bar{\alpha}_t}X + \sqrt{1-\bar{\alpha}_t}N$ with $N\sim\mathcal{N}(\mathbf 0, \mathbf I)$. As shown in Figure R1 in https://diffrdprebuttal.github.io/, the ODE sampling trajectories correctly reflect the intended effect of $\rho$ and $t$. **We will discuss this theoretical limitation in the revision.**
>
> _**Response to W6: On RCC module implementation**_
>
> We evaluate the coding rate **using the length of the actually compressed bitstream from an implementable encoder, not theoretical predictions**. We utilize the practical PFR implementation from Vonderfecht & Liu (2025). We process data in chunks of 12-16 bits, resulting in a bounded candidate pool of $2^{12}$ to $2^{16}$. Generating the PFR samples and encoding a 16-bit chunk takes less than 3 milliseconds. This practical implementation incurs a modest bitrate overhead (<30%) while maintaining acceptable overall encoding and decoding times (see Table 1). **We will include these implementation details in the revision.**

---

> > ### Author Rebuttal · Reviewer_ncaM · 2026-04-02
> >
> > The rebuttal clarified some issues, and I decided to raise the score by 1 for this reason. However, I believe the authors are addressing some simpler questions instead of the more important ones, particularly given the results already available in the literature.
> >
> > As mentioned earlier, the two extreme points are easy to obtain, and the linear interpolation is easy. Since the ODE is linear, it produces a linear interpolation as well. So we immediately know that we can use diffuse to do it. So the real questions are: 1) Is this simple linear interpolation approach indeed causing artifacts in the image generative setting? How severe is it? Is the proposed approach able to solve the issue consistently, and if so, why? 2) The vector Gaussian-like setting where the optimal solution requires some water-filling, the i.i.d. Gaussian in diffusion approach would break. How should this be handled? If there is no solution, a true transversal seems unlikely.
> >
> > About the universality: if the fixed encoding is not RDP optimal, then that tradeoff does not appear to be a meaningful universality.
> >
> > The main theory in the submission is developed for the simpler settings, and the transversal is not truly solved. Therefore, I still remain somewhat negative in my opinion, though less so than before.

---

> > > ### Author Response · Authors · 2026-04-06
> > >
> > > We sincerely thank the reviewer for acknowledging our rebuttal and raising the score. We appreciate the opportunity to clarify these deeper theoretical and practical questions, as they highlight the core contributions of our work.
> > >
> > > 1. **_Obtaining the extreme points_**: The reviewer noted that the two extreme points are easy to obtain. However, existing methods lack a principled way to obtain these estimators **across different noise levels**. For instance, Freirich et al. (2021) propose obtaining the extremes by selecting the best distortion and perception estimators from 11 candidates for a fixed degradation model (i.e., for each specific rate in the RDP case). It is highly inefficient to store pairs of extreme models for every rate. In contrast, **our framework naturally derives these extremes ($\rho=0$ and $\rho=1$) for any compression level using a single model**, and provides the achievability proof of joint RDP optimality for scalar Gaussian sources.
> > >
> > > 2. **_Compared to pixel-level linear combination_**: We would like to first clarify a mathematical point: **While the PF-ODE is linear with respect to the score term, the score function $\nabla \log p_{Z_t}(\mathbf{z}_t)$ is non-linear with respect to the data $X$.** Therefore, scaling the score by $\frac{1}{2}(2-\rho)$ in the ODE does **not** result in a simple linear interpolation in the pixel space. To answer your specific questions:
> > >
> > >    - _"Does simple linear interpolation cause artifacts? How severe?"_: Yes, simple pixel-space linear interpolation between an MMSE image (which is inherently blurry and averaged) and a perfect perception image (which is vivid and rich in detail) causes highly noticeable artifacts, especially at high resolutions. We have provided visual comparisons in Figure R5 (https://diffrdprebuttal.github.io/) and built an interactive slider demo (https://diffrdprebuttal.github.io/diffrdprebuttal2.github.io/) comparing ODE reconstructions with linear combinations at the same BPP and MSE. **The linear combinations exhibit double edges and ghosting, whereas our ODE reconstructions do not.**
> > >    - _"Does the proposed approach solve it consistently, and why?"_: By traversing the DP tradeoff via the ODE trajectory, the generation process is continuously guided by the learned data manifold. This ensures that **intermediate outputs for any $\rho$ remain on or near the natural image manifold**, yielding sharp, artifact-free images that smoothly trade off distortion for perception. The empirical experiments on image datasets verify its effectiveness against other SOTA baselines.
> > >
> > > 3. **_The vector-Gaussian setting_**: We thank the reviewer for raising an excellent theoretical point. For a non-isotropic vector-Gaussian source, achieving strict RDP optimality indeed requires water-filling—which, in our framework, would correspond to assigning different noise levels $t$ at the encoder and different $\rho$ at the decoder across different dimensions.
> > >
> > >    We acknowledge that applying an i.i.d. Gaussian diffusion with uniform $t$ and $\rho$ breaks this strict theoretical optimality. Implementing exact water-filling dimension-by-dimension in practical high-resolution diffusion models is computationally prohibitive. However, the pre-trained score network inherently captures the complex covariance structure of the data. While we do not claim strict theoretical optimality for the vector case, our empirical results (e.g., the updated Figure R2) demonstrate that applying a uniform $t$ and $\rho$ across dimensions still yields a highly effective and competitive DP traversal in practice.
> > >
> > > 4. **_Universality_**: We agree that because strict RDP optimality is difficult to prove for high-dimensional data, the term "universality" must be carefully contextualized. In our paper, we use "universality" to describe operational flexibility rather than strict information-theoretic optimality for arbitrary sources. **We have revised the relevant descriptions in the manuscript to avoid overclaiming and ambiguity.**
> > >
> > > It is standard practice to ground exact theoretical proofs in simpler settings while demonstrating empirical success on complex, high-dimensional datasets. Our work bridges the gap between foundational RDP theory and practical, state-of-the-art image diffusion models. **We view our method as a primary step in exploring the theoretical and empirical potential for flexible RDP traversal using pre-trained models, and we hope it stimulates further theoretical study into optimal and efficient high-dimensional traversal**.
> > >
> > > Thank you again for your rigorous and constructive feedback, which has deeply enriched our work.

---

### Official Review · Reviewer_xbLV · 2026-03-11

**Soundness:** 3
**Presentation:** 3
**Significance:** 3
**Originality:** 3
**Overall Recommendation:** 4
**Confidence:** 4

**Summary:**

This paper proposes a training-free framework that leverages pre-trained diffusion models to traverse the entire RDP surface. The authors combine a RCC module with a score-scaled probability flow ODE decoder. They theoretically prove that the proposed diffusion decoder is optimal for the distortion-perception tradeoff under AWGN observations. And they prove that the overall framework with the RCC module achieves the optimal RDP function in the Gaussian settings.

**Compliance With Llm Reviewing Policy:**

Affirmed.

**Final Justification:**

The authors have addressed most of my concerns. I am pleased to upgrade my score.

**Key Questions For Authors:**

1. More analysis comparing the method with DiffC is needed to clarify the specific core advantages and contributions relative to DiffC.

2. Is it possible to mitigate the high latency issue through engineering or methodological improvements? Currently, the encoding and decoding latencies are too high, and the latency further increases as the bitrate changes. This is a major drawback compared to other current mainstream SOTA generative codecs.

I will raise the score if the authors answer my questions/weakness thoroughly and provide more discussions.

**Limitations:**

The authors may need to extend the analysis beyond Gaussian settings and W2 distance for complicated natural image scenarios.

**Strengths And Weaknesses:**

Strengths:
1. I believe the exploration of the RDP surface and tradeoff issues in this paper is inherently highly significant. The authors prove the optimality of their score-scaled ODE and the overall compression framework under the one-shot assumption and Gaussian setting.

2. Supporting multiple bitrates using a single pre-trained model is a significant advantage compared to existing SOTA ultra-low bitrate models.

Weaknesses:
1. Regarding novelty, the core idea of achieving the RDP tradeoff still essentially relies on interpolation. There is a lack of particularly profound insight, as the method primarily adds a scaling parameter to the DiffC framework, which represents an incremental improvement over DiffC.

2. The inference cost is expensive, requiring multiple steps and multiple ODE iterations.

3. The theoretical analysis is mainly about Gaussian distributions and W2 distance. I am curious whether this is too simplistic for natural images and fails to fully cover the actual scene distributions of natural imagery.

---

> ### Author Rebuttal · Authors · 2026-03-31
>
> _**Response to W1&Q1: On the novelty and comparison to DiffC**_
>
> We thank the reviewer for the comment and question. We would like to clarify that **our method is neither a simple interpolation nor a trivial improvement over DiffC**.
>
> - **A scaling parameter brings a fundamental capability leap to DiffC**: DiffC is limited to a single operating point on the RDP surface: perfect perception (i.e., a special case of our scheme at $\rho=1$). It cannot trade perception for lower distortion (MSE). By introducing a principled scaling parameter, **our framework unlocks the *entire* continuous RDP surface using a single pre-trained model**.
> - **RDP is a non-trivial extension to DP interpolation**: Freirich et al. (2021) address the DP tradeoff for a *fixed* degradation model via linear interpolation. However, lossy compression (RDP) requires the joint design of the encoder (rate control) and decoder. Previous interpolation methods lack a principled way to define or obtain the extreme estimators **across different noise levels**. Our framework naturally derives these extremes ($\rho=0$ and $\rho=1$) for any compression level and provides the achievability proof of **joint RDP optimality for Gaussian sources**.
> - **Beyond simple pixel-level interpolation**: Linear interpolation in pixel space (e.g., Freirich et al., 2021) typically produces visual artifacts. Instead, our score-scaled ODE operates intrinsically within the diffusion process, naturally avoiding these artifacts and preserving perceptual quality.
>
> As noted, **DiffC is mathematically equivalent to our method at $\rho=1$.** The $\rho=1$ data points in our figures directly represent DiffC's performance. Our core contribution is the non-trivial extension that enables flexible traversal of the entire RDP plane, a capability the original DiffC lacks.
>
> _**Response to W2&Q2: On inference cost and possible improvements**_
>
> We thank the reviewer for the concern about the inference cost. While diffusion-based codecs inherently require more compute than standard feed-forward models, our current implementation utilizes DDIM-style sampling (maximum 50 steps) and CUDA-accelerated RCC from Vonderfecht & Liu (2025). As a result, our encoding and decoding times are comparable to HiFiC and CDC, and significantly faster than DDCM.
>
> Meanwhile, **the latency can be reduced through both engineering and methodological improvements.** Because our framework is derived from the reverse ODE and is agnostic to the specific solver or RCC method, it can directly incorporate future advancements. Integrating advanced few-step ODE solvers (e.g., DPM-Solver),  distillation techniques, or faster RCC implementations will directly accelerate our method. **Our primary focus in this work is on establishing the theoretical and empirical foundation for flexible RDP traversal using pre-trained models.**
>
> _**Response to W3: On Gaussian assumption**_
>
> Deriving closed-form RDP functions for non-Gaussian sources remains an open problem in information theory. **Analyzing the Gaussian case is the standard, necessary first step to establish theoretical optimality** (e.g., Zhang et al., 2021) and provides a principled derivation for *why* score scaling controls the tradeoff. Practically, latent diffusion models (like SD2.1) explicitly use a KL-regularized latent space to approximate a standard Gaussian (Rombach et al., 2022), bridging our theory with empirical practice.
>
> Furthermore, **the core intuition of our method holds for non-Gaussian data**: the variance of the reconstruction distribution scales with $\rho$. A smaller $\rho$ yields lower variance (favoring distortion), while a larger $\rho$ yields higher variance (favoring perception).  We demonstrate this using a Gaussian mixture source $p_X = w_1*\mathcal{N}(u_1, \sigma_1^2) + w_2*\mathcal{N}(u_2, \sigma_2^2)$ and $Y=\sqrt{\bar{\alpha}_t}X + \sqrt{1-\bar{\alpha}_t}N$ with $N\sim\mathcal{N}(\mathbf 0, \mathbf I)$. As shown in Figure R1 (https://diffrdprebuttal.github.io/), the ODE sampling trajectories correctly reflect the intended effect of $\rho$ and $t$. **We will discuss this limitation and potential extensions via interpolating ODEs in the revision.**

---

> > ### Author Rebuttal · Reviewer_xbLV · 2026-04-02
> >
> > I still concern about the performance gap and high computational cost, especially compared to recent one-step generative codec (StableCodec [1] , OneDC [2]). They should be compared in details.
> >
> > But the overall response is satisfying. I believe it is meaningful.
> >
> > [1] Taming One-Step Diffusion for Extreme Image Compression
> > [2] One-Step Diffusion-Based Image Compression with Semantic Distillation

---

> > > ### Author Response · Authors · 2026-04-06
> > >
> > > We sincerely thank the reviewer for the encouraging feedback and for satisfying our overall response.
> > >
> > > Following your suggestion, we have compared our method with StableCodec and OneDC regarding DP performance, inference time, and model size. We have updated our manuscript to reflect these discussions. Below is a detailed breakdown of this comparison and further discussions:
> > >
> > > 1. _**DP performance**_: We evaluated our method against StableCodec, OneDC, RDEIC (Li et al., 2025), and DLF (Xue et al., 2025). As shown in the updated Figure R2 (available at https://diffrdprebuttal.github.io/), **our method demonstrates highly competitive DP performance**. For instance, at a bitrate of bpp=0.24, our model (represented by the purple solid curve with dots) achieves significantly lower MSE (around 6.71-6.85$\times 10^{-3}$) than both StableCodec (bpp=0.030, MSE = 7.25$\times 10^{-3}$, FID=57.36) and OneDC (bpp=0.024, MSE = 7.09$\times 10^{-3}$, FID=56.84) while maintaining a similar FID (our FID=55.29 for $\rho=0.85$ and 38.91 for $\rho=1$). Note that **all rate-MSE-FID curves for our scheme are achieved by a single pre-trained diffusion model without any retraining**.
> > >
> > > 2. _**Inference time and model size**_: We acknowledge that one-step models are inherently faster at decoding. As shown in our updated Table 1 (Figure R4 at https://diffrdprebuttal.github.io/). StableCodec and OneDC achieve decoding times of 0.47s and 0.43s, respectively, compared to our 0.33s–2.10s (depending on the chosen timestep $t$). However, **their effective storage cost is much higher if flexibility is required**. Our method utilizes a single pre-trained SD-2.1 model (950M parameters). In contrast, the models of StableCodec (1.1B) and OneDC (1.4B) are larger, and specific modules must be retrained for different bitrates and DP points. **To cover the same wide range of bitrates and DP tradeoffs that our single model traverses, one would need to retrain and store dozens of separate modules** ($N \times 492\text{M}$ for OneDC, and $N \times 109\text{M}$ for StableCodec, where $N$ is the number of RDP points to be traversed), making their practical deployment size prohibitively large for adaptive streaming scenarios.
> > >
> > > 3. _**Flexibility vs. speed tradeoff**_: We would like to emphasize that existing baselines (CDC, HiFiC, StableCodec, OneDC, etc) operate at **fixed points** on the RDP surface. They lack the mechanism to dynamically navigate the balance between distortion and perception at inference time. Our method provides the unique ability to traverse the __*entire* RDP surface__ using a single, off-the-shelf diffusion model. Users can dynamically adjust the bitrate (via $t$) and the DP balance (via $\rho$) on the fly, without any retraining.
> > >
> > > 4. _**Potential acceleration under the proposed framework**_: As noted previously, because our framework is derived from the reverse ODE and is agnostic to the specific solver or RCC method, it can directly incorporate future advancements. Below is more detailed discussion on this point:
> > >
> > >    - **Advanced ODE solvers**: High-order solvers like DPM-solver (Lu et al., 2022) analytically compute the linear part of the ODE, allowing for much larger step sizes. This can potentially reduce the NFE from 50–100 down to 5–10 steps with a negligible performance drop, while remaining a well-defined score-scaled ODE.
> > >    - **Parameterized trajectory distillation** (with one-step additional training): Techniques like consistency trajectory models (Kim et al., 2024) can be adapted to distill a family of ODEs. By treating $\rho$ as a conditioning variable (similar to timestep $t$), a student network can be trained to map any noisy latent $Z_t$ directly to the reconstructed $X_0^\rho$ in one step.
> > >
> > >    While a full empirical investigation of these acceleration techniques is beyond the time limits of this rebuttal, we believe our proposed method provides a theoretically grounded foundation for training-free RDP traversal and will stimulate future studies.
> > >
> > > The comparison highlights a fundamental tradeoff in current generative compression: **highly optimized, task-specific distillation** (StableCodec/OneDC) versus **training-free, universal flexibility** (Ours). We believe both directions are highly valuable, and future work could potentially integrate our traversal mechanisms with distilled one-step models to achieve both ultra-fast inference and RDP flexibility.
> > >
> > > Thank you again for helping us strengthen the paper.
> > >
> > > > Li et al., RDEIC: Accelerating Diffusion-Based Extreme Image Compression with Relay Residual Diffusion, TCSVT2025.
> > > >
> > > > Xue et al., DLF: Extreme Image Compression with Dual-generative Latent Fusion, ICCV 2025.
> > > >
> > > > Lu et al., DPM-Solver: A Fast ODE Solver for Diffusion Probabilistic Model Sampling in Around 10 Steps, Neurips 2022.
> > > >
> > > > Kim et al., Consistency Trajectory Models: Learning Probability Flow ODE Trajectory of Diffusion, ICLR 2024.

---

### Official Review · Reviewer_xzEF · 2026-03-12

**Soundness:** 3
**Presentation:** 3
**Significance:** 2
**Originality:** 2
**Overall Recommendation:** 4
**Confidence:** 4

**Summary:**

This paper proposed a socre-scaled probability flow ODE decoder to enhance diffc algorithm to achieve RDP optimality. And provide theoretical proof under AWGN channels with scalar Gaussian data. The proposed method can achieve full traversal of RDP surface.

**Compliance With Llm Reviewing Policy:**

Affirmed.

**Final Justification:**

My concerns have been resolved and I will raise my score to 4.

**Key Questions For Authors:**

- According to Table 1, DDCM is significance slower than the proposed method, but they both use SD 2.1. Could the authors clarify the reasons? Are they use different denoising steps or sampling schedules?
- While the RDP optimality is proved in pixel-space, practical applications often rely on latent spaces. What are the specific theoretical differences when implementing this framework in latent-space? How can the "optimality" be guaranteed when the latent distribution deviates from the Gaussian assumption?

**Limitations:**

No, the authors could give more comparisons with SOTA baselines.

**Strengths And Weaknesses:**

### Strengths

- The paper provides a comprehensive exploration of the Rate-Distortion-Perception (RDP) optimization problem.
- The paper provides a theoretical proof of RDP optimality under the specific conditions of scalar Gaussian data and AWGN channels.

### Weaknesses

- Lack of novelty: The proposed method seems to be the combination of DiffC and the proof of [1], primarily by introducing the parameter $\rho$ to achieve it. However, it fails to address the limitation of DiffC, such as high computational complexity.
- Strong assumption: The theoretical proof assumes scalar Gaussian data. However, the actual image data are highly non-Gaussian. Furthermore, for latent diffusion models, the features in latent space are not strictly follow a Gaussian distribution, leading to sub-optimal performance in empirical experiments. The comparison between SD2.1 and DDCM in Figure 5 proved it.
- Experiment:
    - Figure 5 is not clear to compare the DP performance under the similar bpp.
    - The paper lacks comparisons with some SOTA diffusion-based methods[2,3,4,5] and token-based methods[6,7].
    - There is no comparison with original DiffC[8], which is essential to quantify the gains and the innovation of the proposed method.

Reference

[1] Freirich, Dror, Tomer Michaeli, and Ron Meir. "A theory of the distortion-perception tradeoff in Wasserstein space." NeurIPS 2021.

[2] Li Z, Zhou Y, Wei H, et al. Toward extreme image compression with latent feature guidance and diffusion prior[J]. TCSVT2024.

[3] Li Z, Zhou Y, Wei H, et al. RDEIC: Accelerating Diffusion-Based Extreme Image Compression with Relay Residual Diffusion[J]. TCSVT2025.

[4] Zhang T, Luo X, Li L, et al. StableCodec: Taming One-Step Diffusion for Extreme Image Compression[J]. ICCV2025.

[5] Ke A, Zhang X, Chen T, et al. Ultra Lowrate Image Compression with Semantic Residual Coding and Compression-aware Diffusion[J]. ICML2025.

[6] Jia Z, Li J, Li B, et al. Generative latent coding for ultra-low bitrate image compression[C]. CVPR 2024.

[7] Xue N, Jia Z, Li J, et al. DLF: Extreme Image Compression with Dual-generative Latent Fusion[J]. ICCV 2025.

[8] Vonderfecht J, Liu F. Lossy compression with pretrained diffusion models[J]. ICLR 2025.

---

> ### Author Rebuttal · Authors · 2026-03-31
>
> _**Response to W1: On novelty**_
>
> We would like to clarify that the primary focus of the proposed framework is not to address the limitation of DiffC (e.g., high computational complexity). Instead, __our core contribution is enabling full RDP traversal using a single pre-trained model.__ Achieving this requires a non-trivial extension that goes well beyond simply combining DiffC and Freirich [1]:
>
> - **Fundamental difference between DP and RDP:** Freirich [1] addresses the DP tradeoff for a *fixed* degradation model. Lossy compression (RDP) requires the joint design of the encoder (rate control) and decoder. [1] lacks a principled way to obtain the extreme estimators **across different noise levels**. Our framework naturally derives these extremes ($\rho=0$ and $\rho=1$) for any compression level.
>
> - **Methodology:** [1] uses pixel-level interpolation, which produces visual artifacts. Instead, our score-scaled ODE operates intrinsically within the diffusion process, naturally avoiding these artifacts and preserving perceptual quality.
>
> - **Theory:** Neither DiffC nor [1] proves joint RDP optimality. We provide the achievability proof for traversing the optimal RDP surface in the Gaussian case. Under our broader framework, __the original DiffC is a special case representing the perfect-perception extreme.__
>
> _**Response to W2: Gaussian assumption**_
>
> Deriving closed-form RDP functions for non-Gaussian sources remains an open problem. **Analyzing the Gaussian case is the standard, necessary first step to establish theoretical optimality** (e.g., Zhang et al., 2021) and provides a principled derivation for *why* score scaling controls the tradeoff. Practically, latent diffusion models (like SD2.1) use a KL-regularized latent space to approximate a standard Gaussian (Rombach et al., 2022), bridging our theory with empirical practice.
>
> Furthermore, **our core intuition holds for non-Gaussian data**: reconstruction variance scales with $\rho$. A smaller $\rho$ yields lower variance (favoring distortion), while a larger $\rho$ yields higher variance (favoring perception).  We demonstrate this using a Gaussian mixture source $p_X = w_1*\mathcal{N}(u_1, \sigma_1^2) + w_2*\mathcal{N}(u_2, \sigma_2^2)$ and $Y=\sqrt{\bar{\alpha}_t}X + \sqrt{1-\bar{\alpha}_t}N$ with $N\sim\mathcal{N}(\mathbf 0, \mathbf I)$. As shown in Figure R1 (https://diffrdprebuttal.github.io/), the ODE sampling trajectories correctly reflect the intended effect of $\rho$ and $t$. **We will discuss this limitation and potential extensions via interpolating ODEs in the revision.**
>
> _**Response to W3 and comparison with DDCM**_
>
> - **On Figure 5 and comparison with DDCM**: We thank the reviewer for the constructive comment. We have updated the MSE-FID plots with explicit bpp labels near the curves (Figure R2, https://diffrdprebuttal.github.io/).  It can be observed that **our method outperforms DDCM at comparable bitrates.** For instance, at $\rho=1$, our model (bpp=0.080, MSE = 3.79$\times 10^{-3}$, FID=25.9325) achieves significantly lower MSE than DDCM (bpp=0.099, MSE = 5.23$\times 10^{-3}$, FID=26.4484) while maintaining a similar FID.
> - **More SOTA comparisons**: We thank the reviewer for providing more SOTA baselines. We have evaluated against REEIC [3], StableCodec [4], and DLF [7] on Kodak (See Figure R2). Our method yields superior performance (e.g., at bpp=0.024, we outperform StableCodec at bpp=0.030 and REEIC at bpp=0.0245). Crucially, unlike these baselines, our method flexibly traverses the RDP plane using a single pre-trained model. **We will include these comparisons and a full discussion of [2-7] in the revision.**
> - **Comparison with DiffC**: As noted, **DiffC is mathematically equivalent to our method at $\rho=1$.** The $\rho=1$ data points in our figures directly represent DiffC's performance. Our core contribution is the non-trivial extension that enables flexible traversal of the entire RDP plane, a capability the original DiffC lacks.
>
>
>
> _**Response to Q1**_: Yes, the official implementation of DDCM uses 1000 DDPM steps and requires solving a codebook optimization problem at every step. In contrast, our scheme uses a DDIM-style schedule with fewer than 50 steps (depending on the target bitrate). Thus, even with the addition of RCC module, our approach remains faster than DDCM.
>
> _**Response to Q2**_: The theoretical bridge relies on the VAE, which is trained to faithfully reconstruct images. Statistical properties in the latent space correlate strongly with those in pixel space. Therefore, preserving the distribution or reconstructing the mean in the latent space effectively achieves the same goals in the pixel space.
>
> While strict theoretical optimality isn't guaranteed if the latent distribution deviates from a perfect Gaussian, we empirically observe highly effective RDP traversal across models (e.g., SD2.1, Flux). **We will detail this theoretical gap and the autoencoder's role in the revision.**

---

> > ### Author Rebuttal · Reviewer_xzEF · 2026-04-03
> >
> > Thanks for the authors for the detailed rebuttal and more results. My concerns have been resolved and I will raise my score to 4.

---

> > > ### Author Response · Authors · 2026-04-06
> > >
> > > We sincerely thank the reviewer for acknowledging our rebuttal and raising your score. We truly appreciate the time and insightful feedback you have provided throughout the review process. Your comments have been instrumental in improving the quality and clarity of our manuscript.

---

### Decision · Program_Chairs · 2026-04-30

**Decision:**

Accept (regular)

**Comment:**

After the rebuttal and discussion, the consensus leaned slightly toward acceptance. Reviewers agreed that the paper presents a technically interesting and practically relevant training-free framework for traversing the rate-distortion-perception tradeoff, with promising empirical results and competitive baseline comparisons. The main remaining concern is that some of the theoretical claims, especially regarding the optimality interpretation and the breadth of traversal beyond the analyzed settings, are not yet fully convincing to all reviewers. Nevertheless, the rebuttal substantially improved the clarity of the theory, added important implementation details, and strengthened the empirical support. On balance, the AC recommends acceptance and encourages the authors to include the key rebuttal clarifications and additional results in the camera-ready version.